# Bioprinting of Cells, Organoids and Organs-on-a-Chip Together with Hydrogels Improves Structural and Mechanical Cues

**DOI:** 10.3390/cells13191638

**Published:** 2024-10-01

**Authors:** Claudia Tanja Mierke

**Affiliations:** Faculty of Physics and Earth System Science, Peter Debye Institute of Soft Matter Physics, Biological Physics Division, Leipzig University, 04103 Leipzig, Germany; claudia.mierke@uni-leipzig.de

**Keywords:** collagen, viscosity, cell–matrix bidirectional interaction, 4D bioprinting, organoids, cancer, polymers, stiffness, assembloids, tumoroids

## Abstract

The 3D bioprinting technique has made enormous progress in tissue engineering, regenerative medicine and research into diseases such as cancer. Apart from individual cells, a collection of cells, such as organoids, can be printed in combination with various hydrogels. It can be hypothesized that 3D bioprinting will even become a promising tool for mechanobiological analyses of cells, organoids and their matrix environments in highly defined and precisely structured 3D environments, in which the mechanical properties of the cell environment can be individually adjusted. Mechanical obstacles or bead markers can be integrated into bioprinted samples to analyze mechanical deformations and forces within these bioprinted constructs, such as 3D organoids, and to perform biophysical analysis in complex 3D systems, which are still not standard techniques. The review highlights the advances of 3D and 4D printing technologies in integrating mechanobiological cues so that the next step will be a detailed analysis of key future biophysical research directions in organoid generation for the development of disease model systems, tissue regeneration and drug testing from a biophysical perspective. Finally, the review highlights the combination of bioprinted hydrogels, such as pure natural or synthetic hydrogels and mixtures, with organoids, organoid–cell co-cultures, organ-on-a-chip systems and organoid-organ-on-a chip combinations and introduces the use of assembloids to determine the mutual interactions of different cell types and cell–matrix interferences in specific biological and mechanical environments.

## 1. Introduction

In medicine, biomedical and biophysical research, it is commonly known that there is and will be a continued massive demand for tissues and organs for transplantation, experimental drug screening and the fundamental analysis of developmental and diseased processes of cells or cell clusters in tissues. As there are not enough organs available for patients, there is no question that tissues and organ models must be developed and engineered for research, which is the overall purpose for organoid cultures in general. For many decades, biologists have employed two-dimensional (2D) culturing, veterinary models or dead bodies to gain precious knowledge about disease mechanisms, drug screening and safety (toxicity) studies, but the transfer to humans is debatable with regard to its accuracy, validity, significance and repeatability [1,2]. In addition, due to their limited efficacy, more durable, perhaps more radical, replacement methods are needed for existing cell therapy approaches for the treatment of chronic diseases. An extremely important task for tissue engineering is to replace or possibly avoid animal testing altogether, which is also a commonly defined well-known aim in the field. This is highly desirable from a bioethical point of view and can also be the answer to practical problems arising from species-specific variations in cell function and tissue organization [3]. Moreover, realistic three-dimensional (3D) tissue models are increasingly needed for toxicological in vitro screening and drug development [4]. Advanced tissue engineering via bioprinting of 3D organoids is occupying an emerging key role in biomedical, cell biological and biophysical or mechanobiological research [4,5].

Of the tissue engineering techniques, 3D bioprinting is set to revolutionize the biomanufacturing of tissues and organs [6]. Scaffolds have been utilized as a possible vehicle for producing artificial organs and tissues prior to the emergence of the 3D printing of living cells [7]. While scaffolds are not a novel tissue engineering methodology, their traditional techniques and structures entail multiple constraints for the generation of efficacious tissues and organs. The main restriction of scaffolds is that they are not able to imitate the inherent functionalities of an extracellular matrix (ECM), as the precise mechanisms of their functioning are still not completely comprehended. In addition, traditional methods of scaffold production also fail to replicate the natural mechanical characteristics that are essential for correct biological performance [7], such as the issue of inhomogeneity of natural ECMs, which can only be partially reproduced in conventional hydrogel cell culture models [8]. In contrast, the 3D bioprinting process for tackling these issues is built on the principles of biomimicry, autonomous self-organization and microtissues, which are typically intricate in vivo tissues composed of several simpler units where the composite structure and function form the complete picture [9,10].

The combination of in vitro cultured 3D cell structures such as organoids, which can self-organize and mimic real organs in structure and function, with bioprinting techniques appears to be highly suitable for rapidly advancing model systems and offers enormous potential for physiological and pathological questions. Organoids come from stem cells or tumor tissue (tumor organoids) taken from patients and are grown in vitro in a specific 3D microenvironment. The integration of tumor organoids into 3D-printed tumor models is currently in its fledgling stages but provides new opportunities for more precise tumor microenvironment (TME) reconstitution. The TME is composed of non-malignant cells (immune cells, stromal cells and (tumor) endothelial cells) and ECM protein contributing to a 3D ECM scaffold [11]. Non-malignant cells can interfere among each other and with cancer cells, such as those of solid tumors, which consequently critically affect cancer biological analyses [12]. In addition, the TME is intricate and required for tumor growth, and hence, simplifying tasks are needed for gaining insights [13]. The cutting-edge technique of 3D bioprinting accurately mimics collective cell performance, individual patient-specific physiological properties, and the accurate monitoring of geometric, biophysical, and biochemical parameters in the TME, such as cell-derived products released via extracellular vesicles. Thereby, the mechanical environment of cells and 3D organoids can be rebuilt [14]. Moreover, by continuing to implement organ-on-a-chip technology in biomedical and biophysical research, fluid dynamics and immune cells, such as macrophages and neutrophiles [15], can be incorporated in an easy manner, which can then remodel the external environment of cells by mechanical signals, and thus, for example, enhance the accuracy of drug screenings.

Tumor organoids are increasingly deployed as in vitro models in cancer research due to their ability to recapitulate the complexity of tumors [11]. Such patient-derived cancer cell assemblies can closely replicate key tumor characteristics, comprising tumor-like cell–cell junctions, heterogeneous cell populations, (epi-)genetic environment and parental tumor growth characteristics [16]. Following the triumph of the first tumor organoids derived from colorectal cancer tissue by Sato et al. [17], organoids of diverse cancer types have been evolved with considerable impact for the modeling of cancer heterogeneity and personalized medicine. As organoid culturing systems have progressed, several trials have established biobanks of patient-derived tumor organoids (PDOs) for bladder [18], brain [19], breast cancer [20], cervical cancer [21], colorectal cancer [22], esophageal cancer [23], gallbladder cancer [24], gastrointestinal cancer [25], glioblastoma (GBM) [26], head and neck cancer [27], liver cancer [28], lung cancer [29], ovarian cancer [30], pancreatic cancer [31], prostate cancer [32], thyroid cancer [33] and multiple other cancer types. PDO biobanks offer a resilient foundation for patient-specific high-throughput screening of therapies, incorporating chemotherapy, immunotherapy and radiotherapy [20,29,34], and warrant multiple trials on tumor advancement and TME hallmarks [28,35].

Besides heterogenous cancer cells, the value of including heterogeneous stromal cell types in in vitro tumor models is increasingly acknowledged [11,36], and it has been shown to impact mechanical cues of the TME [36]. In turn, tumors can reprogram nearby or remote (via extracellular vesicle release) supporter cells into activated subtypes to sustain cancer progression, promote resistance to chemotherapy and circumvent the immune defense reaction [37]. Integrating these heterogeneous stromal cell types and regulation of their switch aids in obtaining substantial tumor-specific expression data profiles in in vitro tumor organoid models. Whereas standard culture conditions of PDOs eliminate specific key features of cancers, innovative co-cultures of PDOs with matching cancer-associated fibroblasts (CAFs) sourced from the same patients have been shown to increase transcriptome stringency and the direct subtype-specific expression of stromal genes [38]. Thereby, the mechanical cues of the environment can be mirrored, as CAFs can stiffen the surrounding ECM via lysyl oxidase (LOX) enzyme crosslinking of collagen I [39] and hence, indirectly alter cellular shape, mechano-phenotype and function [40]. Moreover, the composition of the ECM and elevated ECM stiffness, which are both major contributors of cancer’s progressive and invasive capacity, whereby the TME stiffness as emerged as a biomarker for cancer [41]. Consequently, the ECM stiffness impacts the outcome of cancer therapies [42]. Similarly, a co-culture model that paired PDOs with endogenous tumor-infiltrating lymphocytes as cohesive entities has been found to adequately capture the tumor-infiltrating immune milieu [43]. These organoid frameworks also fostered the emergence of new immunotherapies and made it possible to examine the tumor reaction to anti-programmed cell death protein 1 (PD-1) and anti-programmed death ligand 1 (PD-L1) therapies through uncoupling the cancer-infiltrating and cancer-surrounding elements, such as tumor stroma modifications or mechanical characteristics alterations [44].

Tumor organoids hold tremendous power for rebuilding 3D architecture and heterogeneous cellular elements. They offer distinct advantages over other culture models, among them an improved capacity to mimic the physiological and pathological condition of tumor organs, a moderate expense and an improved compatibility with several new emerging technologies. However, they are inherently restricted in their capacity to mimic other crucial elements of the TME, like tumor-specific biochemical/biophysical characteristics, anatomical sizing, hierarchical blood/lymphatic vasculature and fluid dynamics [45,46]. The application of 3D bioprinting with tumor organoids can transcend these limitations to create comprehensive, concise models with greater clinical efficacy [1,47,48]. Moreover, this technique can reduce the animal testing for drug screening approaches in the future [1]. Alongside the applications of 3D printing in cancer research, tissue engineering approaches and the generation of functional biological structures that are able to substitute or rebuild compromised tissue have continued to advance in recent years and have affected a broad spectrum of medical disciplines, encompassing the bioprinting of vascular channels [49], osseous implants [50], dermal transplants [51], intestinal transplants [52] and heart tissue [53].

The purpose of the review is to discuss the recent progress in 3D bioprinting and new 4D bioprinting approaches through which the structural and mechanical cues of ECM environments can be rebuilt in a more precise and reliable manner from a biophysical or mechanobiological perspective. These techniques also enable the dynamic restructuring of mechanical cues. Bioprinting technology overcomes certain constraints of organoid production, allowing it to be increasingly utilized in drug testing, regenerative medicine, cancer research, and mechanobiology. The focus of the overview is on its use and applications in cancer research; however, other biological applications in healthy and diseased individuals are also included to illustrate the broad application of this technique. Special attention is paid to the interaction of 3D-printed cells and the surrounding hydrogels in 3D cell printing to produce organoids. Aspects covered in the review comprise a brief overview of bioprinting techniques, including extrusion bioprinting, microextrusion, inkjet bioprinting and stereolithography (SLA), laser-induced forward transfer (LIFT), an introduction to the role of bioinks, the physiochemical and biological properties of polymeric hydrogels (natural, synthetic and hybrid), scaffold-free and scaffold-based bioprinting, the combination of organoids or organ-on-a-chip and bioprinting techniques, the role of cell alignment in printed scaffolds and finally a brief outlook on future developments in the field of mechanobiology. Subsequently, the relevance of organoids compared to the organ-on-a-chip technology and important architectural factors for the design of integrative organotypic tumor models are emphasized. In the future, the further development of microfluidic systems, controlled mechanical stimulation, advanced organoid models and four-dimensional bioprinting technology could help to create better bioprinted organoids.

## 2. Brief Overview of Major Bioprinting Techniques for Cells, Hydrogels and Organoids

In this section, a brief and basic introduction to the complex field of major bioprinting techniques is provided from a cellular biophysical perspective. The advanced reader is referred to more detailed review articles [54,55]. The first fast layer-by-layer prototyping technologies for the production of 3D designs was invented in the 1980s and was used for SLA-based 3D printing [56,57]. New techniques such as selective laser sintering (SLS), fused deposition modeling (FDM), laminate object manufacturing (LOM) and electron beam melting (EBM) have all been pioneered for a broad array of materials [58], such as metals [59], ceramics [60] and diverse thermoplastics [61]. The layer-by-layer fabrication ability of intricate constructs is specifically beneficial for the creation of in vitro tissue-engineered models consisting of cells and other biological materials. Different types of 3D printing have been explored for the production of complex tissues like bone, cartilage, heart, muscle and liver [57,62,63,64,65]. Particularly for tissue engineering purposes, there are continuous novelties in the development of different synthetic and natural polymers [31,66,67], nanomaterials [68,69,70], high internal phase emulsions (HIPEs) [71], ceramic composites [72,73,74], decellularized extracellular matrix (dECM) materials [75,76] and conductive materials [77,78,79] for 3D printing applications. Cells and bioactive molecules are embedded in the bioink, which increases the complexity of the material and the overall approach. It is easy to incorporate vascularization and compartmentalization into 3D organoids in a pre-designed manner, which requires changes to the printing process, such as temperature, pH, printing speed and mechanical pressure when bioprinting cells [80].

Multiple 3D printing techniques have been advanced for the bioprinting process, among them inkjet, (micro)extrusion, laser and SLA printing [81]. The focus lies on microextrusion bioprinting processes, as these are most often utilized for hydrogel materials. Microextrusion processes employ uninterrupted pneumatic pressure or mechanical forces driven by motors or screws to eject the bioink out of the printing nozzle as a continuous filament [55]. In contrast to 3D printing with a thermoplastic material, the printing parameters such as temperature, pH value and pressure as well as the material characteristics of the bioink are severely limited when living cells are deployed for bioprinting [82].

These bioink materials often suffered from a general lack of biological performance, problems with cell survival and the creation of intricate architectures. Although much progress has been made in these initial pioneering efforts, a remaining obstacle in this field is the suitability of bioinks to support cells at all stages of the printing process. In particular, firstly, cell survival during suspension in the syringe; secondly, cell survival during extrusion from the nozzle; thirdly, cell survival at all stages of material bonding and fourthly, not least, survival as the finished construct ages and takes on a tissue-like form after printing [83].

### 2.1. Microextrusion or Extrusion Bioprinting (Screw, Piston, Pneumatic)

Extrusion (or microextrusion) bioprinting technology is an upcoming technology in which biomaterials are accurately layered with living cells (termed bioink) to form 3D functional structures for tissue engineering (Figure 1).

Printability, the ability to build and maintain a replicable 3D structure, and cell viability (surviving cells in the printing process) are two of the critical features of the extrusion bioprinting process [84]. Extrusion bioprinting is extensively utilized to produce cell-integrated designs for tissue engineering with manufacturability and cell viability being two key aspects [85]. The discrepancy between printed and engineered structures is a major difficulty, and limits progress in mimicking native tissue organs or TMEs for use in tissue engineering and cancer research [80]. Among the many factors that can influence the printability of structures are the bioink characteristics, the settings of the printing technique and the shape of the structure [86]. An advantage of the extrusion bioprinting technique are the viscosity regulation of the bioink can improve the printing process [85]. In addition, nanoparticles can be included for mechanical analysis [87], various types of crosslinking, such as reversible, chemical, physical and enzymatic, multiple available bioinks and the on-going development of new bioinks, printing of dispersed cells, cell spheroids and tissue strands and it is widely used technique [88]. Major disadvantages of the extrusion bioprinting technique are its limited resolution: cells cannot be precisely pattered and organized. In addition, the bioprinting process could induce quantifiable cell death caused by alterations in dispensing pressure, nozzle geometry, printing time and bioink concentration. Moreover, the bioinks for extrusion bioprinting need to perform “liquid to solid” transition at the right time. However, the application of cell spheroids is limited, as they should not be too large, as otherwise the core cells will become inactive or necrotic due to a lack of oxygen [89]. It is challenging for extrusion bioprinting to recreate the blood supply reticulation. In addition, the extrusion of the bioink from the nozzle using pneumatic pressure or mechanical force by means of a piston or screw can mechanically stimulate the cells and change their function [90]. The key benefit of extrusion bioprinting over other processes is the capability to embed cells into the biomaterials for printing structures, whereas the process-related forces may compromise the embedded cells (or the viability of the cells)—another non-negligible concern in the field of bioprinting [91]. Strain stress and shear stress represent two important process-related forces that cause cell injury. Some key factors like needle type and size, the concentration of bioink and dosing pressure contribute to cell deterioration. Despite numerous promising investigations of the printability and viability of cells, this research area is in its infancy and the accurate identification of effective factors continues to be a fundamental concern for future advances. A trial-and-error determination or improvement approach is costly, challenging, tedious and, at times, infeasible; thus, computer-aided techniques are emerging as powerful instruments. Many interdependent factors are relevant to the improvement of the bioprinting procedure [61,92]. Machine learning is a new technology that can be used in the field of 3D bioprinting to greatly advance this technology [93]. The biggest challenge in the further development of machine learning in the field of bioprinting is presently the scarcity of existing data. For this reason, it is necessary to create a global database sharing system for bioprinting. Sharing data due to the different brands of bioprinters and software across the globe can raise many new questions [94]. Therefore, standardized data for every bioprinter using similar open-source software for all the printers seem auspicious. Machine learning, although new in the field of bioprinting, is anticipated to transform bioprinting and thus tissue engineering in the coming years [93]. In addition, (micro)extrusion bioprinting facilitates the production of heterogeneous structures exhibiting high form accuracy by depositing a bioink possessing the targeted physicochemical and biological properties [95]. A novel semi-synthetic hydrogel, composed of gelatin methacrylate and Pluronic F127, has been custom formulated to meet the demands of the (micro)extrusion bioprinting technique [96]. The combination of the thermosensitive properties of Pluronic with the crosslinking properties of gelatin methacrylate provides the compound with a printability range offering good dimensional stability and chemical stability after photocrosslinking [97], as revealed by a rigorous evaluation of printability using predictive empirical models. The mechanical characteristics of the structures are similar to that in soft tissue, which expands the scope of soft tissue engineering. The bioink has been effectively used to produce multilayer porous assemblies that retain a high degree of cell viability [95]. Interestingly, the spatial layout of the cells exhibited a high level of alignment following the direction of deposition [98]. Finally, this manufacturing process could provide a versatile approach for the creation of 3D models with a predefined cellular orientation [99]. In the specific case of tissue engineering, (micro)extrusion bioprinting has been applied in various areas, from the production of vascular prostheses to skin grafts toward 3D organoids with a vasculature [100,101]. This technique is based on the ejection of ink through mechanical or pneumatic forces. Compared to other techniques, such as inkjet, the advantage of this technique is that high-viscosity liquids and structures with a very high cell density can be processed. (Micro)extrusion bioprinting involves extruding the bioink straight in its gel phase, with no support structures or cross-linking substances introduced to the material as it flows out [63]. Therefore, the printing approach is highly reliant on the rheological characteristics of the bioink, which is made of a biomaterial hydrogel harboring living cells [102,103,104,105]. In this setting, a key difficulty is the design of appropriate inks that exhibit both the desired properties of extrudability and stability after printing [106]. Hence, multiple mixtures of natural and synthetic biomaterials with complementary characteristics, such as shear thinning or strain hardening, have been analyzed to generate appropriate chemical, mechanical and biological attributes for intended purposes [107,108]. Multicomponent inks have proven to be an ideal choice to tackle the disadvantages of single-material formulations, like the restricted printing ability of natural polymers or the deficiency of cell-specific activity associated with synthetic polymers, thereby incorporating the benefits of every ingredient and extending the range of biofabrication [109]. When various hydrogels are blended together, the final polymer blend can benefit from different crosslinking mechanisms, including both chemical and physical crosslinking modes, resulting in robust gel stability [57,110]. This approach has been widely pursued and material systems with improved printability have been identified for the manufacture of functional tissues featuring intricate architectures [107,111,112,113,114,115]. Pluronic F127 (PF127), a poly(ethylene oxide)-poly(propylene oxide)-poly(ethylene oxide) triblock copolymer, has been utilized as a sacrificial template medium in a multi-component proprietary design to define the entire thermal performance of the bioink and facilitate the straight laydown of low viscosity alginate [116,117], that is realized to have inadequate extrudability in its non-crosslinked form [96], increasing the bulk porosity of the hydrogel scaffold upon its withdrawal.

A new type of proprietary formulation of Pluronic F127 and gelatin methacrylate (GelMA) has been presented that optimizes the printability and biocompatibility. GelMA is a natural-based hydrogel of great potential because it is photohardenable by gelatin functionalization with methacrylate groups and can be manipulated easily [97]. For instance, its mechanical characteristics can be easily controlled by adjusting several key factors like the level of substitution and the amount of polymer, while its rheological characteristics are mainly governed via the processing temperature and dosage of UV radiation [118,119,120]. GelMA can also be an optimal biomaterial for tissue engineering purposes, as it strongly mimics the tissue microenvironment because of the existence of natural cell adhesion moieties and provides the expected amount of bioactivity anticipated from a tissue engineering framework [121,122,123,124]. In (micro)extrusion-based bioprinting, the viscosity limitations of the bioink, the gelling time and operating parameters, including pressure and printing velocity, are critical for the deposition of fibers with a predefined shape and scalable dimensions [125,126,127]. Besides other characteristics, an optimal bioink needs to have a distinct shear thinning characteristic and be able to regain its viscosity at a standstill directly following extrusion [128].

### 2.2. Inkjet Bioprinting (Piezoelectric Actuator, Heater, Jet-Based, Drop-on-Demand)

The advantages of low price, high efficiency and high accuracy have ensured that inkjet printing is widely used [129]. The inkjet bioprinting process involves the production of small, size-adjustable droplets of liquid ink, which are applied to the surface of the substrate at specific points (Figure 2).

Besides its exceptional efficiency and high accuracy, inkjet technology possesses several unique properties, such as drop-on-demand technology and non-contact material feed [130]. As a result, it has also been utilized for printing electronics equipment components, structural polymers, sol-gel materials, biomolecules and living cells since approximately the year 2000 [131,132,133,134]. After the concept of 3D printing was unveiled, inkjet printing has been progressively customized for 3D manufacturing. Three-dimensional inkjet printing technology has attracted a great deal of interest in numerous research fields, such as polymer molding [61,135], nanocomposites [136], drug delivery systems [137], organ and tissue engineering [55,138,139,140] and the generation of cancer model systems [141]. Inkjet printing is a pioneering 3D printing platform for cell printing with the purpose of manufacturing whole tissues or organs. Based on this advancement, the new concept of “inkjet bioprinting” is a branch of “bioprinting” that has been formerly characterized and authorized [63,142,143]. In short, “inkjet bioprinting” refers to the manipulation, structuring or assembling of biologically important substances, like generic biomaterials, biomolecules and cells, in a prefabricated way to serve specific biological purposes using inkjet fabrication technology. In addition, inkjet bioprinting has been integrated with these established key technologies to promote their further development. Inkjet technology can simultaneously deposit a wide spectrum of materials and cells to specific locations in a non-contact, tailored way, providing the opportunity to construct intricate heterogeneous biomimetic patterns. The droplet size generated by inkjet printing can reach the order of picoliters, which is suitable for the high-precision positioning of micro-scale biological entities in a digitized design. In addition, the diameter of the nozzle of inkjet printers is generally around 50 μm, which falls in the same size range as cells and thus opens up the possibility of printing cells and individual cells [144].

In 1988, collagen and fibronectin were printed with an thermal inkjet device [145]. In 1997, poly(lactic-co-glycolic acid) (PLGA) and poly-lactide (PLLA) scaffolds were fabricated with 3D inkjet printing technique [146]. The frameworks were colonized with primary hepatic cells and endothelial cells, whereby the cell mixture attached to the framework and a specific tissue pattern could be rebuilt. In 2000, denatured DNA was loaded into the ink cartridge of an thermal inkjet printer and printed out as a text design [147]. The prepared sample was then hybridized with the printed DNA array, which led to the formation of a distinctly visible hybridized array. This confirms the suitability of inkjet printing for producing DNA arrays and illustrates the huge, tremendous possibilities of inkjet printing [148]. Bioactive proteins can also be applied by inkjet printing and their folding conformation can be preserved post-printing [149]. A protein-containing buffer suspension has been deposited on a glass plate to form a protein microarray with spots of approximately 150–200 μm in diameter. This technique has been employed to investigate protein–protein, enzyme–substrate and protein–small molecule interactions, demonstrating both the activity of the printed proteins and the wide range of potential applications. In 2003, Biotin was printed with an inkjet printer and the idea of printing cells with inkjet technology was raised for the first time [150]. The first printing process with living organisms was performed with bacteria [151]. The bacterium Escherichia coli was selected to print in precise designs and density gradient arrangements, demonstrating that cell ink, like ordinary ink, can create specific designs. Mammalian cells are bigger than bacteria and are easier to injure when being printed, so printing them is a greater difficulty. A thermal inkjet printer was employed to print Chinese hamster ovary cells on soy agar gel medium and collagen gel medium, reaching a cell survival rate of more than 90% [152]. In 2009, a half-heart with connected ventricles was printed [153]. Later, chondrocytes were printed in an effort to create artificial cartilage [154]. Since natural animal tissues consist of several types of cells with intricate architectures that cooperate to fulfill specific biological functionalities, inkjet printing must be able to apply multiple cell types and materials at the same time. In 2013, 3D inkjet printing of heterogeneous cells was implemented with success [155]. Different cell types were transferred into different ink reservoirs of an inkjet printer and then deposited layer by layer at specific locations to form intricate multicellular 3D hybrid structures. Subsequent in vitro and in vivo assays revealed that each cell type remained viable and could perform physiological activities [156].

Although inkjet printing offers unique advantages for clinical use, there are several limitations that hinder the further development of this technology [130]. Currently, only a limited number of materials are suitable for inkjet bioprinting. A general disadvantage of inkjet printing lies in the fact that the ink must be fluid to prevent clogging and that droplets are created in the process. In addition, the requirements and specifications for bioinks are much stricter than those for conventional inks, since it is not only biocompatibility, degradability, mechanical characteristics and commercial advantages but also fluidity and viscosity that must be taken into account when selecting a bioink. Due to the reduced viscosity prior to crosslinking and the auxiliary function post-crosslinking, hydrogel-based substances like alginate and calcium chloride, as well as acrylated polyethylene glycol (PEG), are currently used extensively in 3D inkjet technology. Nevertheless, certain crosslinkers are toxic and are not approved for use in printing of cells. Conversely, the bioink utilized in inkjet printing should have a fairly low concentration to prevent blockages, which can render the printed 3D patterns unsuitable in terms of geometry or performance [157]. The limited viscosity and restricted materials are therefore the key challenges of inkjet printing in the field of biotechnology [158].

The technique of inkjet bioprinting demands a reasonably low viscosity of the printed composite and sufficient mechanical strength to retain its structural characteristics post-printing. The existing photocrosslinkable media offer new possibilities. A cell-loaded bioink comprising acrylated peptides and acrylated PEG was subjected to simultaneous photopolymerization upon release [159]. The cell exhibited excellent viability and blockages were reduced to a minimum because of the reduced viscosity of acrylated PEG [160]. In addition, excellent mechanical characteristics were obtained, and the grafted human mesenchymal stem cells remained in place in the printed pattern and formed homogeneous skeletal bone and cartilage. The strategy of inkjet printing with photocrosslinkable inks employing in-situ crosslinking can therefore enable an increase in the selection of inks for inkjet printing; among them are hyaluronan methacrylate, GelMA, polyethylene glycol diacrylate (PEGDA) and norbornene-functionalized HA (NorHA) [161].

### 2.3. Stereolithography (SLA)

Stereolithography (SLA) is a 3D printing method based on polymerization in a mold. In this process, light-sensitive fluid resin is dispensed into a mold (or container) and selectively polymerized (i.e., hardened and solidified) by exposure to UV light (Figure 3). Most resin 3D printers work that way, but a few “top-down” systems exist as well. In a top-down setup, the light source sits above the resin tank in a top-down setup, therefore curing the surface instead of the bottom. The build plate moves down to leave room for new layers atop the previous ones until the object is complete and appears upright. The UV light hardens the resin layer by layer so that the pieces can be built up in layers to create the final object. In SLA, layer height (layer thickness) is usually about 50 µm but may also be as little as 10 µm. Typically, the thinner the layers, the greater the quality and the increased time is required for printing. Although SLA 3D printing is extremely versatile and precise and creates smooth surfaces, a disadvantage of the method is that the parts can degrade when subjected to sunlight over time due to the photosensitive characteristics of the 3D printing resins.

Once the entire part is finished, it is removed from the tank and placed in a solvent-based chemical bath to remove excess material and create a smooth surface. Finally, the workpiece is hardened in an ultraviolet furnace to ensure that it is properly strengthened.

The functional principle of SLA bioprinting works as follows. SLA represents an initial and original technique of 3D printing that forms the basis of modern 3D bioprinting. In 1984, the first setup has been patented, and four years later the first commercial device has been generated [162]. It is characterized by an extremely versatile choice of materials and features the highest resolution and accuracy, as well as fine details. It is ideally suited for functional prototyping. The SLA process originally contributed to establishing 3D printing as a feasible option for manufacturers and inspired others to explore new printing techniques and new uses for the technology. During the construction process, support structures must be added so that the overhangs have some support. 3D printing with resin also requires additional finishing steps, such as washing out of residual resin, breaking away the support elements and post-curing, which is required to subject the printed element to extra UV light for additional hardening.

What types of SLA 3D printing are there? The acronym “SLA” usually refers to stereolithography, and in its original form, the light source used in resin 3D printing came from lasers reflected by mirrors. Laser printers work very accurately, but they are very costly to purchase and keep in good condition. The stereolithography of the present day also includes other technologies such as digital light processing (DLP) and masked SLA (MSLA). In DLP, a projector is used as the light source in place of a laser. The beam projector contains a digital micromirror device (DMD), where a micromirror constitutes a pixel. The DMD is coupled with a visible LED, laser, or lamp light source for illumination and can also employ UV or IR wavelengths for light-steering purposes. The light source and the micro-mirrors work in synchronization to provide the desired projected output. DMD controls the angle of each mirror and determines the brightness of the light transmitted through the mirror. The DMD can therefore regulate the light intensity within a small part of the projected beam. In DMD, the intensity of the pixels is color-coded. The light beam produced from the DMD travels through a system of lenses and is focalized on the pre-polymer hydrogel suspension. In zones exposed to strong light, the photoinitiator captures enough photons to initiate polymerization, whereas in zones exposed to little or no light at all, no polymerization takes place. Thereby, spatial crosslinking and SLA bioprinting are implemented [163]. A new visible light-crosslinkable bioink that is designed on the basis of cell-adhesive gelatin has been introduced [163]. The bioink comprises a photoinitiator derived from eosin Y (EY) and a GelMA prepolymer suspension, which is suitable for the printing of cells and organoids. While laser SLA printers apply layers of print dot by dot, DLP printers harden each layer at once in a single pulse of light. This process is quite fast compared to conventional laser-based resin 3D printers. Similar to DLP, MSLA hardens all complete layers at once. MSLA printers, however, use a series of LEDs as a light source in place of a projector. The LED lights illuminate through an LCD screen that selectively blocks the light by brightening or turning off certain pixels. The resolution of an MSLA printer is therefore determined by the resolution of its LCD display. SLA-based bioprinting offers benefits in terms of resolution and short printing time, which is why it is currently attracting a lot of attention in the printing of cells and organoids. Traditional SLA bioprinting, however, involves the use of UV light as a photopolymerization principle, which can cause mutagenesis and carcinogenesis of cells [163].

### 2.4. Laser-Induced Forward Transfer (LIFT)

In 1988, a type of laser-assisted printing technology was presented, the laser-induced forward transfer (LIFT) (Figure 4) [164]. In 2004, LIFT was initially used for bioprinting, whereby cell patterns were printed with excellent cell viability [165]. Thus, the emphasis was placed on the LIFT technique, as it is highly useful to design 3D organoids and 3D tumoroids. Unlike inkjet and extrusion printing processes, LIFT technology is characterized by high printing precision and high resolution, which is down to the micrometer range, high output and a high rate of cell survivability [166].

As no nozzles are required during the printing procedure, there is no problem with ink blockage while printing. In addition, this technique can be integrated with different bioprinting methods to broaden printing possibilities and offers the opportunity for in situ printing [167]. As a consequence, LIFT has already been extensively exploited for the bioprinting of pharmaceuticals [168], DNA [169], proteins [170], human osteosarcoma cells [171], human endothelial cells [172,173] and mesenchymal stem cells [174] with pharmaceutical drug administration and screening capabilities, nucleic acid microarrays, protein microarrays and printing of living cells, tissues and entire organs. The high cost of LIFT technology is a major issue restricting its exploration and commercial deployment, but this may change quickly due to high demand. The different variables associated with LIFT bioprinting, comprising laser energy, laser spot size, physical characteristics of the bioink and absorbing layer height, are discussed for efficient and successful bioprinting. The principle of the LIFT process is to focus a light beam passing through a transparent substrate onto a metal or polymer thin sheet, where part of the light is being absorbed and transferred into internal energy [107,175]. This process raises the temperature, stretches and distorts, and can even lead to fluidization or vaporization, resulting in the transfer of material [176,177]. The LIFT system primarily comprises a laser unit, a donor with multiple sheets and a recipient medium. The laser unit is usually a pulsed single-mode laser of a specific wavelength.

The LIFT bioprinting donor layer typically comprises three elements: the substrate, the absorbing coating and the biomaterial sheet [178]. Transparent glass is typically utilized as a substrate for laser wavelengths in the near infrared or visible spectrum, while quartz and fused silica are being employed for ultraviolet wavelengths. In addition, flexible organic carriers are being investigated as possible substrates. The donor substrate, such as transparent quartz glass with virtually no laser light absorption, a metal or metal oxide-coated laser absorbing film and a coating of a biological solvent comprising biological matter like DNA, proteins, or cells. The biomaterial film acts as an ink, which means that it is printed, and the cell ink is made up of cells, cell culture fluid and matrix material. The ink necessitates biochemical characteristics akin to those of the native ECM and usually comprises cell culture fluid [179], fibrinogen [180] or glycerol [181]. The matrix material needs to be very close to the architecture and formulation of the ECM and should have outstanding biocompatibility, moldability, minimal cell damage and be easily decomposable. An absorber layer is usually placed between the transparent substrate and the donor film to avoid direct laser interference with the material to be deposited. Titanium (Ti), titanium dioxide (TiO_2_) or gold (Au) is usually utilized as the metallic absorber coating [165,182,183]. Different kinds of UV-absorbing films, such as polymer coatings [177], have also demonstrated similar printing outcomes. The recipient substrate, generally a coverslip, is placed parallel to the donor and both are mounted on a moving 3D stage to collect printed droplets [176]. The glass substrate is covered with a hydrogel coating or other biocompatible substrate that is key to keeping the biomaterial vital following printing [184]. Primarily, the layer works like a buffer that efficiently minimizes the shear impact damage of the printed biological material on the substrate. Secondly, the hydrogel moisturizes the biomaterial and avoids the volatilization of tiny droplets on the receiving surface. Thirdly, the collagen and laminin present in the hydrogel also ease the attachment of the printed organisms to the surface and assist their ongoing differentiation. For example, the thickness of the layer on the capture substrate has been found to influence the cell activity when printing multipotent embryonic cancer cells [165]. Without any buffer substance, the survival rate of the printed cells lay at 5%. When the coating height has been raised from 20 to 40 μm, the cell survival rate increased from 50% to more than 95%. Nevertheless, the ideal thickness of the layer varies according to the experimental setup, including the viscosity of the bioink, energy of the laser and size of the dot. LIFT is based on the concept of light–matter interference, where a portion of the light is absorbed by a metal or polymer layer of the donor and converted into internal energy, resulting in a temperature rise and blister generation when a laser beam is centered on the layer through a transparent medium [176,183]. The bubbles then expand and deform, leading to their collapse and the formation of a jet or droplet of bioink, which, in turn, enables mass transfer and ultimately the printing of cells or other biological substances within the bioink [177,178]. The receiving surface is a slide coated with buffer. As a heating device in LIFT, pulsed laser systems with pulse widths of a few nanoseconds constitute the majority of laser systems, even though ultrafast laser devices emitting picosecond and femtosecond pulses may also be utilized [177,185]. Optical components like beam splitters and lenses are utilized to manipulate, control and concentrate the laser beam onto the intersection of the donor substrate and the layer of donor material. The laser wavelength must be adjusted to the transparency of the donor substrate and the absorbing ability of the donor layer, even if it does not necessarily affect the process. In addition, the characteristics of the laser system like laser energy density, pulse length, frequency and pulse energy have a considerable influence on the process and the outcome. The selection of the wavelength varies according to the interfacing material (interlayer or material to be deposited), with ultraviolet radiation commonly chosen. With the LIFT bioprinting technique, the laser pulse energy and the beam size are critical variables, whereas a number of other studies have also highlighted the laser fluence, which is proportional to the pulse energy divided by the spot size, as a pivotal factor [186].

To understand the theoretical nature of the material transfer mechanism at LIFT, numerical analysis and simulation techniques have been used to analyze the heat generation, thermal propagation and material transfer characteristics of various materials at a range of laser energies and pulse durations [187,188,189]. Several theoretical models have been articulated, comprising explosive ejection, phase alternation ejection and shock wave ejection. Firstly, the explosive ejection theory assumes that mass transfer is driven as a result of the pressure produced during the laser ablation and vaporization, thereby causing an explosive event [190]. When the melting boundary surface has not yet attained the air boundary layer, the material has eroded and vaporized, and the gas pressure ejects the material in an explosive manner in a very confined area. Secondly, the theory of phase transition ejection can clearly provide an understanding of the ejection of metal microdroplets [191]. Based on this hypothesis, metallic materials stretch under laser irradiation; however, they remain in a solid phase. Simultaneously, the focused laser forces the melted boundary to progressively advance along the metal film in the direction of the air boundary until the film–air border is also liquefied and the fully fluidized film ceases to be trapped at the interface, creating a metal droplet that is discharged and transmitted. Thirdly, the theory of shock waves says that the coating partly melts and volatilizes when heated and simultaneously spreads in the direction of the substrate [192].

Multiple parameters influence a number of important factors including blistering, jet evolution, deposition volume, resolution and cell survival throughout the LIFT bioprinting procedure [176,183], among them firstly, the energy of the laser; secondly, the diameter of the laser spot; thirdly, the physical characteristics of the bioink and fourthly, the absorptive layer height. The generation of the beam during LIFT bioprinting involves three distinct modes that vary with the level of laser pulse energy, such as the sub-threshold mode, the jetting mode and the plume mode [193]. Below the threshold range, the jet is unable to fully develop because of the lack of laser energy or excessive fluid viscosity, which leads to a failure of material transfer. On the contrary, in plume mode, excessive laser energy or insufficient fluid density can induce an unstable jet, which results in the creation of unintended plumes and discontinuous droplets with different volumes. A stable beam that facilitates efficacious and regulated bioink release appears just when the laser energy lies in between the beam and plume threshold values. The size of the laser dot constitutes an additional key factor that influences LIFT bioprinting and defines the printing resolution [166,185]. A narrower laser spot generally results in a better resolution; however, it also carries considerably diminished ink when printing and, hence, leads to less productive printing.

Viscosity is not just an key performance marker for the bioink, but it also has an important part to perform in bioprinting [189,194]. In case the viscosity of the bioink becomes insufficient, splashing can arise while printing. Conversely, when the viscosity is excessive, the laser needs more energy to initiate the inkjet printing process. A suitable viscosity of the ink is essential to ensure a stabilized jet. Like the size of the laser dot, the viscosity of the bioink affects the print resolution considerably, which is affected to a greater extent than the laser energy, particularly in the case of decreased laser energy. The existence of cells in the bioink is critical as it can substantially impact the LIFT bioprinting performance [183]. In comparison to printing using non-cellular bioink, cell entrapment usually needs increased laser energy, leading to lower beam velocities and narrower printing dots [195]. Moreover, the uneven dispersion of cells caused because of the aggregation within the bioink can generate two kinds of non-ideal beam characteristics while printing, such as non-straight beams harboring non-straight trajectories and straight beams harboring non-straight trajectories. As explained above, the depth of the absorber layer and the gap between the donor and receiver layers can affect the success of the printing procedure. Finally, when applying LIFT for laser bioprinting, it is essential to set suitable parameters. Consequently, this may contribute to keeping up the printing pace and prevent biological alterations, like deterioration of phenotypic or nucleic acid cell integrity during LIFT bioprinting. LIFT technology has already been used to print a range of other biomaterials, including lipid vesicles for drug delivery and biosensing applications. Lipid vesicles, however, are molecular partitions made up of lipid bilayers and can be tens of micrometers thick, complicating printing with conventional direct printing methods. As the LIFT technology enables the printing of objects with large dimensions, lipid vesicles have been successfully printed with LIFT without compromising the vesicle membrane [196]. The printing of vascular structures, in contrast, is still in its infancy [173] and requires significant improvements regarding the vessel stability, the vessel modification and its functionality [197].

## 3. Usage of Several Bioinks and Bioink-Database for Cells, Hydrogels and Organoids

The complexity of 3D bioprinting is enormous due to the large amount of different bioinks available. This section describes basically the major issues of bioinks in general from a biophysical cell perspective. There are detailed review articles available for more background and refined information [198,199]. It is therefore natural that the overview and usability of bioinks has been summarized in a first bioink database for 3D extrusion printing [200]. The database enables the easy identification of combinations of extrusion pressure, temperature and speed that have been optimized for the printing of specific biomaterials and, even more importantly, to highlight the areas in which printing cannot be accomplished. The database allows scientists and prospective bioprinting users to quickly find the right bionics for the respective application and helps with the execution of the printing by utilizing decisive parameters that must be considered in each case. This database is constantly being expanded through the voluntary input of new bioinks and their printing parameters. The collected results also permitted a correlation analysis among all printing variables, such as needle size and type, that showed suitability for cell-based 3D printing. Although bioprinting is still in its infancy, the important issues of standardization and evaluation of factors, such as the shape accuracy of the printed structures, repeatability, material characteristics and the used hardware and software, have been purposely addressed from a regulatory and clinical viewpoint [201,202,203]. However, that does not account for the vast variation in printing regimes, which are specific to every printing mode and 3D printer utilized and reported by researchers worldwide [204]. As a result, there is a wealth of data that is both useful and sometimes confusing and contradictory in the 3D bioprinting field. Therefore, the establishment of the world’s first bioink repository for 3D bioprinting is a logical consequence to make it easier to navigate and keep track of things. The database is open source and enables researchers to simply access it and add the results of their work to the database repository. This database concentrates exclusively on microextrusion printing and captures critical printing variables like the composition of the bioink, pressure, temperature, velocity, needle type and the cell type employed. The database is freely accessible at https://cect.umd.edu/3d-printing-database (accessed on 20 August 2024). At present, there are more than 200 various bioink compounds listed that have been utilized for 3D printing. These materials comprise thermoplastics like PCL, PLA, PLGA, natural and synthetic polymers like alginate, collagen, decellularized ECM (dECMs), and PEGs, ceramics comprising hydroxyapatite and β-tri-calcium phosphate, various nanomaterials, nanocomposites, biomolecules and proteins, which have been utilized as additional ingredients in the extruded bioinks. The following are the physiochemical properties of the selected polymeric hydrogels that serve as bioinks are briefly presented. As the hydrogels serve as a scaffold for the printed organoids, apart from biochemical and structural characteristics, the mechanical characteristics seem to be important for organoid survival, growth and further development. In the following, the different types of hydrogels employed for bioprinting are presented. Beyond the mechanical properties of hydrogel scaffolds, there are degradable and non-degradable scaffolds, which may be relevant in processes, where temporal stability is required, but need to be altered over time to mimic physiological conditions.

### 3.1. Physiochemical Characteristics of Polymeric Hydrogels Employed for Cells and Organoids

When using inks that are compatible with living organisms (referred to as bioinks), non-toxicity, degradability, cell adhesion and porosity must be guaranteed [62,198,205,206,207,208]. Inks in which living cells are encapsulated are in a state of conflict, as the properties that constitute a stable printing, such as density or viscosity, are frequently in direct opposition to the maintenance of viability, as cells require a porous and compliant surrounding in order to grow and migrate [209]. The rheological demands on the bioink change depending on the bioprinting method, such as inkjet or droplet-based, laser-based or extrusion-based printing [201,207,209,210]. In inkjet bioprinting, a continuous stream of small droplets is used to generate the 3D structure. This process is, nevertheless, generally time-consuming and inefficient for the production of tissue on a large (clinical) scale [206]. Laser-based bioprinting utilizes a precise laser beam to harden the engineered structure in a pool of bioink, but heat can harm the cells [210] and the process is quite slow [201]. Extrusion-based bioprinting comprises the shape of a low-viscosity filament or thread during printing, which hardens on the print surface, retains its shape and encourages the layering process [206]. A major obstacle with this technique is that the cells are subjected to a perceptible shear stress from the applied extrusion pressure as they travel through the syringe and nozzle, potentially leading to cell injury [201,208,211]. To mitigate this stress, the bioink must exhibit lower viscous properties [212], but this can lead to distortion, collapse and occlusion of pores, which, in turn, reduces accuracy and resolution [209]. Hydrogel-based bioink compositions are a versatile choice for a wide range of techniques to accomplish bioprinting. Polymeric hydrogels consist of 3D interconnected scaffolds of hydrophilic polymer chains that can absorb significant quantities of water and expand up to 99% of their dry weight in water (*w*/*w*) without disintegration [213,214]. Although there are different kinds of hydrogels and different gelling techniques, the emphasis, in this review, is placed mainly on polymeric hydrogels. Polymeric hydrogels offer excellent biocompatibility and tissue-like mechanical characteristics, rendering them ideal for 3D bioprinting and a range of tissue engineering uses [49]. Hydrogels mimic the ECM, the natural environment of cells, in an effective manner and offer a hydrated and texturally supportive surrounding that can be populated with cells homogeneously and in an efficient way [214]. Cells can be dispersed in these polymeric hydrogels to produce a bioink and in a controlled manner in bioprinting applications. Cells can be embedded into a hydrogel emulsion, and this bioink has been extruded to form cell-laden vascular patterns [49]. Hydrogels frequently consist of shear-thinning materials that can be forced into extrusion under high shear stress and subsequently retain their mechanical characteristics [214,215]. Therefore, they are ideal for bioprinting purposes. Substances like gelatin, PEG and Pluronic^®^ behave like fluids during the printing process and return to a gel-like consistency after the extrusion, ensuring that the printing process has the required stability to create the intended texture. The following subsections outline the characteristics of natural and synthetic polymer hydrogels typically encountered in bioprinting processes. Knowledge of the physicochemical characteristics of hydrogels is essential for evaluating the stability, performance and toxicity of hydrogel uses in bioprinting. The most relevant physicochemical characteristics are the pH value, the printing temperature, and the degree and type of crosslinking.

#### 3.1.1. pH

The majority of hydrogels can be stored and processed via bioprinting under physiological pH conditions, such as around 7.4 [216,217]. The pH value of hydrogels significantly affects the swelling properties of hydrogels [217]. The swelling capacity determines the form and volumetric variations of a hydrogel; thus, a higher swelling ability is favored because of the enhanced robustness of the hydrogel [218]. The highest swelling potential of most hydrogels is at a physiological pH value of approximately 7.4 compared to an acidic or basic pH value [217]. A shift in pH generally leads to a modification of the polymer chain charge, resulting in either swelling or non-swelling of the hydrogel and a general modification of stability [219]. In particular, pH-sensitive hydrogels are prone to pH variations, mainly due to their ionic character [219]. At low pH, cationic hydrogels have a natural propensity to swell because of the protonation of amino/imine chains, while anionic hydrogels tend to swell at higher pH values because of the ionization of acid chains [219]. Measuring the swelling ratio can also yield insights into the type, amount and tightness of crosslinking in the polymer matrix and can be utilized to indirectly assess the mechanical characteristics of the gel, like the modulus of elasticity (E) (stiffness) [220,221]. A higher pH value of collagen during the gelation process leads to higher stiffness [222], which can reduce the printability and the vitality of cells during bioprinting. When the pH value was raised in the region of 5 to 8, for example, the relaxation modulus of collagen gels increased linearly (in other words, the gels stiffened) and then stagnated. At the same time, the viscosity of the hydrogel changes, such as a low viscosity at higher pH of 8.5 [223], which also affects the shear thinning behavior that is critical for bioprinting. In addition, the pH value affects the gelling time, which impacts the overall cell survival and proliferation.

#### 3.1.2. Temperature

The temperature is inversely related to the viscosity of the hydrogel [224]. The higher the ambient temperature, the lower the viscosity, which is associated with reduced shear stress and minor deterioration of the cells [224]. For bioprinting, a low viscosity of the bioink is required to achieve an optimal printing result in terms of cell viability, but there is often the problem of suboptimal print accuracy and image resolution [225]. However, in bioprinting, the printing temperature required varies according to the type of polymer utilized. In reactive ionotropic polymer printing, the polymer liquid can be kept and printed at cell culture temperature, such as 37 °C, for the production of hydrogels [226]. Since gelation in the reactive printing of ionotropic polymers is initiated in a tank with suitable counterions, gelation is very fast and it is possible to print polymer solutions together with cell culture media [226]. There is a lag phase during collagen gelation in which the primary aggregates of collagen molecules are established (nucleation event). Next, microfibrillar aggregation begins with the lateral aggregation of subunits triggered by alterations in ionic strength and pH and increases the temperature up to 37 °C until reaching equilibrium. In opposition, the fundamental mechanism of gelatin is associated with the reverse coil-to-helix transition induced when solutions are cooled below 30 °C, with the resulting helices resembling the collagen triple helix but not achieving equilibrium. The gelling processes are thermoreversible for both collagen and gelatin, albeit in opposite directions: collagen gels dissolve when the temperature is decreased, whereas gelatin gels dissolve when the temperature is elevated [227].

For hydrogels made up of polymers that react through physical interactions, the optimal temperature depends on the type of polymer to be gelled. Hydrogel materials, like gelatine methacryloyl, often experience a physical sol-gel or gel-sol transition from room to body temperature and can also be chemically crosslinked at these temperatures to achieve dimensional stability [228]. Normally, heated polymer mixtures are printed in a chilled surrounding, in which they attain their gel transition temperature and solidify [226]. Agarose, for instance, is soluble in water at above 65 °C and melts into gel at 85 °C [229]. For this reason, agarose is usually placed in the printer tank at temperatures ranging from 60 to 80 °C [226]. The agarose is subsequently printed in a chilled liquid bath, typically between 17 and 40 °C, which is lower than the gel transition temperature [199]. For polymers gelling by physical contact, the final gel temperature frequently impedes cell embedment and shortly following printing, as the temperature is outside the normal body temperature regime and is potentially detrimental to living cells. Moreover, in dependence to temperature the some materials exhibit a shape memory effect [230] indicating that the temperature is highly relevant.

#### 3.1.3. Crosslinking

The majority of bioinks, such as hydrogels, are stabilized by crosslinking mechanisms to maintain the shape and mechanical strength of the 3D-printed structure. Post-print crosslinking is a procedure in which the interior architecture of the printed hydrogel is altered to preserve the overall structural integrity and obtain the mechanical characteristics of the bioprinted structure [224]. The two commonly occurring crosslinking mechanisms are of physical and chemical nature [109]. Physical crosslinking is achieved through physical processes, and comprises intermolecular interferences between polymer chains like hydrophobic interference, electrostatic interference, hydrogen bonds, stereocomplexes and guest–host interference [109]. The physical crosslinking phenomenon can be reversed, and few or no chemical responses are required to create this connection. In chemical crosslinking, reactants are applied to effect the covalent bonding of chemically responsive functional chains [109]. Chemical crosslinking is generally mechanically more powerful than physical crosslinking, as it produces covalent bonds between the polymer chains, but, whichever agent is applied, it carries the potential risk of causing cytotoxicity. The chemical crosslinking of hydrogels has an irreversible effect; however, its benefits are durability, adjustable structures and excellent mechanical characteristics [109]. In enzymatic cross-linking, covalent bonds are also established among the polymer chains, but the extent of crosslinking is somewhat less strictly regulated [198]. Managing the level of crosslinking is critical in bio-implementations as it allows the stiffness of the structure to be altered and adapted to the respective tissue [231]. A less common cross-linking phenomenon is thermal cross-linking, a mechanism that requires temperature fluctuations in the environment. The majority of natural polymer hydrogels crosslink at 37 °C [109]. Nevertheless, only a few hydrogels, such as alginate and gelatin, crosslink at room temperature [109]. When a gelatin suspension begins to cool, the protein polymers coil into twisted, helical configurations, causing the mixture to harden. Other natural and synthetic hydrogel substances display a comparable temperature sensitivity, including gellan gum (that can also be ionically crosslinked), agarose, polymers of N-isopropylacrylamide (NiPAAM) and Pluronic F127 (synonymously referred to as Poloaxomer 407). The sol-gel transition in the product Pluronic F127 takes place upon heating, which means that the polymer solution forms a liquid state at low temperatures and becomes a thermally crosslinked hydrogel when heated [232]. The viscosity and stability of Pluronic F127 can be improved by adding chitosan [232].

### 3.2. Biological Characteristics of Hydrogels Serving as Bioinks for Cells and Organoids

Hydrogels have the utility for bioprinting as they exhibit numerous characteristics akin to those natural extracellular matrix of tissues, facilitating the embodiment of cells in a highly hydrated form and mechanically maintain the 3D environment [134]. The bionic scaffold should have adequate mechanical strength, biocompatibility, cell proliferation, survival and other biological properties. The drawbacks of hydrogels made from natural polymers are poor mechanical characteristics and low printing performance and dimensional stability. In recent years, a number of synthetic, modified and nanocomposite hydrogels have been designed that can modify their characteristics by physical interactions, chemical covalent bond crosslinking and bioconjugation reactions to meet the specifications [233]. The hydrophilicity of hydrogels constitutes the primordial driver of biocompatibility, rendering them a beneficial channel for the production of tissue structures [234]. Hydrogels offer a proper microenvironment for cell proliferation and are highly adaptable. They enable a range of biochemical and biophysical features to regulate cell behaviors, among them cell adhesion, migration, proliferation and differentiation [235]. A variety of cell types are capable of survival when embedded in hydrogels, as these scaffolds can provide cell–matrix interactions to determine cell fate [236]. Among these cell types are fibroblasts, chondrocytes, macrophages, hepatocytes, endothelial cells, smooth muscle cells, adipocytes and stem cells [134,236]. There is a complicated communication between hydrogels and cells, including stem cells; multiple parameters like porosity, different types of polymers, stiffness, tunable heterogeneous structure via magnetic beads, compatibility and decomposition cause the survival or death of stem cells [237,238,239,240]. Hydrogels imitate the 3D ECM and create a favorable environment for cells. Cells, including cancer cells and stem cells, can perceive their environment via mechanosensing through various elements, such as cell surface receptors, such as α5β1, αvβ3, α1β1 or α2β1 integrins, caveoli, ion channels, such as Piezo 1 and Piezo 2 and extracellular vesicles, including exosomes, in order to further develop, expand, multiply (proliferate) or simply survive [241,242,243,244,245,246]. Hydrogels can be generally made of pure natural or pure synthetic polymers or mixtures of both [199]. Both natural and synthetic substances with various characteristics and performances are utilized in the manufacture of hydrogels [224]. Synthetic hydrogels have increasingly been employed more frequently than natural polymers in recent times due to their increased water absorbency, extended durability and the variety of available chemical ingredients [199]. The natural and synthetic hydrogels are presented below, and some examples of each type are explained in more depth. Moreover, hydrogel compositions and their adaptations are characterized. Modifications enable changes in the chemical performance and mechanical characteristics of hydrogels [247]. For synthetic hydrogels, changes are critical for enhancing biocompatibility and cellular attachment characteristics, whereas for natural hydrogels, changes enhance shaping capabilities. Chemical changes to hydrogels could contribute to the creation of robust hydrogel structures enhancing characteristics like dynamic coupling, shear thinning and self-healing and promoting covalent as well as ionic crosslinking [199].

#### 3.2.1. Natural Polymer Hydrogels Serving as Bioinks for Cells and Organoids

Natural polymers originate from natural material sources. Typically employed natural polymers comprise cellulose, collagen, guar gum, gelatin, chitosan, alginate and fibrin [199,248]. Hydrogels made from natural polymers exhibit superior biological characteristics compared to their synthetic equivalents, as they comprise improved biocompatibility, biodegradability and procellularity [224]. The rationale for this is that natural polymers can coat the surface of eukaryotic cells and then bind with proteins to form a natural ECM [199]. For instance, glycosaminoglycans (GAGs), unbranched high molecular weight polysaccharides that are either covalently bound to protein cores and constitute proteoglycans or occur freely in the ECM, can coat the surface of cells and couple with several proteins to produce a natural ECM, leading to outstanding biocompatibility and cell affinity [249]. The integration of GAGs in biomaterials offers novel pathways for the display of signaling molecules and facilitates the monitoring of development, homeostasis, inflammation and the development and propagation of tumors. GAGs provide the structural foundation for several important functional characteristics of the ECM. Besides the hydrogel characteristics of tissues, such as compressive resistance [250], GAGs convey the local display of multiple soluble signaling molecules [251], involving the generation of morphogen gradients [252]. For example, it has been demonstrated that HS-GAGs regulate the generation of morphogen gradients in vivo [253], and thereby control the adaptive development of tissues and organs within multicellular organisms [254]. Thus, GAGs have been demonstrated to be associated with key events, including in development, and tumor evolution and malignant progression [255,256]. Ultimately, GAG-containing hydrogels can be rendered vulnerable to breakdown by cell-secreted proteases by integrating matrix metalloprotease (MMP)-cleavable peptide crosslinkers [250]. In addition, the majority of natural polymers possess bioactive constituents involved in amplifying extracellular signal transduction to enhance cell proliferation, cell differentiation and cell functionality [199]. These components comprise protein ligands and motifs that attach to cells; thereby, forces can be transduced from the ambient environment toward the cell’s interior [257]. Many of the following natural biomaterials can be employed for 3D organoid culture. Thus, they are briefly introduced in the following.


**Agarose**


Agarose as a natural polysaccharide is derived from marine algae. It is not as commonly employed for bioprinting purposes as some other natural hydrogels because it is challenging to print and, as it is extracted from a plant, it lacks biomimicry for mammalian cell types [258]. Nevertheless, its beneficial gelling characteristics render agarose an attractive hydrogel constituent and supporting framework. In nozzle-based bioprinting, agarose made its first appearance in 2005, when Chinese hamster ovary (CHO) cells were printed in agarose scaffolds [152]. In recent times, an agarose in a compound hydrogel, consisting of gelatin and alginate, in which adipose-derived stem cells (ASC) are suspended, was introduced [259]. Highly precise and robust bioprint textures were printed. It has also been observed that the addition of agarose enhanced the pore size and quantity in the hydrogel, favoring cell proliferation [259]. Other studies show that agarose can be used effectively in a very indirect way. In 2018, Mirdamadi et al. reported a technique of embedded bioprinting that built on the seminal research of Hinton et al. in 2015 [260], in which a cell ink was expressed in an agarose suspension [261]. The agarose suspension offered temperature-resistant textural enhancement to the soft bioprinted structures throughout and beyond printing, so that the printed construct could stay in the suspension even when transferred to the growth incubator. Moreover, the agarose gel was penetrable for constituents of the cell medium, which resulted in media replacement through diffusion in the vicinity of the printed structure with no disturbance of the texture [261]. In 2016, the application of agarose in conjunction with collagen in a nozzle-based bioprinter was published [262]. When agarose was added to collagen, a tissue-like matrix was created. The advantage was that the addition of agarose to the cell ink did not alter the structural topography of the collagen mesh and the collagen solution had no effect on the agarose gelling. The incorporation of agarose into the cell ink resulted in a more viscous ink, a reduced droplet size and increased printing precision [262]. In 2022, efforts in extrusion-based bioprinting characterized the agarose-gelatin hydrogel mixtures characterizing the mechanical and rheological characteristics for bioprinting [263]. Moreover, the human SH-SYn5Y neuroblastoma cells from the neural crest [264] were printed using the above-mentioned agarose-gelatin mixture as cell ink and differentiated into neuron-like cells [263].


**Collagen**


Collagen type I is among the top prevalent fibrous proteins in the ECM and is the principal structural component of the ECM that offers tensile strength, controls cell adhesion and aids in cell proliferation [265,266]. These properties render collagen to be an ideal hydrogel for the application in cell inks for bioprinting, as multiple tissue cells can generally adhere to it [267]. The principal types of type 1 collagen are pig skin, rat tail tendon and cow skin [268]. However, all these types of collagen can exhibit diverse mechanical and structural cues upon scaffold formation [8]. The usage of collagen in cell inks, though, is restricted because of its long gelling time, lasting up to 30 min at 37 °C [266]. In addition, this long gel time can lead to an inhomogeneous arrangement of the cells and, consequently, result in a loss of structural accuracy in the final printed object [266]. Moreover, collagen is fluid at low temperatures and becomes fibrous at higher temperatures or at neutral pH, which can be problematic when printing with nozzles, because the nozzle mechanism is occasionally produced with heating [198]. In 2019, Lee et al. described the application of freeform reversible embedding of suspended hydrogels (FRESH) for the biological engineering of human heart parts at different levels of complexity [269]. In FRESH, a collagen-based cell ink is extracted into a thermoreversible carrier pool consisting of a suspension of gelatine microparticles, which acts as a carrier while the print is being made and is then discarded. A left ventricle has been produced from human stem cells with the help of the FRESH technique. In a two-material printing procedure, the collagen ink and a cell ink with a high cell count density are deposited to generate the ventricle. In the complete ventricle, it was possible to monitor synchronized contractions, the directional propagation of the action potential and the wall-thickening characteristic of a ventricle [269]. An aerosol jet bioprinting process has been developed for printing high-density collagenous textures [270]. Aerosol jet bioprinting involves a printing process in which an aerosol is generated from an ink and a vehicle gas that impregnates and coalesces on a surface [270,271]. This technique may be an attractive way to print collagen into high-density structures to use as a cell substrate [270], although several reports suggest that high-density collagen structures can inhibit cell proliferation and impede the capacity for differentiation and diffusion of by-products [55,272]. Inversely, fibroblasts can be cultured in high-density collagen gels (40 mg/mL) with a high viability rate after culturing for a week [267], which highlights the opportunity to use aerosol jet bioprinting as a novel tool to generate substrates for bioprinting.


**Fibrin**


Fibrin refers to a fibrillar protein formed from fibrinogen that circulates in the blood and is frequently derived from the plasma of mammals. A fibrin clot is the body’s first response to a laceration as it builds a matrix of fibers to stop the bleeding. Fibrin, as utilized in tissue engineering applications, is manufactured exactly as the body makes it by activating fibrinogen monomers to form a polymeric fibrin matrix [258]. Fibrin is degraded in the human body by fibrinolysis that is carried out predominantly by plasmin. In vitro, cells generate enzymes that catabolize fibrin [273]. Therefore, fibrin hydrogels suffer from a major deficiency in structural robustness for use in direct cell contact situations. Moreover, fibrin is a difficult choice for nozzle-based bioprinting due to its high viscosity, and fibrinogen offers poor texture and form accuracy [273]. Therefore, fibrin may be a difficult substrate in cell inks. A number of different approaches can be employed to overcome these restrictions and enable the effective incorporation of fibrin into cell inks for bioprinting. A method is the utilization of fibrinogen, that has a viscosity close to water. Once the fibrinogen has been deposited, the crosslinking reagent thrombin can be given on to the fibrinogen or as a substrate to generate a definitive fibrin network through the crosslinking of the fibrinogen through a calcium-dependent route [274,275]. This technique can be utilized with nozzle-based bioprinters that ink-print human microvascular endothelial cells (HMVECs) with fibrin to produce a microvasculature [274]. The use of an extrusion-based technique to print a fibrinogen-based cell ink in a thrombin-enriched PEGDMA alginate pool has been reported to hyperlink the fibrin [273]. With this technique of bioprinting, a soft microenvironment that mimics the soft pericellular matrix of cartilage has been obtained. This enables improved nutrient delivery in a bioprinted cartilage scaffold and thus the production of cartilage that is closely resembling that of natural cartilage. A nozzle-based technique has been adopted to bioprint a three-layer vessel wall for a vascular model [275]. Following printing of a surrogate gelatin core loaded with human umbilical vein endothelial cells (HUVECs), a fibrin-based ink has been bioprinted onto the gelatin core (lumen). The gelatin has subsequently been lysed, and the retained ECs have been permitted to settle and adhere alongside the lumen of the fibrin-based vascular graft [275]. In this manner, the adhesive-like characteristics of fibrin, along with the strength and optimal surrounding for cells that it affords, could be harnessed. Although more advanced printing techniques are necessary than with traditional hydrogels, the advantages of fibrin encompass its biological decomposability, its adhesive characteristics, its adjustable mechanical and nanofibrous textural characteristics [273].


**Gelatin**


Gelatin is another common component of hydrogels. Gelatine is often chemically modified or mixed with another polymer before being processed into a hydrogel because of gelatine’s inferior rheological characteristics [224]. In a number of trials, gelatin has been altered with furfuryl chains to produce furfuryl gelatin (f-gelatin) [224]. f-gelatin can be quickly crosslinked in the physical presence of visible light and retains its textural fidelity following crosslinking [276]. f-gelatin can also be amended with hyaluronic acid to provide improved viscosity and shear thinning and to enhance the textural integrity and rigidity of the reticulated structure [276]. Amending gelatin with free radical crosslinkable methacrylic groups, which results in gelatin-methacryloyl (GM or GelMA), is a new technique to increase the stability of gelatin and enable its utilization in cell bioinks for bioprinting and other tissue engineering applications [277]. The GelMa can be stabilized by fluorenylmethoxycarbonyl diphenylalanine (Fmoc-FF) crosslinking peptide into the gel bioink to overcome the post-printing processing of the bioink [278,279]. The effects of the cooling and heating rates on sol-gel and gel-sol transitions in GelMA can be analyzed with rheological techniques [228]. Crosslinking chemically modified gelatin at low temperatures can lead to a higher modulus (stability) than the crosslinking carried out at high temperatures [228]. The characteristics of the final hydrogel are thus highly sensitive to the temperature of processing and can be adapted to the required use. By chemically modifying a gelatin-based hydrogel with glycidyl methacrylate, a protein-based elastic hydrogel (GELGYM) has been produced that can be specifically engineered for ocular tissue engineering purposes; however, it can also be employed for various other tissue types [280]. An engineered blood vessel could be developed to withstand a pressure of up to 350 mmHg, which therefore qualifies GELGYM as an attractive choice for a cell ink for vascular bioprinting [280]. A mixture of methacryl-modified gelatin (GM), non-modified gelatin and acetylated GM could be utilized to create vascularized osseous constructs [277,281].


**Alginate**


Alginate comes from brown algae and is a natural polysaccharide copolymer that is among the natural polymers most frequently utilized for bioprinting [224,282]. As a bioink for cells, it has multiple benefits as it is non-immunogenic, biodegradable, non-cytotoxic, inexpensive and rapidly gellable [283]. The drawbacks are low cell adherence and the insufficient promotion of cell proliferation [284]. In addition, alginate is difficult to print on, and although it is a biodegradable material, alginate degradation can involve complicated mechanisms. Alginate is hydrophilic and can therefore be blended readily with a series of natural and synthetic polymeric cell inks, such as collagen [285], silk fibroin [286], and decellularized and solubilized ECM (dECM) [287], to create a more favorable environment for cells compared to alginate on its own. The combination of these materials enables a perfect balancing of biological and physical characteristics, with alginate frequently acting as a textural stabilizer and as a thickening material. A chemical amendment to optimize the characteristics of alginate is the oxidation of alginate. Oxidized alginate (ox-alg) exhibits a quicker breakdown capacity and contains a higher number of reactive moieties, thereby improving alginate’s suitability for sustaining cell performance [224]. Another conventional alginate modified form is methacrylated alginate (MeAlg/AlgMA) [224]. Methacrylated alginate offers the capability of photocrosslinking, which opens more design possibilities for adapting the mechanical characteristics of the hydrogel, the pore size scale and the decomposition velocity [288]. In addition to amending the hydrogels on their own, new techniques are also being tried out to achieve better printing performance. A combination of PEG and alginate leads to very long-lasting and extensible hydrogels [289]. The printed structures are made particularly durable by the inclusion of nanoclay. Microstructured alginate hydrogels have been produced by a microreactive inkjet printing method in which a precursor and a crosslinking agent encounter each other in air while printing [290]. This novel technique offers a unique option for jet-based bioprinting and demonstrates favorable characteristics of the bioprinted alginate [291]. Typically, alginate bioprinting is performed by one of two techniques: the alginate is pressed into a bath of crosslinker, such as typically calcium, or the crosslinker is printed onto the precipitated alginate [290]. This technique enables a freestanding vessel system with a small circumference to be printed.


**Hyaluronic acid (HA)**


Hyaluronic acid (HA) is a straight polysaccharide that occurs naturally in the ECM of both cartilage and joint synovial fluid [292]. HA acts to preserve the synovial fluid by enhancing its viscosity and increasing the flexibility of the cartilage. Thus, HA is extremely biocompatible and promotes cell signal transmission, the healing of injuries and the organization of the matrix [293]. In addition, HA has been found to possess anti-inflammatory properties, rendering it an attractive material for the implantation of bioprinted textures [294,295]. HA is negatively charged, which causes the attraction of cations and osmosis to absorb water, forming a gel [258]. Nevertheless, HA is easily soluble at room temperature, which restricts its textural integrity and stability. HA has the potential to be chemically altered with a range of functional chains to reduce breakdown and improve durability [293]. A thiol-modified hyaluronic acid and thiol-modified collagen hydrogel has been reported to be suitable for printing using a nozzle-based (jet-based) bioprinter [221,296]. A drawback is that substantial dilution and chilling is required to jet this hydrogel substrate correctly while avoiding obstruction of the printer nozzles. Although diluted, this hydrogel readily underwent cross-linking at room temperature and offered a sustaining medium for downstream cell ejection. Alginate-hyaluronic acid hydrogels can be networked by multiple mechanisms, which include acyl-hydrazone, hydrazide interactions and calcium ions [297]. It has been possible to prepare an alginate acyl hydrazide:HA monoaldehyde gel with a ratio of 50:50 (A5H5), with a gelation time of approximately 60 s, a viscosity of approximately 400 Pa at a zero shear rate, high resistance to different pH solutions and a prolonged breakdown time of over 50 days [297]. Moreover, intricate patterns like small, empty cylinders could be printed with no difficulty. In 2019, the bioprinting of skeletal grid structures with a cell ink comprising HA, hydroxyethyl acrylate (HEA) and gelatin methacryloyl has been presented [298]. Moreover, stable rheological characteristics and outstanding biocompatibility have been found [299].


**Matrigel^TM^**


Matrigel^TM^ stands for the trade name for the basement membrane matrix obtained from the Engelbreth–Holm–Swarm (EHS) mouse tumor (sarcoma). The Matrigel^TM^ is a mixture of proteins and small molecules, mainly collagen IV, perlecan, laminin and growth factors, and closely replicates the extracellular environment of many types of tissues [258]. Matrigel is usually stored at 4 °C (liquid), and it undergoes polymerization at the body temperature of 37 °C [258]. This property has made this hydrogel an outstanding choice for bioprinting purposes. It is frequently employed in cell cultures as it potently stimulates cell proliferation and cell differentiation. Cells grown on a Matrigel^TM^ display show a complicated cellular response that is usually hard to stimulate in a laboratory [300]. Bioprinting has been performed with pure Matrigel^TM^ suspensions containing human skeletal muscle progenitor cells [301], using a chilled print head to suppress gelling of the hydrogel during printing and only allowing it to gel when deposited on the printing surface at room temperature. After culturing the printed structures, skeletal muscle tissue emerged that contained contractile, cross-striated myofibers that contracted in response to electrical impulse activation. This type of bioprinted microphysiological system (MPS) is beneficial for drug discovery, for instance, when testing drug candidates to treat muscle atrophy [258]. A customized extrusion bioprinter has been utilized for the bioprinting of mouse prostate cancer cells floating in Matrigel^TM^ [302]. A volumetric dosing system has been implemented to ensure that the irregular “splashing” extrusion that can arise when printing plain Matrigel^TM^ is minimalized. Although Matrigel^TM^ has favorable characteristics in terms of cell proliferation, it needs some adjustments concerning its printability [303]. As an alternative for Matrigel^TM,^ a biosafe dECM can be employed [303]. A major weakness of Matrigel^TM^ is the batch to batch variability [304].

#### 3.2.2. Synthetic Polymer Hydrogels

Synthetic polymers are generally grouped into plastics, elastomers, and synthetic fibers [305]. For tissue engineering, it is ideal to imitate the ECM to produce an optimal tissue equivalent. Although synthetic hydrogels offer the benefit of photopolymerizability and a high degree of adaptable mechanical characteristics, they cannot mimic the ECM because of their bioinert nature [306]. Synthetic hydrogels are more water-absorbent compared to natural hydrogels. The proportion of water in the hydrogel is dictated on the basis of the characteristics of the polymer and the crosslinking density [307]. The simulation of the ECM is essential because the ECM is not only a structural scaffold, but also regulates cellular functions such as cell migration, cell proliferation and cell differentiation [308]. Mimetic modification of the ECM in synthetic hydrogels has been shown to be an effective means of eliciting the intended cellular reactions. Synthetic polymers create artificial environments [224], whereby plastics, elastomers and synthetic fibers are the most frequently used raw materials for the creation of synthetic hydrogels. Synthetic hydrogels can be easily manufactured and chemically modified for specific applications [309]. Hydrogels made of natural polymers were initially increasingly debated due to their favorable biological characteristics and later rejected in favor of natural polymer hydrogels [224]. A potential explanation for the latest rise in interest in synthetic polymer hydrogels is the simplicity of their industrial manufacture and their ability to be highly modified, allowing multiple geometries for the construction of tissues [224].


**Poly(ethylene Glycol) (PEG)**


PEG consists of ethylene oxide monomers in its simplest version. PEG is an extremely diverse synthetic substance that is popular in the biomedical field due to its ease of customization [258]. Different degrees of polymerization and varying molecular weights can considerably modify the mechanical characteristics of PEG. The polymer can also have different names depending on its molecular weight, such as PEG with a Mw less than 20 kDa, poly(ethylene oxide) (PEO) a Mw over 20 kDa or poly(oxyethylene) for any Mw [258]. As a non-viscous preliminary solution, PEG is an appealing starting material from which to produce cell inks, because it can be especially adapted for tissue engineering purposes. Photopolymerization is the most common technique to produce PEG hydrogels, in which light is employed to transfer liquid PEG macromer mixtures into solid hydrogels [306]. PEG acrylates, such as PEG diacrylate (PEGDA), PEG dimethacrylate (PEGDMA) and multi-armed PEG (n-PEG) acrylate (n-PEG-Acr) are commonly used for photopolymerization [306]. The utilization of tetrahedral PEG tetracrylates (TetraPACs) could be applied in an extrusion-based bioprinting procedure [310]. Thiolated hyaluronic acid linked with TetraPAC, a PEG derivative and agarose microfilaments have been utilized to create hollow vascular conduits by bioprinting. Fibroblast cells from mice (NIH 3T3) have been embedded in this hydrogel blend and exhibited high viability [310]. Peptide-conjugated PEG has been applied to print human mesenchymal stem cells (hMSCs), where the resulting prints possessed outstanding biocompatibility and the nozzle-based bioprinter hardly occluded [159]. The inclusion of peptides in PEG has been proven to enhance cell adhesion and promote several immunomodulatory actions [311]. A cell ink composed of PEGDA hydrogel and human chondrocytes has been employed for cartilage repair in a nozzle-based bioprinting technique that facilitates concurrent photopolymerization and printing [154]. This work takes advantage of the proven ability of PEG hydrogel to be biocompatible, to be broken down by the body and not to alter the phenotype of chondrocytes [154]. Most crucially, the PEG hydrogel’s compressive modulus can be adjusted to resemble that of human cartilage [312]. The tunability of hydrogels, especially biodegradable PEG-based synthetic hydrogels, has been investigated [313]. For example, a polycaprolactone-poly(ethylene glycol)-polycaprolactone mixture (PCL-PEG-PCL) has been utilized to build a hydrogel with high elasticity and flexibility to facilitate bioprinting with a visible light deposition curing mechanism to 3D print mouse fibroblasts (3T3) utilizing an extrusion-based printer [313]. Since a low degradation rate of PEG in vivo has been reported [314,315], the modification of PEG for bioprinting is a highly relevant issue that has generated encouraging findings.


**Pluronic^®^**


Poloxamers, most frequently referred to by the trade names Pluronic^®^ and Lutrol^®^, belong to the category of amphiphilic triblock copolymers, which means polymers with hydrophilic and hydrophobic regions. Pluronic is heat-sensitive, and its sol-gel transition temperature range is wide, spanning from 10 °C to 40 °C [316]. Therefore, Pluronic is generally stable at room temperature and at human body temperature [316]. Because Pluronic is a synthetic hydrogel, it has a lot of the biological drawbacks of PEG hydrogels, such as weak cell adhesion and the impossibility of enzymatic breakdown. Nevertheless, a key benefit of Pluronic is that it has excellent form retention and is thus precise. It provides structural reinforcement, which means it is also a suitable substitution material. It tends to become soluble in liquids, so it is frequently inappropriate for prolonged physical interaction with cells. The nanostructuring of Pluronic is an attempt to preserve its structural characteristics but also to facilitate a long-term cell culture following bioprinting [317]. A mixture of Pluronic dimethacrylate and non-modified Pluronic has been taken to create stable gels through UV crosslinking. The non-modified Pluronic is subsequently eliminated from the cross-linked meshwork so that the quantity of Pluronic interfacing with the cells can be decreased to improve viability. Methacrylated hyaluronic acid (HAMA) has been incorporated to replace the material removed by elution, which has the benefit of imparting biological properties to the material. An outstanding cell viability for a Pluronic-based hydrogel has been demonstrated. A high-performance printable, biocompatible hydrogel has been introduced for printing of permeable vascular patterns, composed of Pluronic and GelMA [316]. The more Pluronic that is included in the cell ink, the more improved the printability is. Pure Pluronic has been utilized as a carrier substrate for the fabrication of vascular structures. An extrusion bioprinter has been proven to produce intricate vascular patterns, while cell adhesion and proliferation of HUVECs have been reported [316].

#### 3.2.3. Hybrid Hydrogels

Hybrid hydrogel networks consist of more than one kind of polymer chain or hydrogel mesh that is covalently linked together and can comprise both natural and synthetic polymers [109]. Hybrid bioprinting is frequently employed to produce increasingly intricate structures and offer increased flexibility in shaping [10,109]. The mixture of PEGDA with alginate is an exemplary hybrid hydrogel [109]. Whereas the PEGDA structures are chemically linked, the alginate polymers are ionotropically linked [109]. Even though these are two separate gelling mechanisms, they combine to create a single structure that has a higher breaking strength and is more resistant to mechanical loads [109]. Another commonly encountered example of a hybrid hydrogel system is the polyvinyl alcohol (PVA)/sodium alginate (SA) hydrogel [199]. The PVA/alginate blend offered enhanced viscosity and enabled direct 3D printing of rigid scaffolds using a core nozzle tip [199].

## 4. Organoids in 3D Bioprinting

Organoids are 3D in vitro tissue models derived from stem cells that can accurately replicate the architecture and functionality of human organs. The ability to generate organoids that mimic the intricate cellular structure of organs has become an emerging breakthrough technique in biomedical science and the development of pharmaceuticals. Conventional methods of organoid cultivation are, however, time-consuming and frequently provide only small amounts of cells, which has resulted in the emergence of the 3D bioprinting of organoids from bioinks that contain suspended cells and intended scaffolds. The aim of this section is to give a brief description of the traditional production of organoids and to discuss their advantages and limitations. It will also provide an overview of the current status of the 3D bioprinting of organoids and its possible applications in the fields of tissue engineering, pharmaceutical screening and regenerative medicine.

### 4.1. Introduction to the Traditional Culture of Organoids

Organoids represent simple tissue engineered cell-based in vitro culture model systems that mimic multiple features of the intricate structure and functionality of the respective in vivo tissue. They can be dissected and examined for basic mechanistic investigations of development, regeneration and repair in human tissues and can also be applied in the fields of diagnostics, modeling of diseases, pharmaceutical development and personalized medicine. Organoids can be derived either from pluripotent or tissue-resident stem cells, either embryonic or adult, or from progenitor or differentiated cells from healthy or diseased tissues like tumors.

Stem cells are crucial for sustaining organ size, structure and functionality due to cell renewal, migration, differentiation and apoptosis [318]. Stem cells are placed in a certain microenvironment, commonly known as the stem cell niche, to govern the fate of stem cells [319]. Considering the relevance of these environmental factors, there have been multiple efforts in tissue engineering to engineer the stem cell niche in vitro to provide high spatial and temporal support for cell–cell and cell–matrix interfaces and to replicate the mechanochemical drivers using engineered hydrogels and microdevices [320,321].

As Matrigel, a basement membrane ECM comprising a unique mixture of ECM compounds and growth factors, has been extracted from mouse sarcoma tumors, it has advanced cell culture systems and has been widely used to support in vitro cell culture [322]. It has subsequently been found that Matrigel enables mammary epithelial cells to grow in three dimensions and create lumens that secrete milk protein [323]. Adult intestinal stem cells incorporated in Matrigel and containing a tissue-specific cocktail of growth factors have also been capable of self-organizing into 3D crypt-villus architectures [324].

An organoid consists of a self-organized 3D tissue that is usually derived from (pluripotent, fetal or adult) stem cells and imitates the essential functional, structural and biological intricacy of an organ [325,326,327]. The cells that make up the organoids can be sourced from induced pluripotent stem cells (iPSCs) or tissue-derived cells (TDCs), comprising normal stem/progenitor cells, differentiated cells and cancer cells [328]. In comparison to traditional 2D cultures and animal models, organoid cultures allow a patient-specific design of the model and simultaneously replicate in vivo tissue-like architectures and functionalities in vitro. Organoid cultivation is more easily amenable to tampering and in-depth biological investigations [329] compared to animal models. Organoid cultures have been utilized for a multitude of applications, most notably in pharmaceutical research [29,330], personalized concomitant diagnostics [330] and cell therapy [329].

Organoid cultures displaying considerable heterogeneity and varying degrees of compositional intricacy may suffer from insufficiently guided morphogenesis in the self-assembly process and are frequently devoid of stromal, vascular and immunological elements [321,328]. Therefore, there is a strong demand to advance organoid culture by exploiting the knowledge of organogenesis and the interplay of cells with their cellular and physical surroundings in the shape of the stem cell niche. Based on this knowledge, bioengineering approaches could be devised to accurately guide stem cell choices throughout organoid development. It is known from investigations into early embryogenesis, for instance, that morphogen gradients control the patterning and development of tissues [331,332]. With the help of microfluidic devices, the desired concentration gradients can be generated by diffusion of morphogens, resulting in the targeted cell types with spatial structuring [331]. In addition to biochemical signals, stem cells also perceive active and passive forces stemming from their external microenvironment and translate these physical cues into biochemical reactions [333]. These physical properties result from the matrix, external forces and/or cell–cell interactions. Instead of depending on a natural or biologically derived ECM like Matrigel, whose stiffness can only be adjusted to a limited extent, synthetic hydrogels or other ECM mixtures can be used to manipulate the physical characteristics of the matrix. The friction of the fluid against the plasma membrane can also apply shear stress to the cells [334]. The dynamic biofluidic surroundings have different consequences for various cell types according to their extent, direction and frequency [334]. Microfluidic systems and bioreactors can therefore be used for perfusion on a micro- and macro-scale [335,336,337]. Cells are recognized to engage with their neighbors and react in a collective way toward external signals [338]; topographical signals, like the curvature and shape of neighboring cells, can influence stem cell decision making [339]. A newly developed neural tube model has effectively dismantled the folding process and shown that geometric restraints can drive the ultimate morphology of neural tube-like structures through micropatterning [340].

It is controversial whether artificially produced cell-based in vitro models like organoids must accurately reproduce the structures and functions of the original in vivo organ. There is a trend towards reproducing the architecture and functionality of in vivo tissues in vitro as far as possible to prove the physiological validity of increasingly complex models. For bioengineers, the artificially generated in vitro models only have to reproduce certain characteristics of the in vivo tissue that are of particular relevance to physiological or pathological functionality. It is optimistic to build highly intricate models and anticipate that they will precisely replicate the organ of origin in vivo. For the majority of scientific issues, simpler models, such as a model with one or two cells within a monolayer or 3D culture, are more reliable for mechanistic investigations and applications [341,342,343] than more complicated models like assembloids or other multicellular models. Experimental aspects of the structure of organoid-based cultivations, which are divided into four main elements, such as cells, soluble factors, matrix and physical cues, and the discussion of approaches for integrating these elements are shown in (Figure 5). A discussion of key considerations for creating more intricate yet resilient organoids, such as cell isolation and seeding, matrix and soluble factor choices, physical cues and integration has emphasized the 3D bioprinting process.

Most of the collective behavior arises with the correct 3D tissue organization and cell constitution, both of which can be delivered through 3D organoid cultures. In 3D organoid cultivations containing tissue-specific morphogens and growth factors, stem cells, including embryonic stem cells, those induced pluripotent stem cells and tissue-specific adult stem cells, those incorporated in Matrigel or under other experimental settings, perform tissue-specific differentiation and morphogenesis and progress to organ-specific tissues. The organoids possess a similar cell constitution, tissue morphology and tissue functionality like their in vivo equivalents; for reviews of advances in organoid systems, see [46,344,345]. It has been demonstrated that various collective cell behavior patterns have been reproduced within organoid cultures [346,347]. The process of developing human organ systems such as the nervous system, the lung system or disease systems, such as tumor organoids, is based on spatially and temporally controlled interactions of cells derived from different lineages [348]. These interactions take place at an early stage of gestation and are thus not amenable to investigations examining neurodevelopmental phenomena or assessing the effectiveness of drugs which target tissues in their native environment. Human neural organoids, stem cell-derived 3D cultures that self-organize and display tissue-like cytoarchitecture and physiology, have been shown to accurately mimic aspects of brain development in vitro [349,350,351,352]. Thus, they are emerging model systems to provide mechanistic understanding of disease etiology [353,354]. Several neural organoids have been merged into single integrated tissues, termed neural assembloids, to enable cell–cell interactions and circuit generation in the developing brain to be modeled [355,356,357,358,359,360,361]. Traditionally, the fusion of neuronal organoids is accomplished by manually placing organoids using a large-diameter pipette tip into a microcentrifuge tube filled with culture medium, where the individual organoids fuse over the period of several days to create an assembloid [362]. The construction of these structures provides temporal controllability of the interfaces between the organoids, but the multidimensional spatial controllability of their merging continues to be a huge task. The incorporation of various cell types into organoids is of great importance not only for the recapitulation of neurodevelopmental mechanisms and the investigation of the etiology of neuropsychiatric diseases. For instance, organoid-based cancer models have proven to be a versatile framework for preserving inter- and intratumoral heterogeneity, allowing ex vivo examination of patient-specific tumor propagation [16,363]. Thus far, two main strategies have been established to reconstruct the cellular microenvironment of tumor and host in vitro. The first is through the use of genetic engineering approaches to achieve the induction of oncogenic mutations and the second is through the co-cultivation of cancer cells with organoid models of the original tissue or the tissue of the metastasis. These models allow temporal support of tumor–host tissue interfaces but provide restricted spatial guidance of juxtacrine and paracrine signaling inside the tumor microenvironment.

### 4.2. Benefits and Limitations of Organoid Cultures

The advantages of organoid cultures are that they enable high-resolution image-based assessment of the spatio-temporal dynamics of cell–cell interactions inside the tissue under examination. The high number of specimens that can be easily generated with organoid cultures allows quantification at the global/atomic scale and systematic investigation of critical stages that collectively modify tissue characteristics. Relative to 2D cell cultures or animal models, organoids offer a more precise depiction of human tissue and provide more robust and effective drug screening and functional evaluation using patient-derived lung organoids [364]. This property is especially useful in the field of cancer research, where organoids can replicate the tumor microenvironment and offer valuable perspectives on tumor-immune interferences and host–pathogen dynamics utilizing pancreatic cancer organoids [365]. Organoids are considered more clinically robust than traditional models because they can mimic the intricate biological processes of human organs in vitro. This property facilitates fast functional evaluation of pharmaceuticals and improves the effectiveness of the route from drug identification to clinical implementation [366]. In addition, organoids constitute a novel stage for in vitro gene editing treatments. Through the use of CRISPR-Cas9 (either to eliminate a gene or rectify a disease-causing mutation) and other gene-editing approaches, organoids can be utilized by scientists to model genetic diseases and evaluate therapeutic options, thereby substantially advancing the domain of personalized medicine [367,368]. In addition, organoids are suitable for the investigation of organ evolution and pathophysiology in vitro due to their capacity for self-renewal and their amenability to genetic manipulation.

The limitations of organoid cultures are that organoids are usually devoid of a vascular system, as it is not possible to implement a circulation system with a flow conditions. Thus, the organoids are grown under static conditions and most of them lack a vascular system. Thus, in the center of the organoid are hypoxic conditions When the organoids are grown over a period of several weeks, the cells inside the nucleus can no longer be adequately supplied with nutrients and the replacement of waste products is hampered. Thus, necrosis can occur, which can even lead to changes in the mechanical characteristics of the organoid and in the cytoarchitecture of cells of the organoid. For these reasons, the broad utility of organoids is restricted. In addition to these obvious constraints, there are others that only become apparent after further analysis of the organoids, including the lack of highly pure cell types, restricted maturation, atypical physiology, possibly circuit formation and the absence of arealization, all of which are characteristics that may compromise their usefulness for specific purposes. Organoids show an elevated expression of cellular stress indicator genes that point to metabolic stress, endoplasmic reticulum stress/unfolded protein reactivity and electron transport disturbances [369,370,371,372]. These disturbances can lead to alterations in the biochemical and mechanical features of organoid cultures.

Thus far, numerous organoid engineering mechanisms have been described to promote organoid cultivation and growth, proliferation, differentiation and maturation. The multiple impacts of factors that operate in the in vivo environment pose a difficulty for the investigation of causality in animal models. As an alternative model to surmount this problem, in vitro 3D organoid cultures have emerged that offer a reductionist model and nevertheless show similarities to in vivo tissue in terms of cellular make-up and tissue organization. In addition, the combination of 3D organoid cultures with other biological and mechanobiological techniques enables a complex multi-purpose application. Some limitations can by optimized as outlined in the following.

### 4.3. Limitations and Optimizations of Organoid Cultures

A major problem is still the temporal and spatial control of the organoids, such as cell–cell interactions. The reproducibility, in terms of both morphology and functionality, of the 3D organoid systems produced continues to be a huge challenge.

**Limited maturity and function:** While none of the existing organoid model systems replicate the full physiological program of cell types, maturation, and/or functioning of the organ in question, they instead feature specific functionality of the tissue they predominantly make up. The overwhelming majority of tissue-derived organoid models lack tissue-specific cell types, comprising niche-specific mesenchyme, immune cells, vasculature, innervation or microbiome. Co-cultures of ductal cells and liver mesenchymal cells have recently been found to reconstruct a section of the architecture of the hepatic portal vein [373]. A particular difficulty is that not all cell types share the identical proliferation rates, growth factor demands or even oxygen exposure needs, such as the hypoxia for the vascular system. Organoids derived from pluripotent stem cells are far more capable of reconstructing the various cell types and cellular interactions of the evolving organ but lack the structures and functionality of adult tissue and the maturation of cells. A strategy that can provide assistance is in vivo transplantation [374]. However, this comes at the cost of maintaining control over the tissue constructs created. In the meantime, differentiation protocols are being improved to enhance maturation and increase the specific functionality of concern. An additional factor influencing maturation and functionality is the (in)accessibility of nutrients and the accumulation of dead cells in cavities. This is especially relevant for iPSC-derived organoids. As the size of the organoids increases, the nutrient support of the cells in the core of the organoid is limited, which causes cell death. This is often the case with organoids that build a denser structure, like brain organoids. In organoids from tissue that build a hollow cyst, such as cholangiocytes and the pancreas, dead cells build up in the lumen over time, which is inevitable but can be remedied by the mechanical fragmentation of the organoids. The continuous fragmentation of the generated structures hinders the conduct of long-term experiments. Organoids derived from pluripotent stem cells, in contrast, cannot be fragmented and passaged, and new strategies are being explored to solve the issue of nutrient accessibility, including long-term preservation of brain slices in vitro [375].

**Restricted regulatory influence on heterogeneity:** As soon as the cells constitute an organoid, there is a minimal influence on the behavior of the cells inside the organoid. The outcome, though in the same experimental setups, is frequently a plethora of phenotypic features, such as shape, size and cell composition, and not a stereotypic culture. The improvement of morphogenic gradients, tissue-specific cell–ECM interfaces and local biochemical and biophysical characteristics are indispensable for reducing batch-to-batch heterogeneity [376]. In the field of organoids, efforts have been made to produce more elaborate multicellular mature and functional structures by generating assembloids, as is the case with human cortico-motor assembloids [359]. This type of effort enables the generation of more complicated structures that combine multiple tissue types with a well-defined interface, such as the interconnection of cerebral cortex, spine and skeletal muscle with neuro-muscular junctions, albeit at the cost of reproducibility. As discussed in another recent report on the organoids of the liver, bile ducts and pancreas [377], there is a reduction in reproducibility in multicellular and cross-tissue organoid systems, as it is difficult to orchestrate the proliferation and differentiation of multiple cell types. The restricted degree of control of heterogeneity within organoids is disadvantageous for high-throughput screening approaches and hampers investigations that need imaging with high spatial and temporal resolution. Rather than building more intricate organoid systems, simpler models with smaller dimensions are progressively being utilized to recreate the key tissue structures and functionalities of concern. Versions of ECM mixtures, microstructured 2D monocultures or co-cultures [378,379], cell sheets [380], 3D stacked textures [340] and micro-positioned ECM supports [381,382] facilitate the generation of reproducible tissue architectures and functionalities with a high level of spatio-temporal control; for instance, through stretching [383] and osmotic forces [384].

**Optimization of ECM formulation:** Engineering approaches have been developed to overcome these constraints. Two main ways to address the need to use non-specific ECM like Matrigel are the application of synthetic matrices with more full control of both composition and stiffness, and the use of decellularized tissue to produce tissue-specific matrices [385,386]. Significant progress is being made to establish chemically defined, GMP-compatible ECMs that permit the growth and long-term propagation of human organoids. In this respect, some progress has been made with human pancreatic, intestinal and colon cancer organoids, which were able to proliferate in a fully defined dextran-based ECM, but failed to grow long-term [320,387].

In the following subsection future frontiers are discussed and an outlook is presented. There is a tendency to create more advanced models that mimic structure and functionality in vivo as closely as achievable in terms of cell types that undergo reconstitution over time, the architecture of the tissue, quantifiable molecular processes and phenotypic functionality. Instead of concentrating solely on the most relevant landmarks or functional testing, an architectural comparison with native tissue is also required. Using the hepatocyte organoids as an example [388], the functions of the hepatocytes are retained, but the architecture of the liver tissue does not correspond to the native tissue, in which the hepatocytes are organized in strands. Similarly, organoids like pancreatic or colon cancer organoids exhibit isotropic growth and develop a cyst instead of the tubular structure they would otherwise develop in their original tissue. To derive more advanced functions, organoids with multicellular and cross-tissue structures will be relevant, particularly in the investigation of cell–cell interactions [389]. In this sense, assemblies and organs-on-chips are also growing in complexity and are being used more and more widely.

In contrast, the engineer’s attempt [342,390] was to adopt a simpler reductionist models defined by the minimal functional modules controlling a complex cell or tissue function of interest to examine mechano-biological causality in the development or repair, or to design a rugged system for high-throughput compound screening. The fundamental assumption is that a complex biological operation is carried out by the orchestrated functioning of a limited set of functional units, each of which is characterized by a small number of molecules, and by chemical reactions that cause alterations in the physical characteristics of mesoscale, such as subcellular or intercellular tissue/multicellular, structures linked to the functionalities of concern in the distinct spatio-temporal stage/phase/step. The bile canaliculi in the liver, for instance, undergo hourly expansion and contraction loops. To investigate the underlying contraction events at high resolution, only areas of neighboring hepatocytes that constitute the bile canaliculi are examined directly in the relationship with the overall regulatory mechanism of the adjacent hepatocytes [341,391]. A much larger structure can be built with cholangiocytes than that driven by the minimal functional modules, but the model will be noisier and more expensive. Every functional unit is linked to a different one and can be analyzed jointly or separately on various length scales. Fundamental reductionist models have proven valuable in providing high-resolution mechanistic insight into morphogenetic processes in tissues, such as in the development of defects [343,382,392].

Geometrically limiting the size of the initial 2D seeding template and 3D formation through micropatterning and promoting 3D cell growth with Matrigel facilitated the inducement of tissue-like neural tube morphogenesis and the generation of highly reproducible neural tubes. This also enabled the identification of the mechanisms of neural tube convolution and the subsequent modeling of neural tube faults [340]. In another case, symmetry breaking in a uniform cell sphere and the formation of a Paneth cell is a seminal step in the early phase of intestinal organoid development. The mechanism has just been clarified: symmetry breaking is induced via transient activation of the mechanotransducer YAP1, which triggers lateral suppression of NOTCH-DLL1 signaling [393]. YAP1 activation have since been precisely regulated by utilizing geometric restraints in hydrogel scaffolds to generate uniform and reproducible intestinal microtissues [320].

Organoids can be confined by shrinking the third dimension in a 2.5D culture. The 2.5D culture minimizes the depth-related fluctuations of a typical organoid: diffusion restrictions in the hypoxic center, restricted access for medication/transfection agents and restricted transparency in imaging [394]. Typical 3D limitations are the culture of cells on curved or patterned surfaces, a flattened or restricted cellular construction [394] and placing the ECM on a flat cell monolayer at a high confluence, which would drag the cells upwards and force increased cell–cell interactions to attain a 3D cell morphology. Hepatocytes within a collagen sandwich have enough contact area to acquire polarity and create a bile canalicular lumen that constricts in the exact same periodic cycles as it does in vivo, in the absence of the 3D network, it is wider and cholestatic in comparison to native tissue. This cell-based model provides a high-resolution breakdown of the bile canalicular contraction mechanism into individual steps and an insight into the molecular mechanism that governs phase transitions [341,395]. Similarly, there could be more artificial organoid models using CRISPR-edited cells to model diseases, even though these cells and models remain synthetic. In addition to technological progress in generating more physiologically valid, rugged and simple-to-use organoid models; however, the impact on applications is expected to be larger. While there have been discussions about substituting animal testing, these efforts have not yet resulted in specific interventions. Organoids capable of reproducing the complex physiological processes in vivo have also increased trust that the new alternative approaches are now feasible choices. Results from animal research are increasingly being transferred to human organoids to gain a deeper insight into human biology and pathophysiology. Consequently, organoids could be used on a large scale as cell resources for cell therapies, regenerative medicine, in-vitro diagnostics and pharmaceutical research.

### 4.4. 3D Bioprinting for Organoid Generation

Collective tissue behaviors, spanning from morphogenesis to the infiltration of cancers, rely on the interactions between cells and cells and between cells and their microenvironment [396]. These processes are gradually being mapped in self-organizing organoid and assembloid models [397]. Biofabrication of 3D tissues that replicate organ-specific architecture and functionality would benefit from temporal and spatial support of cell–cell interactions. While bioprinting is theoretically able to deliver this level of control, it is not well suited to organoids with conserved cytoarchitecture, which are prone to plastic deformation. A platform named spatially patterned organoid transfer (SPOT) has been created, which comprises a hydrogel loaded with iron oxide nanoparticles and a magnetized 3D printer and facilitates the regulated lifting, transportation and placement of organoids (Figure 6) [398]. Cellulose nanofibers are identified both as an optimal biomaterial for wrapping organoids with magnetic nanoparticles and as a shear-thinning, self-healing carrier hydrogel to sustain spatial placement of organoids to ease assembloid formation.

SPOT is used to generate accurately arranged assembloids consisting of neural organoids from human pluripotent stem cells and glioma organoids from patients. In this way, the potential of the SPOT platform to engineer assembloids that can reconstruct important developmental processes and causes of disease has been showcased. Three-dimensional bioprinting, a technique in which cells, frequently with supporting biomaterials, are laid down and assembled into tissues, has been used to achieve control over the spatial organization of spheroids and organoids. The earlier versions of spheroid bioprinting demonstrated the layer-by-layer extrusion of cell aggregates or cylindrical rods [310,399,400,401,402]. These pioneering efforts used primary cell spheroids that had no internal cytoarchitecture, were generally limited to less than 500 μm in diameter and were anticipated to have standardized sizes so that clogging of the nozzles could be avoided [403]. Organ building blocks (OBBs) printing has since been divided into two different types of approaches (Table 1) [404]: The first approach is continuous bioprinting, in which the OBBs are enclosed in the bioink or the supporting scaffold (Figure 6) [405,406], and the second approach is aspiration-assisted bioprinting (AAB) (Figure 6), in which individual OBBs are manipulated through vacuum pressure (Table 1) [88,407]. While continuous bioprinting of neural organoids can generate thick, patterned tissue architectures [406], it is constrained by the inability to accommodate the placement of individual OBBs and the high expense incurred in generating sufficient numbers of OBBs to colonize the bioink or scaffold. Although the processing throughput is considerably reduced, AAB might be more appropriate for spatially structuring the merging of neuronal assembloids in 3D, as it is able to govern the precise 3D location of every OBB. Nevertheless, it was found that AAB is unsuitable for the production of neuronal assembloids because neuronal organoids possess large diameters, a comparatively weak surface tension and a tendency to plastic deformation and breakdown at quite low vacuum forces [408]. An approach termed spatially patterned organoid transfer (SPOT) eases the engineering of neuronal assembloids in 3D with precise spatial controllability across OBB fusion. SPOT uses a cellulose nanofiber (CNF) hydrogel loaded with magnetic nanoparticles (MNP), a CNF support scaffold enclosed in a tailored container and a magnetized 3D printer to guide the spatial placement of the OBBs (Table 1). After merging, the generated assembloid can be detached from the carrier by bioorthogonal, demand-driven disassembly of the CNF scaffold. SPOT is utilized to constrain the spatial positioning of OBBs in two classes of neuronal assembloids. First, for assembloids utilized in neurodevelopmental phenomenon trials, SPOT is exploited to assist in the construction of assembloids consisting of dorsal and ventral forebrain organoids that enable in vitro interneuron migration and integration assays in the cortex. Second, SPOT is applied to assembloids used in translational research on disease progression and therapeutic effectiveness to generate tissues that integrate organoids from human brain tumors into neural organoids. Overall, SPOT has the capability to accurately and reproducibly guide the spatial dynamics of assembloid assembly and thus provide a high-performance framework for building intricate in vitro models of the human brain.

Nevertheless, the synergy of advancements in OBB generation with innovations in biofabrication will be key moving forward as increasing complex interactions between multiple lineages are to be replicated in vitro [409]. A bioprinting system for organoids, such a SPOT, has been designed to place individual OBBs in 3D space while maintaining both a high level of spatial containment and the internal cytoarchitecture. The placement of these OBBs is accomplished through the utilization of an MNP-loaded, bioinert hydrogel that encapsulates the targeted tissue and enables electromagnetically facilitated uplift, transfer and placement within a hydrogel supporting scaffold. OBBs can merge and build assembloids inside this matrix. SPOT can be applied to design neuronal assembloids that function as in vitro models both for a phenomenon of neurodevelopment, that is the migration and integration of interneurons into the pallium, and for the advancement of neuronal diseases, that is the invasion of cancer cells into various brain areas. This magnetic bioprinting technique is based on existing pick-and-place biofabrication techniques [410], such as those employed in AAB [88,407]. In comparison to vacuum aspiration-managed OBB printing, SPOT decreases concentrated force positioning on the tissue surface and is therefore ideal for OBBs with low deformation resistance and for use in applications where the cytoarchitecture of the OBB is important for physiology. While AAB relies on the manual picking of OBBs within a reservoir of media, SPOT uses a customized chip layout with microwells for each OBB. This enables the usage of G-code to automate the finding, picking up and placing of the OBBs at a certain location in the support pool. It is worth noting that the merging of OBBs has also been achieved previously using the Kenzan method, whereby an OBB is aspirated, spiked with a metal microneedle and merged with other OBBs over the course of several such punctures to form a single assembloid [411]. The OBB approach has been automated and marketed, but its dependence on perforation of the OBB and the associated deformation of the perforated OBB makes it very difficult to apply to OBBs with preserved, biologically relevant cytoarchitecture. In addition, the complexity of OBB geometries that can be generated with this approach is restricted due to the stiffness of the needles. Overall, SPOT is a key improvement over other OBB printing technologies as it enables spatial accuracy in 3D without harming the constitutive OBBs.

It has already been demonstrated that magnetic forces can facilitate the creation of patterned 3D tissue from individual cells in a procedure referred to as magnetic levitation, which has meanwhile been marketed [412,413]. Although both magnetic levitation and SPOT are based on MNPs, there are a number of fundamental distinctions between the two platforms. First, magnetic levitation maneuvers individual cells into a chosen geometry. SPOT enables the guided motion of whole spheroids or organoids and is thus particularly suitable for applications where the cytoarchitecture of an OBB is key to its model accuracy. Secondly, magnetic levitation is predicated on the cellular uptake of a bioinorganic hydrogel containing iron oxide, whereas SPOT temporarily coats the surface of an OBB with an MNP-laden hydrogel. This transient MNP exposure restricts the ability of OBBs to experience MNP-driven alterations in cellular phenotype. Hence, compared to this former magnetic bioprinting approach, SPOT is especially useful for the production of assembloids from organoids with retained cellular architectures.

SPOT seeks to offer a complementary approach to conventional assembloid creation concepts that rely on the merging of organoids as a result of entrapment in a microcentrifuge tube [414]. Although these protocols utilize reagents and equipment easily attainable in the majority of biology laboratories, the ease of assembly itself restricts the amount of control that can be exerted over the spatial placement of the OBBs. In addition, linear assembloids from up to three separate OBBs have been verified [415], whereas the creation of assembloids in X, Y and Z dimensions is still a huge challenge. SPOT has the capability to be an advancement over existing OBB assembly approaches, as the 3D printer modified with an electromagnet enables the operator to manage the placement of multiple OBBs across three dimensions. The SPOT platform is designed to be precise, scalable and adaptable to specific needs. Several technical steps have been indicated that may be of potential interest to those seeking to integrate them into their experimental operations. First, the MNP concentration, deposition time, magnetic rod diameter and magnetic field strength need to be properly adjusted for the largest OBB within an experiment. Second, while SPOT can cover a 300–3000 μm span of OBB diameters, it has difficulty precisely isolating OBBs below 300 μm. Optimizing the connection of these organoids to the magnetic rod might solve this constraint. Since this bioprinting framework is OBB-agnostic, it can be used in a broad spectrum of biological systems, wherein signaling from different cell types, cell lines and oncogenic capacity is important. SPOT is employed to construct multiregional neuronal assembloids composed of regionalized components of neuronal circuits and tumor–host assembloids in which the proportion and placement of each OBB can be manipulated in a controllable manner. The combination of the SPOT platform with spatially resolved single-cell RNA sequencing, multiplex time-lapse immunofluorescence and imaging mass cytometry has the potential to provide powerful mechanistic evidence on the spatio-temporal dynamics of infiltrating tumors.

The tremendous impact and potential benefits of 3D bioprinted organoids are enormous, and as the technology is further developed, more uses in disease modelling, pharmaceutical research and regenerative medicine will be realized (Table 1). There are several hurdles that still need to be overcome before 3D-bioprinted organoids can be implemented on a routine basis in the hospital. Nevertheless, the field of organoid 3D bioprinting has an encouraging future and has the capacity to transform the area of tissue engineering and regenerative medicine.

In the field of nanotoxicology, organoid-based scaffolds have been employed for long-term investigations in immortalized cell lines [408]. The toxicity of nanoparticles ingested through physical contact or when inhaled is a major public health issue. It is imperative to perform continuous assessment of the toxicity of nanomaterials. In vitro nanotoxicology investigations are usually restricted to two dimensions Even though 3D bioprinting has recently been used for three-dimensional cultures related to the liberation of medicines and tissue regeneration, not much is understood about its application for nanotoxicology testing. Organoid-based scaffolds have thus been established for long-term studies in immortalized cell lines with the goal of mimicking the exposure of lung cells toward nanoparticles. Viscous, cell-loaded material is printed using a customized 3D bioprinter and then irradiated with either fluorescent latex with a diameter of 40 nm or silver nanoparticles with a diameter of 11 to 14 nm. The administered fluorescent nanoparticles can diffuse in the 3D-printed frameworks, while this has not been the situation with the unprinted frameworks. A marked increase in cell viability of 3D versus 2D cultures being challenged with silver nanoparticles has been detected. This demonstrates toxicological reactions that mimic in vivo experiments, like inhaled silver nanoparticles. The findings provide a new prospect in 3D protocols for nanotoxicology investigations that avoid animal testing. Toxicological and nanomedical investigations necessitate the obligatory step of in vivo studies [416,417]. This is due to the requirement to completely comprehend the bioavailability, fate and biodistribution of nanoparticles throughout and post-exposure, as well as the local and systemic effects they cause. The foregoing step can be largely attenuated by the employment of organoids [304], which replicate the intricate microarchitecture of ECM constituents and the interactions between different cell types adequately to reproduce biological functionalities [55], thereby decreasing the amount of animal testing required for toxicological/pharmacological preclinical evaluation [418].

Organoids can be produced using new techniques such as additive manufacturing [54], which, through the use of a 3D bioprinter [419], introduces an important novelty in the field of in vitro tissue regeneration and examination [420], by replicating the in vivo environment in both mono- and multicellular culture [421,422]. Extrusion-based 3D bioprinting (robocasting or direct ink writing technologies) offers the possibility of producing cell-loaded scaffolds based on biocompatible hydrogels and enables fast, sterile and reproducible manufacturing processes. [423,424]. Only very few investigations on nanotoxicological analysis using cell-seeded or cell-loaded scaffolds from the 3D printer have been conducted, however, perhaps due to the intricacy of reproducing and guiding bioprinted “living” multilayers [425,426]. A tailored and cost-effective 3D bioprinter has been utilized to evaluate the advantages and limitations of bioprinted cell-loaded hydrogel scaffolds [427] or nanotoxicology and nanomedicine investigations or OBST (organoid-based scaffolds for toxicology studies). Hydrogels based on alginate/gelatin/Matrigel at various concentrations were assessed to pick the one most suitable to maintain cell viability and print the cell-loaded scaffolds with a conventional honeycomb structure [428]. In the second phase, the viability of the cells has been characterized, and the findings indicated that the cells could grow in the OBST for 21 days without significant operator interference, proving that their hydrogel composition preserves the cells for a longer period of time and decreases lipid peroxidation. Ultimately, the nanoparticles applied to the OBST have been characterized by a diffuse engagement with the bioprinted cells, which produced a similar toxicological reaction as the in vivo tests with AgNPs. Atoxic carboxyl-modified fluorescent nanoparticles have been used for mapping the distribution within the OBST by two-photon microscopy [429], whereas AgNPs were used due to their known cytotoxicity [430,431,432]. The proposed OBST technique offers various advantages for nanotoxicology/anomedical studies: first, cells can survive for a longer time without undergoing passages; second, nanoparticles can disperse and diffuse in the cell-loaded multilayer by imitating in vivo exposure; third, nanoparticles arrive at the 3D-printed cells in all layers with a significant increase in internalization time in comparison to the non-printed and conventional 2D cultures and fourth, there is a different dose/reaction of 3D-printed cell multilayer toward silver nanoparticles (AgNPs) compared to 2D, resembling in vivo data in zebrafish [430], in insects [433] and rodents [434,435].

To evaluate nanotoxicology studies using 3D-printed cell-laden scaffolds that replicate real cell–cell mixing in vivo [139] and ECM production [436,437], which is difficult to see in 2D, a 3D bioprinter has been created and constructed. The 3D bioprinter has been conceptualized and built to reduce the chance of contamination while extruding the hydrogel with minimal extrusion force and velocity [269]. The 2D expanded cell lines along with their specific supplemental media have been integrated into the alginate/gelatin-based hydrogel formulation [438], which has also been augmented with Matrigel [79], and the final CAD drawings have been constructed based on bio-inspired honeycomb intersecting layers [428], which determines the trajectory for OBST manufacturing. All printed cells remained biologically active, survived for 21 days with minimal operator interference and were capable of internalizing nanoparticles that were subsequently applied to the OBST to simulate the engagement of skin and mucosa with engineered nanomaterials. In addition, the level of thiobarbituric acid reactive substances in Calu-3 dropped markedly by approximately 90% over a 14-day culture [408], indicating that the hydrogel composition can preserve the 3D-printed cells for long-term studies. Thiobarbituric acid reactive substances pointed out that an adaptation period to the 3D environment has been necessary for all cell lines under investigation. As previously stated for cell proliferation, the cells regained their normal biological functionality after a short period of time and reverted to a condition of decreased membrane lipid peroxidation [408]. The amount of dead cells stayed minimal throughout this time period. It has also been essential to guarantee uniform cell deposition when creating an OBST. Unprinted scaffolds showed not merely unevenly distributed cells, but also air bubbles and uneven internalization of the nanoparticles by the OBST. It is finally noteworthy that the 2D in vitro toxicity of high doses of AgNPs to mammalian cells is unquestioned [439,440], the OBST obtained findings are similar to the in vivo toxicological results in rats following inhalation of AgNP. In addition, a less pronounced decrease in viability of 3D-printed cells subjected to AgNPs has been seen, and the ability to study the same cells imbedded in a 3D architecture for weeks could speed up the process from lab to patient. Cell-loaded 3D-printed multilayers offer the possibility to investigate lipid peroxidation longitudinally, as the damage caused by oxidative stress is similar to that of lung tissue in vivo [417] and, and according to the physicochemical properties of the nanoparticles [441], they can be used for toxicological or nanomedical investigations, as the nanoparticles can diffuse effectively into the printed layers. Although the OBST technique is at an early stage and needs further studies, it could be an effective instrument for nanotoxicology studies where the cells incorporated in the 3D hydrogel are active and can interact with the nanoparticles [442] and the scaffold [443]. In conclusion, the research emphasizes the significant differences between 2D and 3D data, suggesting that consideration needs to be given to reviewing tactics in the fields of nanotoxicology and nanomedicine to account for possible impacts on cell morphology and cell–cell interactions in a 3D environment. Ultimately, the technology can aid the development of safer and more powerful nanomedicine and represent a useful tool for scientists in the nanotoxicology field. There are certain limitations, nevertheless. For example, the number of cells seeded strictly relies on the 3D printing parameters and is usually lower than the amount of cells embedded in the hydrogel-loaded syringe; this is due to the remaining volume in the syringe’s Luer lock and the force applied during extrusion, which is minimized but mechanically destroys the cells. Consequently, at least two major questions arise: How can scaffolds be employed in the 3D organoid culture? How is bioprinting helpful?

## 5. Three-Dimensional Bioprinting of Complex Geometric Models with Tumor Organoids Serving as Structural Elements

Three-dimensional bioprinting is a promising biomanufacturing process that produces multiscale tissue models of remarkable accuracy and physio-mimetic performance [444]. The most widespread technique is extrusion-based bioprinting, also referred to as bioplotting. In this technique, endless filaments of bioink are pushed or mechanically extruded via printer nozzles into predefined 3D structures [445]. Whereas in conventional bioprinting through extrusion, the bioink is deposited layer by layer onto a carrier substrate. In a new technique referred to as embedded bioprinting [446], the bioink is placed in a slurry pool in which it is held against gravity, enabling the construction of intricate textures, such as blood and lymphatic vessels [447], which also extrudes cell-laden bioinks via nozzles. In this process, however, thermal, electrostatic, or piezoelectric techniques are used to supply droplets of bioink instead of continuous filaments so as to obtain a higher level of resolution of the deposit. There are also other nozzle-independent bioprinting techniques, like acoustic printing [448]. These droplet administration techniques can deliver cancer cell-loaded bioink droplets with enhanced spatial resolution into the tumor-specific CAF-harboring scaffold [448]. The most frequently employed technique is extrusion-based multi-material bioprinting, where coaxial printheads or multiple nozzles are deployed to extrude several bioinks with varying cell and material combinations [449]. Inkjet and acoustic printing are emerging bioprinting techniques that facilitate multi-material engineering to enable the concurrent deposit of various cellular components and matrix substances [450]. These advances allow improved design of advanced tumor models with more heterogeneous compounds to mimic the fundamental cell–cell matrix interplay.

A miniature brain model has been designed with the aid of the multi-nozzle bioprinting technique in which GBM cells and macrophages have been placed in specific compartments of the GelMA [451]. This model has consistently replicated the cellular interplay between neoplastic and immune cells, involving the enrollment and the transformation of macrophages into the GBM-associated macrophage phenotype and the invasiveness of GBM cells into the mini-brain tissue, correlating well with clinical transcriptome results. Most investigations are performed with monodispersed cancer cells that serve as bioprinting building elements. Monodisperse cells, however, are unable to accurately mimic tumor propagation, as volumetric cancer cells hardly ever occur in isolation [452]. Organoid-based bioprinting constitutes a viable mechanism to incorporate miniaturized tumor aggregates within a heterogeneous 3D cavity with assisting hydrogels and stromal cells. The combined effect of these features enables self-organization of tumor-sized anatomy with hierarchical functional modules [453], which provides a more accurate reflection of intrinsic TME features. A growing demand and unparalleled possibilities are emerging for the creation of novel and more efficient tissue engineering techniques, among which 3D bioprinting is considered one of the most encouraging. Although biomaterial-dependent 3D bioprinting is progressing continuously, it is still slow to yield the expected therapeutic outcomes. Alternative “scaffold-free” 3D bioprinting methods are therefore currently undergoing rapid progress. The readiness of bioprinting techniques and the quality of the bioprinted structures should be assessed before they can be employed for therapeutic features [48,454].

### 5.1. Scaffold-Free Bioprinting of Tumor Spheroids

Various bioprinting techniques for the accurate placement of cell assemblies have been established. The initial approaches of these techniques employed a more accessible form of tumor aggregates termed tumor spheroids that are more robust for the bioprinting procedure compared with PDOs. Among the initial concepts of bioprinting of tumor aggregates is the Kenzan technique. In this technique, the aggregates are positioned on a microneedle arrangement with the aid of a robotic stage in a cohesive template designed in advance. The microneedle array acts as a guide for the aggregates to merge into a larger cellular entity and initiate the synthesis of their individual ECM scaffold. The Kenzan technique eliminates the reliance on biomaterial frameworks, which is why it is also referred to as a framework-free method. It is suitable to produce microtissues with elevated cell density and direct cell–cell interfaces, including adipose tissue, cartilage, nerves and heart muscle-like assemblies. In tumor sculpting, the Kenzan technique offers a powerful way to replicate the two-way interaction between tumor mass and adjacent mature tissues. Using this approach, the neuro-like parenchyma enveloping glioma tumor spheroids has been recreated to assess real-time invasion of the cancer [455]. Nevertheless, this model cannot represent the relevant gliosis features of glioma disease pathology as it is not able to reproduce various cell–ECM compounds and interferences [455]. The Kenzan method is restricted in its spatial resolution and does not control for biochemical/biophysical issues associated with the ECM framework. In addition, this technique has not yet been implemented on tumor organoids, probably since the mechanical interruption of this procedure can lead to exaggerated injury to the tumor organoids, which would lead to decreased survivability. Another scaffold-free technique is the liquid-based singularization approach, where single tumor aggregates are trapped and liberated one after the other through a fluid-controlled rear pressure [456]. This singularization unit can be incorporated into bioprinting nozzles to accurately place individual spheroids into 3D bioprinting frameworks. This approach facilitates the assembling of tumor organoids with the other key features of the intricate TME and guarantees high accuracy of printing [456]. It also allows the creation of multi-level structures with tissue-specific forms and quantities. For instance, the liquid-based singularization method has been applied to produce a macroscale ovarian model in which heterotypic spheroids comprising both ovarian adenocarcinoma cells and fibroblasts have been assembled in a 3D-printed hydrogel framework [457]. The growing scale and sophistication of TME has decreased the sensitivity to doxorubicin in comparison to individual spheroid units. Nevertheless, the slow operating pace restricts the practicability of this technique for large-scale and high-performance processing.

### 5.2. Scaffold-Based Bioprinting

In bioprinting processes, hydrogels are often used as carrier substrates in 3D bioprinting. These carrier media consist of so-called sacrificial inks and carrier baths, which can temporarily mechanically reinforce the bioink during the printing process. Hydrogels featuring reversible sol-gel phase properties have been employed as temporary sacrificial inks to enable printing of cavities, incorporating permeable channels [444,445]. Hydrogels with thixotropic mechanical characteristics may be employed as carrier baths that facilitate the printing of intricate structural characteristics with enhanced print detail while maintaining physical constraint throughout printing [281,458,459]. The utilization of gel-phase carrier media has significantly expanded both the level of complexity of printed geometries and the spectrum of materials that could be applied as bioinks [82,460]. Poly(ε-caprolactone) (PCL) and GelMA are two well-known biomaterials for printing. Multiple scaffold-based bioprinting efforts have employed biocompatible hydrogels to embed spheroids/organoids and several stromal cell types. This approach relies on the bioprinting of monodisperse cells by mitigating mechanical disturbances and simultaneously guaranteeing correct dimensional placement, intricate geometry and hierarchical diversity [407]. Due to their excellent biocompatibility, scaffold-based bioprinting procedures have shown considerable benefit for the generation of tumor organoids, where also mechanical characteristics are important. Bioprinting methods employed for framework-based bioprinting involve drop-on-demand, acoustic bioprinting and extrusion bioprinting. The models produced with every technique differ in terms of scale. The drop-on-demand process can manufacture smaller-sized models ranging from 100 to 1000 µm, with a droplet size from 42 to 960 µm [461], while extrusion bioprinting creates larger-sized models ranging from 10 nm to 100 nm with a nozzle diameter ranging from 260 µm to 1200 µm [462,463]. The drop-on-demand method enables the production of textures with high output and a high level of uniformity. This method is generally ideal for the printing of very small structures measuring 100–1000 μm and utilizing low viscosity bioinks. With this setup, bladder tumor assembloids harboring PDOs originating from luminal/basal phenotypes and fostering CAFs and ECs could be established [464]. Accurate recapitulation of the TME resulted in the tumor assembloids having similar structural features to the original cancer tissue with interconnected vasculature [465]. The assembloids also effectively prevented the transition of phenotype from luminal to basal, which invariably arises in the long-term culture of organoids of luminal urinary tract carcinomas [466].

Acoustic bioprinting represents a further method for the droplet-based administration of tumor spheroids/organoids in a physically defined way [467]. In acoustic bioprinting, the droplets are expelled through a mild sound field. This technique can shield living cells from harmful stress agents like heat, intense pressure, high tension and substantial shear forces that are present in alternative bioprinting approaches [468,469]. Nevertheless, acoustic bioprinting is limited in its capacity to expel droplets of a high viscosity bioink, in a similar way as the drop-on-demand technique [468]. A patient-derived colorectal cancer microtissue has been bioprinted with tumor and healthy organoids via droplet release [467]. Extrusion bioprinting can also be expanded to the bioprinting of spheroids/organoids [470]. In comparison to drop-on-demand and acoustic bioprinting, a broader spectrum of bioinks can be utilized in this approach as shear-thinning hydrogels can be employed. This strategy also allows manipulation of several materials and thus supports the development of a tumor-sized assembly with various ECM compounds and diverse cell types [471]. A co-culture model of breast cancer preshaped spheroids and ECs has been fabricated via extrusion bioprinting [472]. This model demonstrated increased tolerance to paclitaxel therapy when compared to a monodisperse cell printing model, emphasizing the significance of cell association in the therapeutic outcome [471]. Although advances have been achieved to this point, the bioprinting of tumor spheroids/organoids needs additional fine-tuning to accurately adjust the characteristics of the bioink and broaden the manufacturing options. Volumetric bioprinting (VBP), where liquid resins are photopolymerized into a volumetric 3D pattern with light, has emerged as a high-performance bioprinting method with high resolution and manufacturing capability for the fast build-up of demanding patterns. VBP takes computed tomography as its inspiration and uses a digital light processing engine to create a sequence of 2D light designs projected onto the paint reservoir from various angles, causing the build-up of light dose and solidifying the resin as it approaches the gelling level. Bernal and colleagues [473] used VBP and liver organoids have been employed to fabricate intricate, centimeter-sized hepatic reconstructions in seconds. This advanced method is without nozzles or layers and guarantees a high degree of survivability and form accuracy of the emerging organoids [473]. While VBP continues to suffer from the issue of light scattering and is therefore not applied for tumor modeling, it is nevertheless a powerful tool for replicating human-scale cancer models [473].

### 5.3. Mechanical Cues of Scaffolds Impact Organoids

Classical strategies for building organoids still cannot accurately reproduce the key features of real organs as it is challenging to regulate the self-organization of cells in vitro. The current constraint stems from the impossibility to manipulate the organoid environment in the classical production of organoids and the scarcity of information and actual measurements of the biomechanical characteristics of tissues in developing organs, such as the human brain, or in diseased organs suffering from cancer, injury or inflammation [474,475,476,477,478]. Nevertheless, the knowledge of mechanical characteristics has increased continuously due to new developments in biophysical techniques [241,475,476,479,480,481,482]. Thus, this information on mechanical cues can be used to mimic them in scaffolds for the generation of organoids. Mechanical characteristics of 3D bioprinted scaffolds, such as softness (or rigidity) or curvature, can contribute to the morphology and functionality of organoids [483,484]. For instance, it has been found that the conversion of a round intestinal stem cell (ISC) colony into a crypt-containing organoid inside a synthetic hydrogel necessitates a softening of the matrix [320]. Nevertheless, the global matrix softening model employed led to stochastic and spatially unsupervised budding, just like in traditional organoid cultures using native ECM 3D matrices [324,485].

## 6. Organ-on-a-Chip Convergence for Optimized TME and Bioprinting

Organ-on-a-chip technique can be partly bioprinted. The combination of organ-on-a-chip technique and bioprinting seems to be interesting for developing more intricate model systems in a reliable manner. Intercellular communications are critical for the proper operation and evolution of organisms. Cells can communicate with one another both directly (in physical contact) and indirectly (paracrine signal transmission). These interactions constitute the fundamental elements of physiological communication and are vital for the generation of tissue, the immune response, homeostasis and regeneration. During direct cellular communication, cell surface contact can involve gap junctions, cell adhesion, tunneling nanotubes and ligand–receptor signaling. In indirect cellular communication, cellular messages are exchanged by signaling extracellular vesicles, including exosomes and ectosomes, cytokines, chemokines, growth factors, miRNAs and metabolites. They all contribute to the development of tissue and physiological responses [486]. A healthy immune system can accurately recognize and eliminate cancer progenitor cells before they cause damage via a mechanism referred to as tumor immunosurveillance. Numerous extrinsic and intrinsic agents interfere with the communication between the immune system and cancer progenitor cells, leading to tumorigenesis. Cancer cells escape the host immune system and perturb it, resulting in an immunosuppressive TME. The TME consists of a complicated ecosystem of cancer cells, immune cells, fibroblasts, ECMs and regulatory molecules. The immune cell–cancer cell exchange within the TME evolves and can lead to the generation of either pro- or anti-tumorigenesis [487]. Reestablishing the immune response and the interaction of healthy cell–cell communication within the TME is an integral part of cancer immunotherapy [488]. While a variety of other immunotherapies such as immune checkpoint inhibitors, oncolytic viruses, bispecific T-cell engagers, cytokine therapies and adoptive cell therapies have been identified, the central idea of immunotherapy is to reconstitute or reenable the patient’s host anti-tumor immune system [488]. Despite the potential of these immunotherapies, clinical responsiveness differs greatly between patients, primarily due to differing combinations of immunosuppressive TMEs and varying interference patterns with cell–cell interactions. Approximately 30–40% of immunotherapy patients achieve a positive response, with only a few obtaining a permanent response [489]. The TME versatility is mostly linked to the varying degrees of tumor-infiltrated lymphocytes (TILs) and their functionalities. Certain patients experience “hot” cancers, where the cancers contain increased TILs, and usually respond favorably to immunotherapy [490]. Conversely, some patients present with “cold” cancers, with few or virtually no TILs, and these cancers frequently evolve into immunotherapy resistance [490]. Clarifying the resistance mechanism and determining the aberrant nature of intercellular communication among cancer cells and immune cells in TMEs is key to developing more effective immunotherapies. Although advances have been achieved lately in the bioprinting of organoids/spheroids to implement the build-up of heterogeneous TME compounds in a biomimetic matrix [463], there is clearly an enormous amount of work to be accomplished to replicate intrinsic cancer progression, pharmacokinetics and pharmacodynamics in vivo. These pathological events rely on tumor-immune communication and interactions among various functional organs, which are greatly streamlined in tumor models cultured in a static environment because of the absence of a functional circulatory network [491]. Organ-on-a-chip systems are being investigated as an advancement in cancer models, as they have excellent capabilities to mimic natural vascular perfusion and blood microcirculation. The organ-on-a-chip comprises complex compartments and microchannels capable of being interconnected to mimic multi-organ interaction and facilitate dynamic liquid circulation. This enables the manipulation of the physical and biochemical variables of TME, such as oxygen levels, pH equilibrium, nutrient distribution, molecular gradients and, most importantly, the flow of circulating cellular elements [492]. Moreover, organ-on-a-chip technology enables the utilization of novel integrated physical, biochemical and optical sensor devices. These modular devices can be utilized for in situ observation of TME variables and dynamic tumor reactions to active drugs over a longer period of time [493]. In spite of their ability to apply dynamic biochemical and fluidic feedback signals in vitro, conventional methods for fabricating chip devices, including micromachining and soft lithography, are labor-intensive, time-consuming and offer a restricted capacity to accurately dispense living parts [494]. Thus, the organ-on-a-chip for prototyping bioprinting is a challenging way to tackle these hurdles [495]. The technology can improve the way it mimics the TME’s intricate characteristics and support the advancement of powerful cancer treatments [496].

### 6.1. Assembly Strategies and Shapes for Bioprinting Tumor-on-a-Chip Models

A potential strategy for combining bioprinting and organ-on-a-chip technology utilizes a post-integration technique in which the living parts are bioprinted into pre-engineered microfluidic assemblies. This approach optimizes the application of microfluidic fabrication methods to create chip carriers with intricate microflow channels and immature vascular meshes. The downstream bioprinting procedure facilitates the insertion of biological elements into the chip carrier in a space-time fashion [497]. As part of the post-integration approach, nozzle-based bioprinting techniques like extrusion and inkjet bioprinting are commonly utilized to incorporate multiple bioinks and mixed cell types into microfluidic systems [498]. Two different types of organ-on-a-chip device, namely OrganTrial^®^ Dolores and OrganTrial^®^ Hive, have been presented that employ the post-integration technique wherein bioprinting models of different types of tissues can be incorporated into commercially available mounting devices fitted with culture compartments, microfluidic channels and pulsation-controlled features to enable the dynamic cultivation of macro-scale tissues and regulate multi-organ interaction [499]. In addition to traditional chip manufacturing techniques, 3D printing enables the fabrication of chip covers and microfluidic channels. This technology forms the foundation for a one-step production approach in which chip components and biological parts can be manufactured at the same time. The one-step manufacturing approach opens the door for diverse architectural patterns and facilitates the creation of customized tumor-on-a-chip models [500,501], such as brain tumor models [502]. This strategy, nonetheless, demands multi-material manufacturing capabilities that can introduce cell-loaded bioinks and cladding materials into intricate structures concurrently. Extrusion printing meets these demands because it can employ several print heads for various purposes. For instance, a breakthrough printing system adopts a photopolymer head, a UV head, a microplasma head and a biologics head to implement the free-form manufacturing of a tumor-on-a-chip apparatus in a single step [503]. Stereolithography represents a further established method for the fabrication of high-resolution chips, although it is extremely challenging to customize for the manufacture of multi-materials. Hybrid printing methods could be more practical and offer more freedom to create a broader model with a one-step manufacturing approach [504]. Another problem is the limited choice of the shell substrate. The circumferential material utilized in one-step manufacturing needs to be adequately printable, biocompatible and sustainable to preserve the structural integrity of the complicated chip architecture in long-term culture environments, which limits the selection of appropriate circumferential compounds [140]. Compared to the post-integration procedure, the living cells must undergo the lengthy printing process in one-step production, which represents a considerable challenge for the cells’ ability to survive. In addition, the higher degree of difficulty of the devices leads to lower accessibility, as there are only a few commercial offerings and most of the devices used in scientific research are manufactured by the scientists themselves. Before the one-step procedure can be introduced into clinical practice on a larger scale, the limitations in material selection, turnaround time and accessibility of the devices must be eliminated [504]. In the last few years, considerable advances have been achieved in the creation and utilization of bioprinting-based tumor-on-a-chip models that emulate TME’s spatially distinct features and dynamic culturing conditions [505]. Several important engineering aspects play a crucial part in the production process of these intricate 3D cancer models. Due to the advantages in terms of the accessibility of equipment, the availability of materials and the fast process of cell loading, the post-integration approach has prevailed in modern 3D bioprinting tumor-on-a-chip models compared to the one-step option [506]. In addition to the manufacturing process, the selection of cell components is another decisive aspect [506]. The use of tumor organoids rather than monodisperse individual cells as components is more efficacious in simulating natural tumor–stromal relationships. This design rationale has been accomplished with organoid/spheroid bioprinting methods that guarantee the construction of 3D tumor assemblies with other major TME constituents, comprising stromal cells, vascular/lymphatic vasculature and supportive ECM scaffolds inside the chip units [92,504]. In a 3D bioprinted neuroblastoma-on-a-chip model, Ning et al. [504] demonstrated dynamic tumor–vessel interfaces with success, showing the aggressive response of the tumor and allowing the evaluation of metabolic, cytokine and gene expression patterns in various TME environments. Notwithstanding present advances, full recall of the intricate interactions between tumor and stroma still needs additional work to establish the native TME’s composition and its unique geometric features. Innovative digital image processing and quantification methods with artificial intelligence are highly beneficial for determining the exact patient-specific TME criteria [507]. In improving the accuracy and reliability of tumor models by helping to develop a more informative structural blueprint, these technologies can offer a powerful tool for investigating cancer biology and drug sensitivity [508].

### 6.2. Enhancement of Integrative 3D Tumor Models: Vascular System, Immune Monitoring and Tumor Metastasis

Tumor microvascular structures are key drivers of tumor evolution as they supply nutrition and oxygen for the survival of the cancer and can create routes for cancer metastasis [492]. Similar to healthy tissue, the development of blood vessels in cancer can occur through angiogenesis, i.e., the growth of blood vessels through the sprouting or splitting of already formed blood vessels, and/or through vasculogenesis, i.e., the formation of new blood vessels from endothelial progenitor cells formed in the bone marrow [509]. The new neoplastic capillaries, which are connected to established blood vessels, form a hierarchical vascular system with vessels of diverse diameters. Mimicking this hierarchical organization is key to replicating the inherent characteristics of tumor–vessel interfaces and pharmaceutical kinetics. The organ-on-a-chip approach has enabled two major approaches to design vascular replicas at various length scales. The first technique is the vascular patterning, in which prefabricated microfluidic tubules layered with ECs form microvessels of diameters larger than 100 μm. The second technique involves self-organization, in which single ECs pass through a process resembling vasculogenesis, allowing the generation of capillary equivalents with a lumen width of between 15 and 50 μm [492]. Whereas traditional microfluidic techniques are only able to fabricate single-size vessels, the integration with bioprinting facilitates advances in the direction of integrated hierarchical vessels. With the help of multi-nozzle bioprinting, it is possible to introduce substitution material into vessel-like 3D geometries and encase them in EC-loaded hydrogels. After discarding the voiding material, the void channels can be layered with ECs to create a sealed vessel-tissue boundary. Three-dimensional bioprinting can be employed to produce vessel-like 3D embodiments that are far more intricate and lifelike than traditional “2.5D” designs in which 2D channel patterns are extrapolated into the third axis [510]. Simultaneously, the ECs enclosed in the encapsulating hydrogels are subjected to vasculogenesis-like self-organization to create an equivalent capillary mesh. This hierarchical vascular architecture has been recreated in a neuroblastoma model effectively mimicking the organization and functionality of cancer tissue with major EC-lined vessels facilitating perfusion and capillary meshes facilitating tumor–vessel communication [504]. In the engineering of engineered tumor-on-a-chip models, lymphatic vessel systems should also be addressed. Microcirculatory systems have been recreated in vitro with matched vascular and lymphatic vessels to improve the simulation of the trafficking kinetics of therapeutic cancer drugs [449]. Numerous biomanufacturing techniques such as electrospinning, decellularization of xenogeneic vessels, 3D printing and melt electrolysis have been investigated [511,512,513,514,515,516]. Nevertheless, obtaining a homogeneous cell dispersion within the vascular beds is still a huge task [516]. None of the methods listed so far enables a selective and accurate positioning of cells within a 3D-like architecture. In addition, the manufacture of multilayer functional blood vessels with these techniques necessitates a complicated multi-stage production process in which each layer demands a specific maturing time. Overcoming these main disadvantages has prompted the recent adoption of 3D bioprinting technology, which offers unparalleled benefits [517]. This new technology facilitates the production of cellular 3D configurations with various layers in a single manufacturing stage, thereby reproducing the inherent hierarchical complexity of vascular tissue [157,518]. Pneumatic extrusion-based bioprinting appears to be the most versatile bioprinting strategy to generate large vessels characterized with centimeter-sized tubular structures. In this process, the bioink is extruded through a nozzle using a pneumatic system and applied layer by layer to a mounting substrate. A wide range of bioinks can be used in this technology. The most employed kinds of bioinks are hydrogel-based, water-swollen polymer structures whose formulation can be tailored to mimic the ECM environment of biological tissue. Nevertheless, the weak mechanical characteristics of hydrogels restrict their usability for vascular tissue engineering. The use of a synthetic material that acts as a scaffolding could be a potential solution [100,115]. For example, three-layered vascular scaffold has been generated with PCL as a supporting material [519]. Between two PCL layers, a layer of bioink comprising cells and 3% sodium alginate has been printed to provide proper structural support [80]. Similarly, poly(ethylene glycol) tetraacrylate (PEGTA) has been utilized as a carrier system for a cell-responsive bioink consisting of GelMA and sodium alginate [520]. The bioinks utilized in these studies, nevertheless, are not sufficiently representative of the native ECM constituents. The main components of the ECM like collagen, elastin, microfibrils, proteoglycans, GAGs and several growth factors are indispensable for the preservation of the structural integrity of large vessels in tissue-engineered vasculature [521,522]. Bioinks on the basis of dECM have recently been established. The process of decellularization makes it possible to retain the native microenvironment of the vessels established by the ECM, which encourages cell growth and a non-immunogenic tissue, while cellular and nuclear components—with a special emphasis on DNA and RNA—are removed from the tissue [523,524,525,526]. This strategy therefore permits the yielding of a bioink consisting of biochemical features existing in the natural surroundings. The development of a novel bioink made from dECM and natural hydrogels has been presented, capable of replicating large vascular substitutes that meet the requirements of the form, functionality and integrity of natural tissue [527]. This strategy brings benefits of the 3D bioprinting and decellularization processes together. After optimizing the decellularization protocol, the resulting dECM has been integrated into a bioink whose composition ensures printability and thus overcomes one of the biggest hurdles of extrusion-based 3D bioprinting [157,528,529]. Finally, the biocompatibility of the bioink and cell penetration have been verified by observing cell growth in 3D-printed constructs over an extended period of time [527].

Moreover, the tumor vasculature is both in structure and function aberrant relative to the vascular system of healthy tissues. Tumor blood vessels become untight and tortuous, marked by the presence of malformed ECs, detached or displaced pericytes and incomplete basement membranes. The impermeable vessels, along with the high compressive pressure from condensed cancer cells, compromise the blood perfusion of the tumor mass and result in interstitial high blood pressure, hypoxia, and acidosis, which have been demonstrated to ease the infiltration of cancer cells, interfere with drug distribution, and lead to immune cells invading with cytotoxic capabilities [530]. These circumstances highlight the relevance of delineating abnormal characteristics of tumor vasculature in in vitro cancer models. Interaction between tumor and vessels yielded abnormal alterations in preformed blood vessels due to the surrounding of inflammatory breast cancer [531]. Vascular abnormalities have been characterized by evaluating endothelial shape, cell–cell connections, matrix porosity, endothelial confluency and permeability [532]. The approach of quantification enables the investigation of tumor–vessel interfaces and eases the upcoming trend of vessel normalization, where the organization and functioning of tumor vessels is adjusted to optimize the effectiveness of cancer therapies. The congenital tumor vascular arbor, nevertheless, is more intricate than the EC-lined cavity channels engineered in recent reports. Therefore, additional work is required to decipher and evaluate tumor vasculature aberrations. With recent progress in immunotherapies demonstrating encouraging anti-cancer effectiveness, there is a pressing demand for vigorous research frameworks that can fully uncover the interplay between tumor and immune system. Besides the tumor-infiltrating immune compounds that have been demonstrated in various PDO models [26,44], the involvement of peripheral immune communities residing in the vasculature deserves attention [533]. With the help of microcirculation meshworks delivered through bioprinting tumor-on-a-chip technology, peripheral immune cells can be delivered through perfusion, allowing the synchronous replication of tumor-infiltrating and peripheral immune factors in a person-specific TME. This concept supports basic immuno-oncology research, enables efficient therapeutic combination screenings and opens the door to the development of accurate immuno-oncology.

Tumor metastasis models are fundamental for unravelling the intricate mechanism of metastasis of cancer to remote organs, which accounts for the majority of human cancer-related mortalities. Multiple frameworks concentrating on the extravasation mechanism have been established using replicas of the human microcirculation, cancer cells circulating intravascularly and immune cells [534]. A high-resolution imaging technique facilitates the visualization of the extravasation mechanism to examine the fundamental regulatory mechanism [535]. However, the extravasation of cancer cells is a single step in cancer metastasis, which is a dynamic and multifaceted phenomenon involving surrounding the ECM invasion of cancer cells from the primary tumor mass, intravasation, vascular spread, extravasation and dissemination to distant target organs [536]. To identify the critical stages of metastasis, it is of great importance to connect the primary tumor and its potential metastases through a microcirculatory system. The integration of 3D bioprinting to generate this multi-organ-on-a-chip structure has been shown [537,538], but still requires further research efforts. Thus, the complete elucidation of the complicated multi-organ crosstalk during cancer metastasis requires further investigations, which may require the establishment of new techniques [538].

## 7. Combination of Organoid Bioprinting with Partially Bioprinted Organs-on-a-Chip Approaches

As organoids lack the flow-driven cultivation of organ-on-a-chip, this weakness can be overcome by using a combined approach called organoids meet organs-on-a-chip systems. Moreover, organoids or assembloids and organoid-on-chips will also provide new platforms for the analysis of collective cell performance in settings of organ-to-organ interactions and host–pathogen interferences [539,540,541]. Although there are numerous advantages to utilizing organoids, many present systems lack the cross-tissue cell–cell interaction that can induce and sustain collective cell performance in the in vivo milieu. For instance, the geometric boundary can change the cell stage via mechanochemical feedback. In embryonic intestinal tissue, such a boundary is established initially by mesenchymal condensation forming under the intestinal epithelium [542]. It was found that co-culture of the intestinal organoid with fibroblasts in a 3D collagen I scaffold replicates the microenvironment of the intestinal stroma and allows the investigation of interactions between epithelial cells and fibroblasts [543]. In the future, the co-culture of fibroblasts, immune cells and/or neuronal cells within organoids embedded in Matrigel or artificial scaffolds, which is currently being established, will expand the study of collective cell behavior in a tissue-authentic environment.

Cells expanded in organs-on-a-chip cultures have been found to upregulate their functions through maximizing mass transfer and reducing shear stress in the perfusive soluble microenvironment, bringing them one step nearer to a true natural tissue [544,545,546]. A recent experiment demonstrates how the presence of fluid flow facilitates the maturation of renal organoids and their vascularization in vitro [547]. Physical limitations have been built into the organoid surroundings, and the intestinal cells self-organized into crypts of the identical size when delineated by artificial scaffolds [381]. Simultaneously, they surmounted the impenetrability of cystic organoids and the elimination of cell debris through the generation of a permeable culture of mini-intestines in which the cells are positioned to generate tubular epithelia and have a similar spatial organization to the tissue in vivo [548]. Future investigations could use the stage to examine the development of different tissues or the etiology of a variety of diseases, thus enabling the identification and preclinical testing of drugs. In all these techniques, organoids or assembloids are preferred for bioprinting. There are fewer arguments for the use of organs-on-a-chip systems due to the weaknesses in mimicking the native in vivo environment of organs. In addition, organ-on-a-chip systems can only be partially printed [549]. However, 3D bioprinting can be utilized to produce not only microfluidic chips made of materials like resins and polydimethylsiloxane, but also biomimetic tissues derived from bioinks like cell-laden hydrogels. The combination of both techniques, such as bioprinted organoids and bioprinted organ-on-a-chip systems, holds great potential for improved organoids or assembloids-based model systems in a dynamic environment, such as a fluid-flow condition.

## 8. Four-Dimensional Printed Materials for Cells, Hydrogels and Organoids

Four-dimensional printing is a process for producing a model system from one or more materials that can be transformed from a 1D strand into a pre-programmed 3D shape or from a 2D surface into a pre-programmed 3D shape and is also capable of transforming between different dimensions [550]. These transformations are accomplished, for instance, by heating, light or swelling in a liquid, electrochemically and by programming different degrees of sensitivity, e.g., for swelling, in various areas of the designed geometry [551]. These techniques offer flexibility and dynamic behavior for structures and systems of all sizes and open up new possibilities for incorporating programmability and simple design principles into non-electronic materials [552]. In biological systems, the development of the macrostructure of the engineered material is frequently altered by the soaking of the hydrogels, as the engineered material is grown in aqueous solutions to maintain the cells in a hydrated state. The swelling characteristics of a bioprint are determined by the intrinsic polymer solubility, the crosslinking degree and the degree of heterogeneity of the structure [553,554]. For instance, the frequently employed hydrogel PEG expands substantially due to its highly hydrophilic, water-soluble characteristics, whereas the amphiphilic polymer Pluronic (a PEG-polypropylene glycol (PPG)-PEG triblock polymer) can only absorb water to a limited extent because of its hydrophobic nature [555]. The degree of swelling of hydrogels also reduces as the level of crosslinking rises; whereas the crosslinking level can be controlled to regulate swelling, the rheological characteristics of the hydrogel are influenced at the same time [556]. For heterogeneous bioprints with several different material types or intricate geometries, limited spatial or non-isotropic swelling can lead to significant geometric variations after printing. These temporal fluctuations in the biologically printed structure can be used to create what are known as “4D” bioprinted materials [557]. For instance, a simple 3D-printed structure can transform into a more complex structure over time [558]. The principle of printed active composites (PAC) has been established, which can transform a printed film into a complex structure using the shape memory effect [559]. Global research into 4D printing has grown exponentially. There are many definitions of the 4D printing process. An early definition claimed that 4D printing is just 3D printing over time [558,560,561]. The definition that clearly specifies 4D printing, nevertheless, states that 4D printing is the evolution of the shape, properties and performance of a 3D-printed structure over time as it is exposed to heat [562,563], light [564,565], magnetic field [566], pH [567], water [561,568] and other similar things. Another definition says that 4D printing is the creation of scaffolds that alter their form as they leave the 3D printer. These objects self-assemble when they encounter water, heat, air, etc., due to the chemical reaction of the material used. Four-dimensional printing is a combination of a 3D printer, a smart material and a correctly programmed arrangement [569,570]. In 4D printing, various metamaterial structures are formed when the environment undergoes a change. Currently, most of the research in 4D printing technology focuses on the ability of 4D-printed materials to alter their form, such as by stretching, bending, curling and twisting. In biological and medical fields, the 4D-printed structures can alter their physical/chemical properties like stiffness or density. In addition, they display various phenomena, including shape memory effects and shape transformation [571], which, in turn, may impact cellular functions. The shape memory effect is a mechanism by which a system/structure can remember a particular shape and shift from one shape to another; for example, from the original shape to a programmed shape, in a planned way facing external cues. Shape shifting is a natural process in which a system/structure can change its appearance from one shape to another due to external influences. Compared to 3D printing, 4D printing offers several advantages, such as rapid growth of smart and multi-material materials, more flexible and malleable patterns and more applications for 4D or 3D printing. In addition, 4D printing offers higher efficiency, quality and performance compared to conventional techniques, as the 4D-printed structures can self-improve their characteristics and performance.

### 8.1. Key Drivers of 4D Printing

Four-dimensional printing is based on primarily five parameters. All these five parameters need to be accounted during the process of 4D printing. These five parameters comprise the AM process, the printing material, the stimuli, the mode of interaction and the type of modeling [570]. The first issue is the AM technique employed for printing. The AM procedure permits the manufacture of print media from the digital data received by the computer without the need for an intermediate device [550]. Several AM techniques exist like SLA, 3D printing with nozzles (3DP), selective laser melting (SLM), selective laser sintering (SLS), fused deposition modeling (FDM), direct ink writing (DIW), electron beam melting (EBM) and similar, and virtually all of these can print 4D media, provided the material being printed is suitable for the type of printer [572]. The next element is the material utilized for the print, which needs to react to the impulses as it is deposited layer by layer. These materials are also referred to as programmable or smart materials [282]. The nature of these smart materials dictates the type of stimuli to be applied, and the reaction of these materials to the various stimuli defines the material’s capacity for self-transformation. The third factor is the type of stimuli involved in 4D printing. The stimuli employed may be physical, chemical or biological in nature [571]. Physical stimuli comprise light, humidity, magnetic and electrical energy, temperature and UV light. Chemical stimuli encompass chemicals, the pH value, the utilization of oxidizing substances and reducing media. Biological stimuli involve enzymes and glucose [573]. When the stimulus is implemented, physical or chemical modifications like relaxation of the stress, movement of the molecules and phase modifications in the network are induced, which cause the structure to distort. The fourth and fifth features are the mode of interaction and its mathematical modeling [550]. When an intelligent material is subjected to a stimulus, not all materials can experience the intended alteration. An interplay mechanism like mechanical stress or physical force must be supplied to design the sequence of form modification. Following the provision of the interaction mechanism, mathematical modeling is necessary to schedule the time, during which the stimulus affects the smart matter [550].

### 8.2. General 4D Printing Laws

Three laws of 4D printing have been defined that determine the form-changing properties of all 4D-printed textures [574]. These laws allow a better comprehension of the physics underlying the shape-alteration capability of 4D-printed patterns. They are formulated as follows: The first law says that all form-alteration phenomena like the winding, rolling, twisting, bending, etc., of 4D composite patterns are caused by the mutual extension between active and passive components [575]. The second law says that four physical drivers are responsible for the capacity of all multi-material 4D patterns to transform their form: diffusion of mass, thermal dilation, molecular transformation and organic outgrowth [576]. All these features result in a mutual stretching between active and passive substances, which causes a modification of the form when a stimulus is applied. The absorption or adsorption of irritants, such as water or ions, leads to a modification of the mass of the network. This transportation of material ultimately results in a relative extension of the material and therefore a distortion of the form. The alteration in mass can also be caused through thermal, electrical, chemical, or light cues [550]. Thermal expansion can lead to distortions in the physical structure, as the average spacing between atoms and molecules grows or shrinks with rising or falling temperatures, leading to a relative dilation [577]. Thermal extension can also arise when electrical, light and UV signals are applied, as these may modify the temperature of the texture. In structures where the mass and temperature are unchanged, there may be relative expansion through molecular conversion. In these instances, electrical fields, magnetic fields, light or mechanical forces can affect a molecular conversion [576]. For instance, when an electric or magnetic field is imposed, the dipoles in the substance orient themselves in the direction of the imposed field, leading to a conversion of the molecules. Likewise, applying mechanical force to polymers forces the polymer chains to orient in a particular orientation, and irradiating a photosensitive material with UV light causes it to convert from trans to cis [578]. Organic growth refers to the gain in length and weight of a living organism within a specific time frame. The increase in weight and elongation causes a mutual extension of active and passive substances, which leads to form-modifying characteristics. In living organisms, organic growth can usually be induced through electrical impulses [579]. However, in addition heat, water, pH level and mechanical stress can also be utilized. Organic growth is employed to characterize the deformation response of cells, scaffolds, tissues, organs and stents, designed using 4D bioprinting [579] Organic growth is employed to characterize the deformation response of cells, scaffolds, tissues, organs and stents, designed using 4D bioprinting [550]. The third law of 4D printing says that “the temporal deformation characteristics of nearly all multi-material 4D-printed patterns are governed by two kinds of temporal constants”. These constants may be the same or vanish based on the type of input stimulus and what material is utilized for 4D printing [580].

### 8.3. Material Types Used in 4D Printing and Generation of Tissue-like Constructs and Organoids

In 3D printing, latest advances have allowed materials to be positioned more accurately and with more flexibility, which has greatly benefited 4D printing [581]. The materials employed for 4D printing are usually known as smart materials, because they can alter their characteristics as time passes (Figure 7) [582]. These materials can react to outside impulses and have characteristics including self-organization, self-repair, form retention and self-sustainability [213]. In addition, 4D printing involves not simply materials that can transform their form but also undergo color alterations upon exposure to UV or visible light [574].

#### 8.3.1. Materials React to Moisture: Hydrogels

Materials that react to moisture or water have gained a lot of interest because of their wide array of applications. These materials are also referred to as hydrogels, as they have an exceptional capacity to respond to water or humidity. These are a class of 3D polymer chain meshes that are formed by networking and can extend by up to 200% of their initial volume after encountering humidity. Hydrogels also have a high compressive strength, as different textures have been designed with hydrogels that can be wrinkled, flexed, stretched and geometrically expanded. These are extremely biocompatible and simple to print on when they are written with direct ink [583]. The only issue is their slowly reversing response, which means that drying and shrinking takes several hours. Overcoming this problem depends on programming the hydrogels so that their swelling is enhanced by anisotropy. Cellulose fibrils have been paired with the hydrogen ink and these have been oriented through the evolution of shear forces generated by the physical contact of the print plate and the hydrogel ink [568]. This orientation made the transverse swelling four times larger than the longitudinal swelling, which enabled the programming of the printed 4D texture. Another possibility is to limit the hydrogels in a single direction with rigid materials, causing an anisotropic expansion of the hydrogel [581]. Films of stearoyl ester (CSE) cellulose have been prepared, and these hydrophobic films exhibited a more accurate and rapid reaction than the previous films [584]. Usually, hydrogels are added to water and take up the water until their saturation level is achieved. The problem with this mechanism, however, is that it restricts the capacity of hydrogels for intermediate regulation. This problem can be solved through regulating the temperature of the aqueous medium. This was seen when producing the microgrip compound utilizing (poly-N-isopropylacrylamide-co-acrylic acid) pNIPAM-AAc hydrogels [585]. Reverse actuation is possible when the temperature of the water in which the hook has been dipped is altered. The 4D material has also been printed with alginate/pNIPAM ICE gel inks [586]. The use of hinge structures to avoid undue swelling has been proven. A self-pleating pattern has been manufactured from PolyJet printers that ceases to pleat at specific angles that were pre-programmed to prevent over-swelling [558].

#### 8.3.2. Materials React to Temperature: Thermo-Responsive

These objects are smart materials that react to heat or temperature signals. The variations in the form of these materials in response to thermal impulses are primarily attributable to two mechanisms, such as the shape change effect (SCE) [587] or the shape memory effect (SME) [588]. SME refers to the transformation of a shaped (plastic) material into its initial form through external stimulation [589]. Smart materials displaying the SME effect are referred to as shape memory materials (SMM) and are categorized as shape memory alloys (SMA), shape memory ceramics (SMC), shape memory hybrids (SMH), shape memory gels (SMG) and shape memory polymers (SMP) [590]. SMMs are classified into one-way, two-way and three-way (or multiway) materials based on the degree of shape transitions (Figure 8).

With one-way SMMs, the original form cannot be restored following deformation, while with two-way and three-way SMMs, the initial form can be returned to a temporary form after deformation through an intermediate shape [591]. SMMs can exhibit SCE alongside with SME, according to the ambient requirements. Among the different types of SMM, the SMPs are frequently employed as they can be printed with ease. The SMPs can regain their initial form after being deformed with the appropriate irritant [592]. The SMPs feature a typical glass transition temperature (Tg), generally exceeding the temperature where they are normally worked. Beyond Tg and under certain thermal and mechanical constraints, SMPs undergo programming and, as they cool, take on a transient shape that is devoid of any outside stress. When the temperature is increased beyond Tg once again, they take on their initial shape [593]. Below the Tg, the internal energy of the polymer chains is minimal, and they are unable to move freely, resulting in the material becoming glassy and stiff [594]. Above the Tg, however, energy is supplied to the polymer chains allowing them to move, resulting in the material appearing like rubber and prone to distortion and tampering [595,596]. An SMP sphere has been produced with the SLA printing process, and the sphere could switch between a flat plane and its initial form with high sustainability [597]. It has been demonstrated that an SMP may be pre-programmed in FDM printers through heat [593,598]. The SMPs have been adapted to utilize their distinctive characteristics for printing applications, that is, thermoset and thermoadapted SMPs [599,600,601,602,603]. The SMPs exhibit two or three interim transitions, and it is even possible to keep an intermediate state, which is equally stable. Another SMM commonly utilized in 4D printing is SMA, which is capable of altering its shape depending on the temperature fluctuation [604]. SMAs exhibit a typical temperature, referred to as the transformation end temperature, beyond which they display a high yield strength and a high Young’s modulus, which means that they are superelastic when above this temperature [604]. Nitinol (nickel-titanium) has been found to be the widely utilized SMA because of its favorable SME characteristics, its high ductility and toughness, and its robust cyclic characteristics, which renders it to be more biocompatible and amenable to actuation [605,606,607]. A Ni-Mn-Ga-based SMA has been examined for printing 4D components with the binder jetting technology, where the printed components displayed a reversed form alteration upon cooling and heating [608]. Cu-based SMAs have also been investigated for 4D printing because they can efficiently endure post-printing operations and are inexpensive. Although they are not as favored as SMAs because of their limited ductility, several pieces of research have been carried out involving these Cu-based SMAs [609,610,611]. Another type of SMA on the basis of iron (Fe) is also being explored for 4D printing utilizing the SLM printing method. They are inexpensive and feature pseudoelastic elongations [612]. SMAs have been mainly studied for their use in the biomedical sector, including surgery, orthodontics and physiotherapy [613]. The key restriction to the use of SMAs in 4D printing lies in their high expense. In addition, SMPs are lighter, more elastic, biocompatible and require lower energy consumption compared to SMAs [614,615]. To address the constraints of SMP and SMA, shape memory composites (SMC) have been designed through the integration of SMP with SMA or SMP along with a strengthening fiber [616]. The strengthening fiber may be a long or short fiber, nanoparticle or nanofiber with elevated mechanical characteristics and high deformability [616]. In addition, SMHs represent the intelligent materials created by the combination of SMAs, SMPs and hydrogels. These can react to temperature, pressure and a number of different inputs simultaneously. In SCE materials, the deformation is linear to the exerted stimulus or the deformation can be characterized as changing from one extreme state to another [587]. When the material reacts thermally, SCE occurs in the two-layer constructions. The structure flexes when the load is applied, but the interface between the layers stays the same. A graphene-based network has been produced and demonstrated to transform into a flat plate when heated and regain its initial cylindrical form upon cooling [617]. Such drastic alterations in form necessitate a strong variation in temperature.

#### 8.3.3. Materials React to Light: Photo-Responsive

Light also works as an implicit impulse for the distortion of smart materials. When a region of a smart material that interacts with light, which is referred to as a photosensitive material, is subjected to light, it absorbs the light, causing the material to heat up. Heat acts as a form of impulse for the shaping of intelligent materials, resulting in a form alteration of the photosensitive material. A sequential self-folding structure has been demonstrated in which the light is absorbed by the joints, and these are subsequently heated, causing a transformation in form [618]. The speed at which the heat is taken up by the joints varies according to the light source applied and the joint color. The light can be employed in a different manner to evoke a deformation within the photo-responsive substance. A light-sensitive chromophore can be incorporated in certain areas of a polymer block (gel) so that just these areas are distorted when light falls on the pattern [564]. Additional work demonstrates that UV light (weak) and visible light have been applied to distort the 4D architecture [619].

#### 8.3.4. Materials React to Electric Energy: Electro-Responsive

Electricity acts as an indirect impulse in the same manner as light, as it has been demonstrated to induce heating because of the resistance of the substance it passes through. For this reason, materials that deform because of their reaction to electric current are referred to as electro-sensitive materials. An engineered muscle has been manufactured from a blend of ethanol and silicone elastomer. Current conducted across the muscle leads to the vaporization of ethanol, thereby expanding its volume, which inevitably deforms or stretches the muscle [620]. The absorption or desorption of water in polypyrrole (PPy) can be guided through electricity, and this concept has been utilized to fabricate microrobots (origami) of PPy. The absorption of moisture when the robot has been held in a humid surrounding caused a tension that propels the head of the robot to the front. The tail of the robot trails the head when desorption takes place and there is no tension [621].

#### 8.3.5. Materials React to Magnetic Energy: Magneto-Responsive

The magnetic field or magnetic energy acts as an indirect impulse that can induce distortion in smart materials. The materials employed for printing 4D structures based on their distortion reaction to magnetic energy are termed magneto-responsive materials. By using magnetic nanoparticles in microgrippers that were printed from hydrogel, a state of remote controllability has been introduced. When a magnetic field has been placed on the printed pattern, it started to display a response that can be managed remotely [585]. The 4D-printed structures utilizing magneto-responsive media have enormous potential in the area of metal and polymer printing, with the sole disadvantage that the print size needs to have a low mass so that it can be influenced through the magnetic field. Magneto-active 4D printing is based on magnetic field induced systems. The printing materials, such as resin, powder or filament need to contain magnetic field sensitive features/components, which are generally fillers, that are activated through an outside magnetic field to exhibit the 4D phenomenon. The first key stage in the creation of 4D structures with 3DP is therefore the process of adapting the printing materials through the inclusion of active elements. The most frequently deployed magnetoactive filling materials are carbonyl iron powders (CIPs), iron(II, III) oxides and Fe-Nd-B micro/nanoparticles [622,623,624,625,626]. Not all these magnetic fillers can still be applied in all 4D printing processes which is mainly because of the dimensions of the fillers. Nevertheless, there are alternative printing processes, like DIW, in which nanometer-sized to micron-sized filling materials have been employed with great efficiency [627,628,629,630,631]. The fillers have been designed to improve the ability to re-extrude compound filaments for FDM [632,633], the manufacture of composite or surface-decorated (with nanofillers) micropowders for SLS [634] and high strength in fluid resin for SLA/DLP [635,636,637]. Pure SLA is amended to its derivatives, such as direct laser processing (DLP), micro-continuous liquid interface production (μCLIP) or two-photon polymerization (2PP). However, all variations are still reliant on the light-driven transition of liquid resin [135,638,639,640].

#### 8.3.6. Piezoelectric Materials Distort upon Force

Piezoelectric materials are employed in 4D printing purposes as they are able to distort under the impact of a mechanical force [641]. As a type of smart material, piezoelectric materials can create an electrical current when they are exposed to mechanical stress that inevitably cause alterations in the structure, because the charge can induce deformation. These piezoelectric materials can therefore alter their geometry when an electric current is supplied to their surface. These materials consist of crystals or ceramics that exhibit a certain crystal structure, like quartz or barium titanate. The effect of generating electrical charges due to the action of a mechanical force is termed piezoelectricity. The piezoelectric effect is due to the orientation of the crystal structure, which creates an electrical charge while the material is distorted [642].

#### 8.3.7. Materials React to pH

These are intelligent materials that react to the pH value and can alter their form and volume depending on the pH level [643,644]. The form modification in reaction to various pH values renders them appropriate for 4D printing applications. Polymers that react to pH have been utilized for 4D printing, like polyelectrolytes, as they can absorb or release protons when the pH is altered due to an ionizable side chain. Upon the liberation of a proton, the polymer string expands due to electrostatic repulsion, leading to a deformation of the network, and when a proton is absorbed, the network neutralizes. Polyelectrolytes include polycations or polybases like ammonium salt as a functional chain and polyanions or polyacids, such as carboxyl or sulfone moieties as ionizable side groups. The side chains donate the proton at higher pH values (stretching) and absorb the proton at lower pH values (neutralizing). The functional group, however, donates the proton at lower pH values and absorbs the proton at higher pH conditions [645,646,647,648,649]. pH-responsive materials can be used in drug administration [647,650], soft robots, actuators [643], valving, biocatalysts and in stabilizing of colloids [567,648]. Smart materials can react to a single or two kinds of inputs. For example, SMPs can be made to interact with temperature, light and electrical energy, while compound composites can react to various irritants.

#### 8.3.8. Maturation of Hydrogels via Physical Stimuli Toward Tissue-like Constructs and Organoids

The maturation of hydrogels and cells into tissue-like assemblies has also been speeded up by the introduction of physiologically important external factors. In the tissue engineering of cartilage tissue, the mechanical pacing of structures under pressure enhances ECM production and encourages cartilage cell diversification [651]. In vitro models of cardiac tissue take advantage of electrical pacing that encourages differentiation along a cardiomyocyte lineage and helps synchronize the heartbeat [652,653]. It is anticipated that continued evolution of bioink materials that facilitate these types of post-print extrinsic inputs will result in more “mature” engineered tissues. For instance, the utilization of injectable conductive hydrogels as bioinks in combination with the use of external electrical stimuli following printing may provide a bespoke approach for cells that need electrical signaling for proper performance. By taking into account the physiological factors specific to the target tissue type, bioprinted designs can ripen to get more similar to natural tissue.

## 9. Cell Alignment in Printed Scaffolds

The alignment of cells is critical for cellular functions, such as the generation of forces and cellular motility, like cancer cell invasion, or developmental processes, such as organ formation. Moreover, the functionality of organs relies on alignment-driven functions. For instance, the electrical and mechanical characteristics of the heart [654] and the multinucleation of muscle fibers during the creation of myotubes [655] need, for musculoskeletal tissue, a high amount of cellular alignment to carry out key cellular roles. In addition, aligned cells inside a highly organized, anisotropic ECM set off a series of events that are crucial for defining the function of the tissue [656,657]. The alignment of cells appears to contribute to several cellular behaviors, including the reorganization of the cytoskeleton, nuclear gene expression, and rearrangement of the ECM scaffold. The alignment and elongation of cells in the direction of anisotropic and aligned topographies are important phenomena of cellular contact guidance and are seen in multiple cell types. Hence, a question arises whether there exists a universal mechanism behind cell alignment [658]. The most commonly acknowledged model of cell alignment is topographically-driven orientation, which proposes that anisotropic topographies constrain the growth of focal adhesions and actin stress fibers laterally, thereby promoting anisotropic force generation, cellular stretching and alignment. There are certain circumstances in which alternative or complementary mechanisms of cell alignment seem to come into effect. These examples involve the cases of certain cell types, like amoeboid cells and neurons, and particular topographies. Moreover, the actin cytoskeleton is involved in regulating topographically based cell alignment, highlighting the importance of elucidating the contribution of other cytoskeletal components. The understanding of cell alignment is critical for identifying the function of cellular contact guidance in healthy and diseased conditions.

Support-assisted bioprinting is employed to bioprint a hydrogel with embedded C2C12 cells into a lattice structure [659]. The purpose of employing a lattice structure is, firstly, to assess the structural accuracy of guided bioprinting. Secondly, the lattice pattern is utilized to imitate the angular variations of fibers in various layers. In this approach, a secondary material is incorporated as a mold for the encapsulation of a primary material. The secondary material can be fully taken out. The bioink consists of 10%*w*/*v* GelMA and 2%*w*/*v* alginate [659]. The purpose of support bioprinting is to ensure the structural rigor of the primary material before the ultimate crosslinking of the bioprinted structure. It can be assumed that the cell alignment parallel to the printed hydrogel rods of the lattice structure can be realized. Extrusion-based bioprinting can be utilized to control cell orientation with the help of a predefined extrusion path. This allows the angle of cell alignment to be specified on various layers, in a similar way to native myocardial features. This novel bioprinting strategy has been shown to achieve macroscale alignment of cells, resulting in an 80% alignment of cells falling within a 15° orientation. Moreover, the printing strategy demonstrated regulated cell alignment corresponding to the different angle modifications on different planar levels.

The orientation of the cells inside the printed substrate can be accomplished through magnetic tagging of MSCs and HUVECs. The magnetization of cells using standard fluorescent MNPs from Chemicell (100 nm) has been carried out following a conventional approach. HUVECs have been magnetized with nano-screen MAG/R-PAA nanoparticles bearing a red fluorescent tag, whereas MSCs have been magnetized with nano-screen MAG/G-PAA nanoparticles bearing a green fluorescent tag. The use of a magnetic framework can orient them to imitate the vascularization of osseous scaffolds [660]. A new, multi-purpose and user-friendly approach has been developed to promote controlled 3D sowing of cells by magnetic guidance. The simple pulling of cells charged with magnetic nanoparticles across an external magnetic field has previously been published and resulted in solutions that are beyond the scope of any other technology [660,661,662,663,664,665]. Specially engineered magnetic frameworks are employed that are featured by strong magnetic gradients on a short scale (100–200 μm) that can align and capture the magnetized cells on the selected face of the framework fibers. Such local magnetic structuring constitutes a practical way to build 3D cell structures with a controlled structure at the microscale. As principal proof of this exceptional capability, a well-defined separation of two cell populations, specifically MSCs and HUVECs, has been achieved on the confronting sides of the magnetic osteogenic framework fibers. This cell composition is anticipated to support the bone microarchitecture restoration with suitable characteristics, especially with regard to the vascularization of the artificial bone [666,667]. The cells have been magnetically tagged with MNPs, whereas the frameworks have been modeled and manufactured on the basis of sophisticated magnetic materials by blending bioresorbable Fe-doped hydroxyapatite (FeHA) with PCL [668]. In concrete terms, the 3D frameworks have been created through injection/extrusion and laying down the fibers in specific orientations in line with the specified laying template [669,670]. The nanocomposite pellets have been exposed to a temperature of in the range of 110 to 130 °C in a cartridge unit attached to the flexible arm of a 3D bioprinter. The magnetic force exerted on the cells is a function of their magnetization and the regional gradient of the magnetic field [671]. To create analogous conditions for cell handling and adhesion to the scaffold, it is essential to obtain analogous magnetically actuated forces for most cells. Performing standard magnetic mapping of magnetized cells is a difficult challenge, as it is hard to keep cell-friendly environments in magnetometers or susceptometers. Therefore, an assay focusing on cell locomotion in an imposed magnetic field has been established. Cell motility in reaction to magnetic guidance has initially been assessed in a qualitative manner by utilizing a cylindrical NdFeB permanent magnet exhibiting 1.2 T magnetic remanence. The magnet has been positioned under the base of the cultivation plate and almost all cells have been magnetically captured, with a marginal quantity of cells staying outside the magnetic field.

Another method of distributing the bioprinted cells at the micro-architecture level is an approach using the acoustophoresis principle. The physics of ultrasound-assisted bioprinting (UAB), exploiting the principle of acoustophoresis can be exploited to orient MG63 cells inside single- and multilayer after bioprinting of alginate constructs using extrusion [663]. The cells have been oriented orthogonally and parallel to the printed filaments, thereby imitating cellular anisotropy in tissues like ligaments, tendons and cardiac muscle. In a similar manner, an acoustic excitation mechanism has been utilized to direct skeletal myoblast cells (C2C12) and HUVECs embedded in GelMA bioink [664].

Aligned cells exhibit directionally sensitive mechanical characteristics that affect biological and mechanical functionality in native tissues. Conventional alignment techniques like casting and uniaxial stretching are unable to completely mimic the intricate fiber alignment of native tissues in organs such as the heart. Bioprinting is utilized to guide cell orientation. A 0°–90° lattice pattern was 3D-printed to determine the strength of the supported bioprinting technique [659]. Changing the angles of the lattice pattern is intended to replicate the variations in fibril alignment in native tissue, where the angles of cell alignment change through the various layers. When a cell-hydrogel blend was bioprinted, C2C12 cells showed an alignment in the direction of the printed beams. Cell alignment is accomplished by inducing structurally stable structures, such as various 0°–90° structures, and by permitting cells to dynamically reshape the bioprinted structure. A heterogeneous scaffold has been developed [672], that can be adjusted in terms of gradient strength, adjustable fiber diameter and pore size. Heterogeneous scaffolds containing ultrafine fibers, like 3–22 µm in diameter, can be printed using high-resolution melt electrowriting (MEW) with single-nozzle printing by adapting the printing conditions. The diameter of the printed fiber is similar to that of cells (tenths of a micrometer), less than that of traditional 3D-printed scaffolds (100 µm or more) and cell adhesion is very susceptible to fluctuations in fiber diameter. By precisely positioning thick and thin fibers, cells can be made to stretch in a grid in the expected direction. For example, cell alignment in a singular scaffold with four regions displayed four different alignments, all separately. The scaffolds may also be non-uniform in pore size, whereby the proliferation velocity of the cells in small pores is three times elevated compared to that of the cells within large pores. In addition, different pore sizes and fibers can be incorporated into a scaffold, enabling cell growth to be guided to various stages by tailoring the scaffold architectures. This approach usually constitutes a method for structure-induced cell growth to better imitate the in vivo microenvironment.

Conventional electrospinning has been widely used to produce tissue-engineering scaffolds [673,674]. Due to its ultrafine continuous fibers, high surface-to-volume ratio and high porosity, the electrospun scaffold has morphological resemblances to the natural ECM [675,676]. Nevertheless, the fibers are aligned randomly, which impedes efforts to regulate the scaffold structure created [676]. While some specialty collectors, such as U-collectors, parallel plate collectors and rotating tumble drums, are designed to maintain aligned fibers, it is challenging to manufacture free-flowing structures [677,678,679,680]. In another application, 3D printing has been used to create scalable scaffolds for tissue engineering, where the size is typically between hundreds of micrometers and millimeters. The advantages of traditional electrospinning and 3D printing have been coupled to create electrohydrodynamic (EHD) printing [672], which facilitates the manufacture of porous scaffolds with precisely arranged ultrafine fibers [681,682]. The technique has also been applied to the production of pliable electrodes [683,684,685]. In comparison to conventional electrospinning, electrical instabilities, also referred to as the “whip effect”, are avoided by decreasing the distance between the nozzle and collector during EHD printing [686]. By applying a relative motion between the nozzle and the collector, EHD direct writing enables high-resolution patterning of the micro/nanofiber [687,688]. Melt-driven EHD is a physical and environmentally safe printing technique that does not involve organic solvents and avoids the restrictions imposed by toxic residues and their build-up [689]. The technique is also better at producing scaffolds for clinical tissue healing. There are even other techniques, such as dielectrophoretic-based bioprinting and rotary jet spinning. For example, the fabricating of a dielectrophoretic microfluidic device has been described that employed 3D-printed molds and silver conductive paint [690]. The latter, rotary jet spinning technique has been employed to produce ECM scaffolds and seems to be suitable for combination with bioprinting. This approach has been seen to decrease the amount of bacterial contamination of printed scaffolds [691].

Finally, bioprinting is emerging as an important manufacturing tool for tissue engineering by controlling cell orientation through the design of the printing path of material laydown. Spatial orientation of 3D-printed scaffolds, such as blended gelatin-sodium alginate 3D-printed scaffolds, regulates the gene expression profile of pre-osteoblasts [692]. Additionally, 3D-printed graphene-PLA scaffolds enhance the cell orientation of iPSC, neuronal cells, immortalized fibroblasts and myoblasts and enhance their differentiation [693]. In addition to orienting cells based on topological cues, graphene is instrumental in driving cell differentiation, as evidenced by iPSC compulsion to neuroectoderm and the merging of myoblasts into multinucleated myotubes accomplished through the 100 µm graphene scaffolds [693]. This research demonstrates the creation of a robust and economical 3D-printed scaffold with the capability of being employed in multiple tissue engineering applications and reveals how the scaffold’s microtopography and the characteristics of graphene synergistically guide cell differentiation.

## 10. Conclusions, Open Questions and Future Directions

The synergistic use of bioprinting, scaffolds, organoids, organ-on-a-chip and advanced biomaterials opens a completely new branch of cancer research with the aim to create tumor models that more faithfully reproduce the TME. However, the use of specific organotypic tumor models for the purpose of mechanobiological analysis and ultimately the study of disease mechanisms from a biophysical perspective still faces major hurdles. Thus far, the mechanical analyses of cancer cells and their environment such as the ECM scaffold and adjacent cells have been performed in cell culture systems mainly derived from 2D cell cultures, or even in a capillary system without significant adhesion of the cells, which surely does not correspond to the natural environment of these cells in tissues. There are still some key questions that remain open, such as the following. What efforts can be made to increase the efficiency of growing personalized tumor organoids and generate patient-specific tumor models for each cancer subtype in the entire patient population, incorporating the structural and mechanical features within tumors or in tumor niches during malignant progression? What are the potential advantages for mechanical studies of single cells within 3D organoids, which are embedded in natural or synthetic or hybrid biomaterials, when using 4D bioprinting? Could these models be used to investigate how these cells respond to stimuli for post-printing modification of in vitro tumor models? Would it be possible to analyze the temporal course of stimulation? How can 3D bioprinted or 4D bioprinted tumor models help to improve the prognosis and diagnosis of tumors and their malignant progression, taking into account mechanobiological aspects? How can a combination of clinical imaging and genetic analysis be used to identify predictive biomarkers and/or mechanomarkers for tumor progression and determine specific biochemical and physical mechanisms involved in the therapeutic response to heterogeneous medications? Overall, the analysis of mechanical aspects in 3D-printed organoids with vascularization is crucial for the accurate analysis of mechanical drivers or mechanical markers for the malignant progression of cancer. There is great potential to identify general mechanomarkers of cancer progression that are not limited to a specific cancer type. In addition, there is a unique opportunity to develop mechanically heterogeneous 3D organoids of cancer cells, which may also contain immune cells, such as macrophages, or lymphocytes, stromal cells and a specific architecture of the vasculature.

### 10.1. Patient-Specific Modeling of Tumors

Current cancer models largely rely on tumor spheroids derived from immortalized cell lines because of their ease of proliferation and capacity to withstand intricate fabrication procedures. Nevertheless, cell lines are prone to accumulate genomic aberrations and cannot replicate the heterogeneity of tumors in individuals, leading to divergent pharmacological reactions. Patient-derived cell collections are preferable for use in precision medicine and personalized drug testing. They nonetheless encounter the challenges of limited specimen resources, poor cell viability during transportation and low organoid retrieval efficiency. For example, the establishment level is under 30% for certain subtypes of cancers [694,695]. Hence, there is an immediate demand for improved primary cancer cell collection methodologies, cryopreservation techniques and enhanced culturing techniques to guarantee the effectiveness of organoid collection in the entire patient population and across the full range of cancer subtypes [696,697]. The pairing of tumor organoids and stromal cells from the identical patient could aid the identification of more potent and less harmful therapeutics, the optimization of dosing schemes and the development of suitable routes of administration for the individual patient [698].

### 10.2. Dynamic Post-Printing Alterations

A challenging aspect of biofabrication is 4D printing, which introduces time as a fourth dimension and permits materials or components of living cells to alter their shape or behavior according to stimuli. Tumor modeling can capitalize on this additional modification after printing, as it can mimic the dynamic characteristics of natural tumors, including tumor invasion and ECM restructuring resulting from matrix deposition and enzyme-dependent breakdown [699]. The development of fine-tunable biocompatible materials that react to outside irritants and the response of cells is indispensable for achieving this. Consequently, stimulus-dependent hydrogels that can remodel and modify their properties in reaction to external stimuli, such as temperature [700], light [701], moisture and enzymes, are needed to enable the downstream modification of tumor models.

### 10.3. Quality Control and Adjustment

The strategy of producing synergistically provides new possibilities for the fast pace, high throughput generation of advanced cancer models. Regulatory hurdles must be overcome, though, prior to the full-scale use of these integrative organotypic cancer models in clinical settings. Additional multidisciplinary work is needed to standardize and develop guidelines for biological issues, comprising biomaterial and medium preparation, specimen retrieval and operational guidelines. Real-time readout techniques incorporated into organ-on-a-chip systems that utilize machine learning-based image analysis can be devised to track tumor performance [702]. These techniques could enhance the process of quality assurance and improve efficiency, uniformity and replicability [703]. While these patient-specific tumor models are impressive with respect to the ethical principles of minimizing animal testing, their scaling and clinical translation continue to pose significant ethical difficulties [704]. Because of the fast pace of technological advances, it is not straightforward to foresee the potential future applications and banking of these models, which raises doubts about getting informed permission from donors [704]. Ethical principles for commercialization are also needed to guarantee a balanced sharing of benefits among donors/patients, scientists, commercial suppliers and other actors participating in the evolution of these models [1].

### 10.4. Predictive Integration at the Clinic

Building a 3D tumor model for individual patients in personalized medicine could be expensive and take time. A potentially more efficacious pathway could be the generation of large arrays of patient-derived tumor models that reflect certain cancer subtypes, which could aid in the identification of biomarkers that can predict therapeutic responsiveness for a particular patient group. The integration of clinical imaging [705] and genetic profiling [28] can lead to the identification of potential biological or transcriptomic pathways involved in heterogeneous drug responsiveness. In addition, reliable signatures of cancer patients that show a response to various therapies can be deduced to make a forecast for cancer treatment. Multi-organ chip screening of pharmaceuticals relevant to various co-morbidities is advantageous to enable specific assessment of cytotoxicity and susceptibility in diverse subpopulations [505]. While the convergence of bioprinting, tumor organoids and organ-on-a-chip techniques is currently in its fledgling stages, a promising pathway is starting to appear for the creation of integrative organotypic tumor models for enhanced therapeutic prognostic capability [463,500]. Ultimately, the bioprinting method could surpass animal models and open the door to quicker and more affordable drug discovery and pathology testing [1]. Nevertheless, multidisciplinary collaboration between clinicians, biologists, physicists and engineers is necessary to ensure that the bioengineering procedure for cancer models is harmonized, all ethical aspects are resolved and their use is fully implemented from the laboratory to the patient’s medical care facility [706].

The future gap of 3D-printed organoids of cancer cells is the localized and dynamic mechanical analysis of the cells within the organoid in a realistic 3D ECM microenvironment, wherein specific structural or mechanical cues can be altered in a functional manner. Additionally, 3D bioprinting provides the basis for mechanobiological analysis in a 3D environment, whereby cancer cells, neighboring other cell types and the ECM structure, mechanical properties and composition can be monitored. Consequently, this allows general mechanisms of cancer progression to be determined based on mechanical and structural features of the cancer environment in 3D advanced cell culture models.

**Table 1 cells-13-01638-t001:** Selected biomedical applications of organoids and assembloids using 3D bioprinting.

Usage	Bioprinting Technique	Cells	Advantages	Disadvantages/ Limitations	References
Nanotoxicological investigation Placement of organoids and assembloids	OBST	Non-small cell lung cancer line Calu-3 (Calu-3)	Long-term cultivation of cell lines based on decreased oxidative stress over time. Enhanced viability of 3D-printed cells once incorporated into the hydrogel. No loss of time due to cell passage. More efficient and cost-saving. Consideration of possible impacts on cell morphology and cell–cell interaction in a 3D environment. Dissimilarities between 2D and 3D data can be uncovered.	The number of cells sown is strictly determined by the 3D printing parameters and is usually lower than the number of cells imbedded in the hydrogel-loaded syringe due to remaining volume in the Luer lock of the syringe. Force applied during extrusion can mechanically rupture cells.	[408]
Insertion of physical limitations in organoids	Physical obstacles	Gastric cancer, breast cancer, prostate cancer and non-small cell lung cancer primary cells	Structuring of ECM mechanics in a localized manner and topographically patterned hydrogel scaffolds. Organoids may create functional vascular systems following transplantation. Organoids can be utilized to invert disease-causing mutation to treat disorders caused by mutations.	Limited cell numbers Limited resolution of mechanical obstacles	[381]
Hybrid bioprinting method for bioprinting cellular aggregates (i.e., tissue spheroids and honeycombs) and organoids with dimensions between 80 and 800 μm into or onto hydrogels for both scaffold-free and scaffold-based applications.	OBB	iPS cell-derived organoids	Large-scale organoid-like structures, or assembloids, can be realized by fast spatial arrangement. Heterogeneity of iPS cell-derived organoids is mimicked. Assembling organoid units of different shapes pre-divided into basic organ elements.	Limited size of organoids can be printed. Complicated spatial arrangement is difficult due to the simple form of the block, such as a sphere. Changes in bioink characteristics during the printing process may hurt organoids.	[404,707]
Spheroids can be moved through a shear-thinning hydrogel that self-heals to accommodate the spheroids and hold them in 3D space; also for targeted merging between spheroids to form micro-tissues with high cell density and well-defined shape, which can then be excised from the hydrogel. Heart disease model that imitates scar formation after myocardial infarction (MI) by bioprinting micro-tissues with spatially controlled density of two cell types. Spheroids are removed from the cell medium by back pressure and lifted into the air to be bioprinted into functional hydrogel, such as fibrin, collagen, GelMA or onto sacrificial hydrogels, such as alginate and agarose.	AAB	Induced pluripotent stem cell (iPSC)-derived cardiomyocytes and primary human cardiac fibroblasts (CFs) Human MSCs (hMSCs) were isolated from fresh unpro-cessed bone marrow from human donors	Custom forms for 3D bioprinting have been designed using computer-aided design (CAD) software. Design of personalized in vitro disease models that deliver comparable functional outcomes to preclinical animal models while maintaining ease of investigation, minimizing expense, and in formats that accommodate a wide variety of imaging and assessment techniques. Facilitate advances in the production of assembloids, where organoids from various tissues or tissue regions are merged in a regulated environment to investigate tissue development and maturation. Mimic pathological scarring characteristics that occur after myocardial infarction, and with the help of measurements of cardiac function (contraction, electrophysiological synchronization), miRNA therapeutics for reparation could be tested.	Simple model AAB is poorly suited for generation of neural assembloids, as neural organoids display large diameters, relatively weak surface tension, and a propensity to perform plastic deformation and break down under relatively low vacuum force.	[88,407,444,708]
Merging and placement of assembloids with internal cytoarchitecture. Consists of an iron-oxide nanoparticle laden hydrogel and magnetized 3D printer to enable the regulated lifting, transport, and deposition of organoids.	SPOT	human pluripotent stem cell-derived neural organoids and patient-derived glioma organoids	Improves OBB technique Generates accurately arranged assembloids composed of human pluripotent stem cell-derived neural organoids and patient-derived glioma organoids. Constructs assembloids that recapitulate major developmental processes and disease etiologies. Construction of neural assembloids in 3D. Construction of assembloids based on dorsal and ventral forebrain organoids.	Magnetic nanoparticle (MNP)-laden cellulose nanofiber (CNF) hydrogel that may possibly impact cell function.	[398]
Merging and placement of assembloids with internal cytoarchitecture.	Magnetic levitation		Implementation of internal structures in organoids. Mimics the in vivo organs and tissues more closely.	Relies on cellular internalization of a bioinorganic hydrogel comprising iron oxide. Magnetic particles alter cellular functionality in 3D.	[709]

SPOT = spatially patterned organoid transfer, OBB = organ building blocks, AAB = aspiration-assisted bioprinting.

## Figures and Tables

**Figure 1 cells-13-01638-f001:**
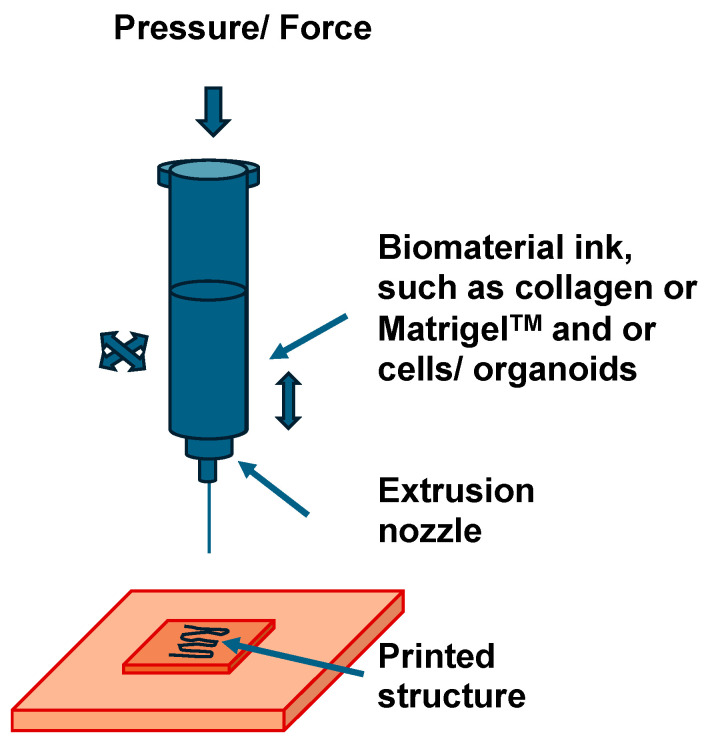
Schematic sketch of a pneumatic extrusion bioprinting device.

**Figure 2 cells-13-01638-f002:**
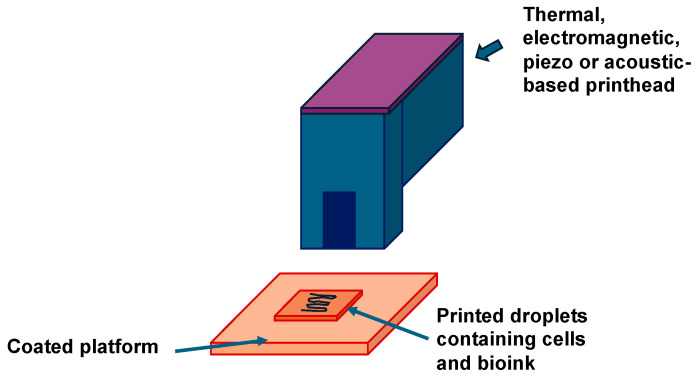
Schematic sketch of a inkjet bioprinting device.

**Figure 3 cells-13-01638-f003:**
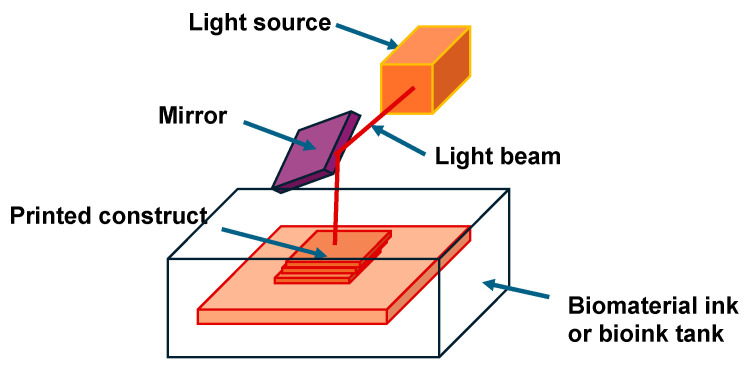
Schematic drawing of a SLA device.

**Figure 4 cells-13-01638-f004:**
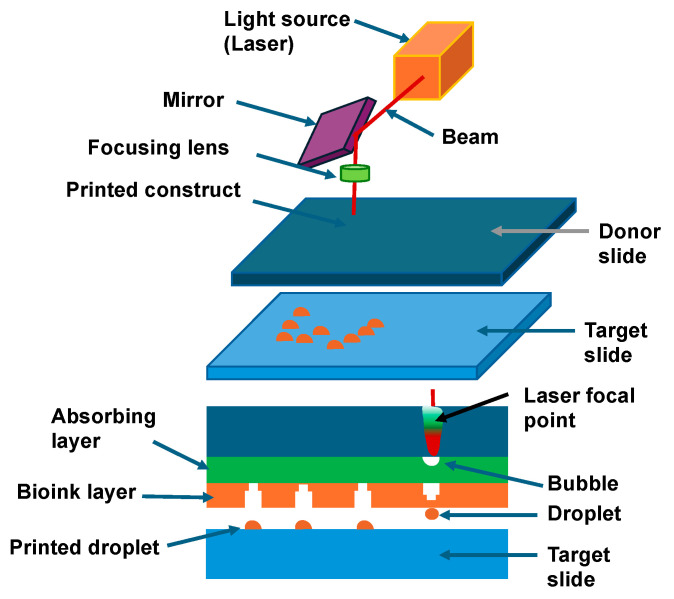
Schematic drawing of LIFT device.

**Figure 5 cells-13-01638-f005:**
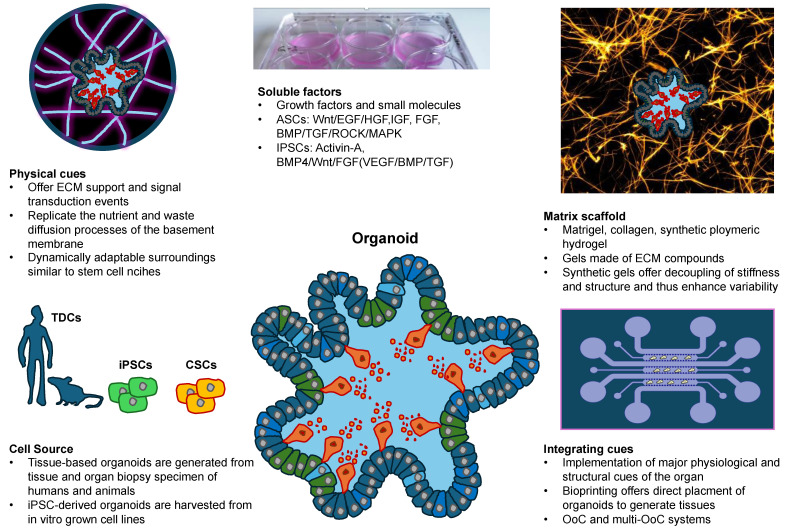
Elements of organoid cultivation technique. Setting up an organoid-based culture necessitates deliberations on the key elements that constitute organoid cultures, comprising cells, soluble factors and matrix, physical cues and the effective incorporation of these elements. ASCs = adult stem cells; CSCs = cancer stem cells; ECM = extracellular matrix; FGF = fibroblast growth factor; iPSCs = induced pluripotent stem cells; OoC = organ-on-a-chip; TDCs = tissue-derived cells.

**Figure 6 cells-13-01638-f006:**
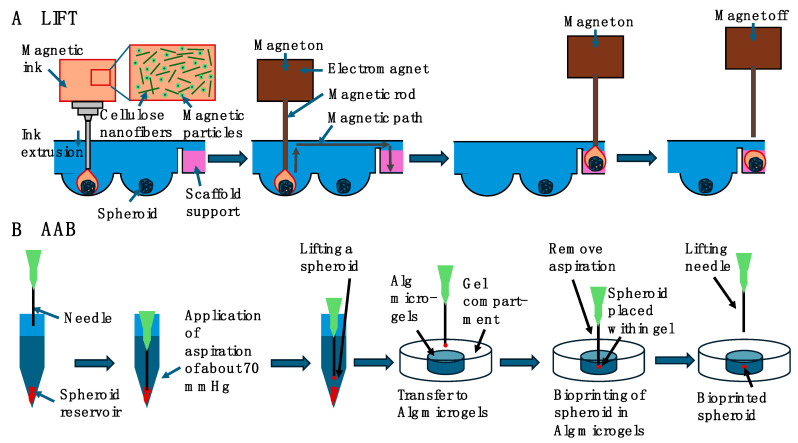
Schematic drawing of (**A**) LIFT and (**B**) AAB bioprinting techniques for spheroids. Alg = alignate.

**Figure 7 cells-13-01638-f007:**
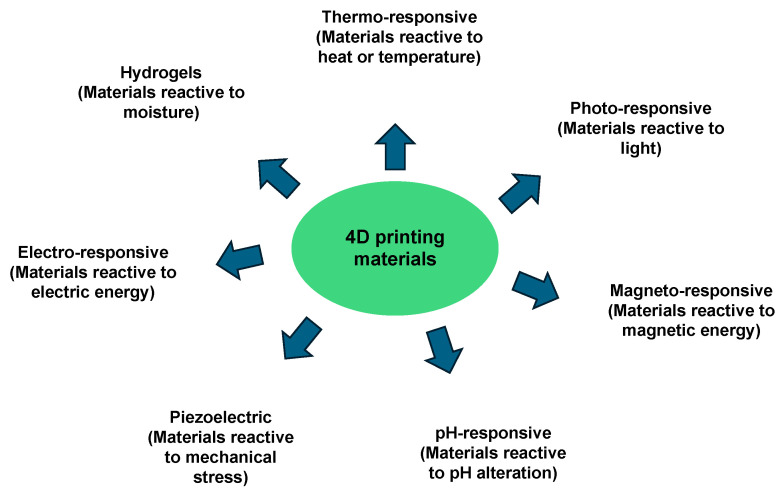
Material types employed for 4D bioprinting.

**Figure 8 cells-13-01638-f008:**
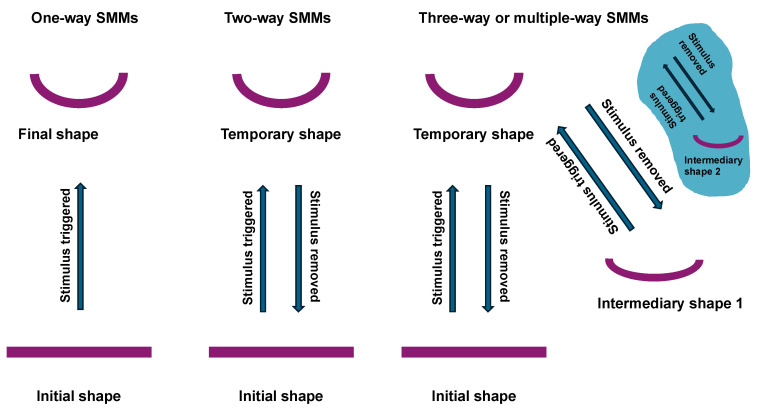
Shape memory materials (SMMs). A one-way SMMs cannot regain its initial shape after being deformed. In contrast, two-way, three-way and multiple-way SMMs can restore their initial shape. Two-way SMMs can switch between the two shapes. Three and multiple-way SMMs can return from a temporary shape to the initial shape via one intermediary shape or multiple intermediary shapes following deformation.

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
