# Peer review of "Bioprinting of Cells, Organoids and Organs-on-a-Chip Together with Hydrogels Improves Structural and Mechanical Cues"

_cells, 2024, doi:10.3390/cells13191638_

Round 1
Reviewer 1 Report
Comments and Suggestions for Authors
In the Review paper, the author aimed to summarize the different aspects of using hydrogels and bioinks for the design, printing and development of organoids and organ-on-a-chip. The review includes a comprehensive description of different manufacturing techniques, and the different organs.
While this review paper offers a wealth of relevant information and spans a wide range of research, its organization is inefficient, and the discussion lacks depth. As a result, several points are repeated throughout the text in similar phrasing. Since the main goal of a review paper is to present comprehensive data well-structured and engagingly, while providing meaningful discussion, this manuscript falls short in delivering on that front. Despite its promising title, the manuscript does not fully meet expectations. The manuscript required some changes before being considered for publication.
Here are some specific comments for improvement:
- Include a schematic diagram highlighting the major bioprinting techniques, briefly outlining the key properties of each type as discussed in section 2.
- In "Subsection 4.2: Benefits and limitations of organoid cultures," the limitations are not addressed as suggested by the title. Please include a discussion on this.
- Correct the subsection numbering for “3D bioprinting for organoid generation.”
- For better clarity, provide a schematic diagram illustrating the various 3D bioprinting platforms for organoid generation (e.g., SPOT, OBB, AAB) as discussed in the relevant sections.
- While the authors provide extensive information on techniques and their advantages and disadvantages, there are numerous instances where no relevant examples or references are given. For example, the "one-step manufacturing approach for organ-on-a-chip" is mentioned without citing a research article. Similarly, section 7 discusses 4D printing in detail but lacks examples.
- The definition of 4D printing is repeated several times in section 7. Furthermore, the statement “For instance, a simple 3D-printed structure can transform into a more complex structure over time” could be supported with examples or references.
- In section 8, elaborate on why cell alignment is important. Explain its significance in printed scaffolds and the specific conditions or tissue models that require cell alignment, supported with relevant examples.
- The authors mention two methods for cell alignment—magnetization and acoustophoresis. Other techniques, such as electrospinning-assisted bioprinting, dielectrophoretic-based bioprinting, and rotary jet spinning, could also be included.
- Despite a promising title, section 9 presents a rather limited discussion. There is little comparison between organoids and organ-on-chip in terms of their suitability for 3D bioprinting. The section is more focused on describing 3D printing techniques (e.g., SPOT, OBB).
- In section 10, the following sentences are unclear and do not lead to meaningful insight: (i) "What are the potential advantages for mechanical studies of single cells within 3D organoids using biomaterials, as represented in 4D models, that respond to stimuli for post-printing modification of in vitro tumor models?" (ii) "How can the implementation of 3D bioprinted tumor models be approached with special consideration of mechanobiological aspects in clinical applications?"
Author Response
Point-by-point answers to reviewers’ comments
I thank both reviewers for their excellent comments that really helped me to improve my review article. All changes are highlighted in yellow in the manuscript. For my detailed response to each comment see below.
Best regards
Claudia Tanja Mierke
Reviewer 1
In the Review paper, the author aimed to summarize the different aspects of using hydrogels and bioinks for the design, printing and development of organoids and organ-on-a-chip. The review includes a comprehensive description of different manufacturing techniques, and the different organs.
While this review paper offers a wealth of relevant information and spans a wide range of research, its organization is inefficient, and the discussion lacks depth. As a result, several points are repeated throughout the text in similar phrasing. Since the main goal of a review paper is to present comprehensive data well-structured and engagingly, while providing meaningful discussion, this manuscript falls short in delivering on that front. Despite its promising title, the manuscript does not fully meet expectations. The manuscript required some changes before being considered for publication.
Answer: I fully agree with you. I have followed your instructions/comments to perform the required changes of my manuscript. Please see below my point-by-point answers. I have carried out some reorganization of parts of the review article.
Here are some specific comments for improvement:
Include a schematic diagram highlighting the major bioprinting techniques, briefly outlining the key properties of each type as discussed in section 2.
Answer: It is a very good idea. I have included schematic diagrams as the new figures 1 to 4 that highlight the major bioprinting techniques, such as extrusion bioprinting, inject bioprinting, SLA and LIFT.
- In "Subsection 4.2: Benefits and limitations of organoid cultures," the limitations are not addressed as suggested by the title. Please include a discussion on this.
Answer: Yes, I agree. The limitations of organoid cultures are included and discussed.
- Correct the subsection numbering for “3D bioprinting for organoid generation.”
Answer: I am sorry. It is now correct.
- For better clarity, provide a schematic diagram illustrating the various 3D bioprinting platforms for organoid generation (e.g., SPOT, OBB, AAB) as discussed in the relevant sections.
Answer: This is a very good idea. I have prepared the new figure 7 that describes the 3D bioprinting techniques for organoid generation.
- While the authors provide extensive information on techniques and their advantages and disadvantages, there are numerous instances where no relevant examples or references are given. For example, the "one-step manufacturing approach for organ-on-a-chip" is mentioned without citing a research article.
Answer: I have now provided a reference for this example. I have reviewed the entire manuscript to add further references where needed.
- Similarly, section 7 discusses 4D printing in detail but lacks examples.
Answer: I included more examples. It is now section 8.
- The definition of 4D printing is repeated several times in section 7.
Answer: I deleted these repetitions. It is now section 8.
- Furthermore, the statement “For instance, a simple 3D-printed structure can transform into a more complex structure over time” could be supported with examples or references.
Answer: It is now supported by an example and a reference is given.
- In section 8, elaborate on why cell alignment is important. Explain its significance in printed scaffolds and the specific conditions or tissue models that require cell alignment, supported with relevant examples.
Answer: It is now section 9. I agree and included why cell alignment is important and pointed out why it is crucial for 3D printing and outlined its impact in tissue models. Examples are given and cited.
- The authors mention two methods for cell alignment—magnetization and acoustophoresis. Other techniques, such as electrospinning-assisted bioprinting, dielectrophoretic-based bioprinting, and rotary jet spinning, could also be included.
Answer: I included electrospinning-assisted bioprinting, dielectrophoretic-based bioprinting, and rotary jet spinning in section 9. The first is outlined more broadly and the latter two are just listed.
- Despite a promising title, section 9 presents a rather limited discussion. There is little comparison between organoids and organ-on-chip in terms of their suitability for 3D bioprinting. The section is more focused on describing 3D printing techniques (e.g., SPOT, OBB).
Answer: I agree. I have reorganized the now section 7 (7. Combination of organoid bioprinting with partially bioprinted organ-on-a-chip approaches) and parts (e.g. SPOT and OBB) have moved to section 4.4. The discussion is expanded and the comparison of organoids and organ-on-a-chip for their suitability for 3D bioprinting is now included.
- In section 10, the following sentences are unclear and do not lead to meaningful insight: (i) "What are the potential advantages for mechanical studies of single cells within 3D organoids using biomaterials, as represented in 4D models, that respond to stimuli for post-printing modification of in vitro tumor models?" (ii) "How can the implementation of 3D bioprinted tumor models be approached with special consideration of mechanobiological aspects in clinical applications?"
Answer: I have reworded these questions. They are now helpful for the reader.

Reviewer 2 Report
Comments and Suggestions for Authors
This manuscript is a narrative review of three-dimensional (3D) and 4D printing techniques and their applications organoid-based tissue engineering. In particular, it focuses on the fabrication of complex 3D systems for disease modeling and fundamental studies of the biomechanics of organoids and other multicellular constructs.
1. The style of this manuscript is unsatisfactory because the information is poorly organized and barely illustrated.
1a. For example, subsection 2.4 is utterly misleading, even for an experienced reader. Stereolithography (SLA) and laser-induced forward transfer (LIFT) are fundamentally different techniques, so they should belong to different subsections. Here, SLA is described briefly from a historical perspective, then digital light processing is discussed in the context of bioprinting, and, finally, LIFT is described succinctly. The reader rather benefits from an explanation of the principle of each technique accompanied by proper figures.
1b. The title of Section 9 does not reflect the content of the section. The discussion focuses on spatially patterned organoid transfer (SPOT) compared with other bioprinting methods capable of organ building block deposition, such as aspiration-assisted bioprinting. Actually, this title is incorrect because organ-on-chip techniques are not a form of bioprinting; rather, they might include a bioprinting stage.
Here are a few examples of good review papers (I am not a coauthor of any of them):
[1*] Ozbolat IT, Hospodiuk M. Current advances and future perspectives in extrusion-based bioprinting. Biomaterials 2016, 76, 321-343, doi:10.1016/j.biomaterials.2015.10.076.
[2*] Murphy SV, Atala A. 3D bioprinting of tissues and organs. Nat Biotech 2014, 32, 773-785, doi:10.1038/nbt.2958.
[3*] Hospodiuk M, Dey M, Sosnoski D, Ozbolat IT. The bioink: A comprehensive review on bioprintable materials. Biotechnol Adv 2017, 35, 217-239, doi:10.1016/j.biotechadv.2016.12.006.
[4*] Li J, Wu C, Chu PK, Gelinsky M. 3D printing of hydrogels: Rational design strategies and emerging biomedical applications. Materials Science and Engineering: R: Reports 2020, 140, 100543, doi:https://doi.org/10.1016/j.mser.2020.100543.
2. I am afraid this paper is unnecessarily long and complex. Bioprinting techniques have been described in excellent review papers (see, e.g. [1*], [2*]), so section 2 can be safely eliminated. The same is true for bioink precursors ([3*], [4*]) and section 3. Actually, these sections are less refined than the rest of the manuscript, so the overall quality of the paper would be higher in their absence.
A good example of a vast but not redundant review paper is [4*] listed above.
3. Regarding the illustrations, the strong points of this manuscript are Figure 1 (information-rich and well-constructed), and Table 4 (applicative, well-documented, but too verbose). The other illustrations are either too abstract or too narrow in their scope.
3a. Table 1 deals with extrusion-based bioprinting (EBB). The problem is that EBB is just a small fraction of the paper's topic and is already covered in the secondary literature (see, e.g., [1*] above). Instead, the author could have included tables of recent developments in a variety of fields discussed in the main text.
3b. Table 2 is completely useless, since the field of biomaterials developed for bioprinting is vast [3*]. In my opinion, it is hopeless to present it in tabular form.
3c. Table 3 is extremely narrow in scope.
4. Parts of the text are hastily written, with little or no revision. For instance, lines 195-197 are a repetition of the previous sentence.
5. Certain citations are questionable because they do not point to the most relevant original work. Regarding 4D printing (line 2117) the author discusses the third law of bioprinting, originally formulated by Farhang Momeni ( Momeni, F.; Ni, J. Laws of 4D Printing. Engineering 2020, doi:https://doi.org/10.1016/j.eng.2020.01.015.) and cites a review article, [550]. A review of reviews is of little help to the reader.
Minor comments
Since many acronyms are used in this text, from a variety of research fields, a list of abbreviations would be helpful. I would use acronyms sparingly (e.g. TBARS is defined on line 1552 to be used only once, on line 1554).
Line 214: the extrusions bioprinting technique => extrusion bioprinting
Line 386: Steriolithography => Stereolithography
Line 2117: The quote starts on this line but it is not closed.
Line 2604: Please delete the Acknowledgments paragraph or replace the explicative text with your own.
Comments on the Quality of English LanguageThe language of this paper is a bit verbose (e.g. lines 1195-99, 1282-84, 1364-66, 2400-02, 2459-62) but mainly clear. Still, certain sentences are confusing, especially when they are too general and cite multiple papers (e.g. lines 1233-36, 1387-88, 2389-90).
Author Response
Point-by-point answers to reviewers’ comments
I thank both reviewers for their excellent comments that really helped me to improve my review article. All changes are highlighted in yellow in the manuscript. For my detailed response to each comment see below.
Best regards
Claudia Tanja Mierke
Reviewer 2
This manuscript is a narrative review of three-dimensional (3D) and 4D printing techniques and their applications organoid-based tissue engineering. In particular, it focuses on the fabrication of complex 3D systems for disease modeling and fundamental studies of the biomechanics of organoids and other multicellular constructs.
The style of this manuscript is unsatisfactory because the information is poorly organized and barely illustrated.
Answer: I altered the organization partly to address this issue and I added 5 more figures.
1a. For example, subsection 2.4 is utterly misleading, even for an experienced reader. Stereolithography (SLA) and laser-induced forward transfer (LIFT) are fundamentally different techniques, so they should belong to different subsections. Here, SLA is described briefly from a historical perspective, then digital light processing is discussed in the context of bioprinting, and, finally, LIFT is described succinctly. The reader rather benefits from an explanation of the principle of each technique accompanied by proper figures.
Answer: I fully agree and there is now a new subsection for LIFT (2.4). The section with SLA is rewritten and expanded by a principle. A new figure 3 for SLA and a new figure 4 for LIFT are provided. Section 2.1 and 2.2 have been combined.
1b. The title of Section 9 does not reflect the content of the section. The discussion focuses on spatially patterned organoid transfer (SPOT) compared with other bioprinting methods capable of organ building block deposition, such as aspiration-assisted bioprinting. Actually, this title is incorrect because organ-on-chip techniques are not a form of bioprinting; rather, they might include a bioprinting stage.
Answer: I fully agree. I have reorganized section 7 and moved parts of it to section 4.4. I clarified the organ-on-chip technique that it comprises partly a bioprinting stage. The title of the section 7 is modified.
Here are a few examples of good review papers (I am not a coauthor of any of them):
[1*] Ozbolat IT, Hospodiuk M. Current advances and future perspectives in extrusion-based bioprinting. Biomaterials 2016, 76, 321-343, doi:10.1016/j.biomaterials.2015.10.076.
[2*] Murphy SV, Atala A. 3D bioprinting of tissues and organs. Nat Biotech 2014, 32, 773-785, doi:10.1038/nbt.2958.
[3*] Hospodiuk M, Dey M, Sosnoski D, Ozbolat IT. The bioink: A comprehensive review on bioprintable materials. Biotechnol Adv 2017, 35, 217-239, doi:10.1016/j.biotechadv.2016.12.006.
[4*] Li J, Wu C, Chu PK, Gelinsky M. 3D printing of hydrogels: Rational design strategies and emerging biomedical applications. Materials Science and Engineering: R: Reports 2020, 140, 100543, doi:https://doi.org/10.1016/j.mser.2020.100543.
Answer: I agree. I have seen that they are more complex and really deep in the field. For the advanced reader, I have cited them now in my review article (see also below).
- I am afraid this paper is unnecessarily long and complex. Bioprinting techniques have been described in excellent review papers (see, e.g. [1*], [2*]), so section 2 can be safely eliminated. The same is true for bioink precursors ([3*], [4*]) and section 3. Actually, these sections are less refined than the rest of the manuscript, so the overall quality of the paper would be higher in their absence.
Answer: The point is justified here, but I have structured my entire manuscript in such a way that both sections (2 and 3) are necessary for understanding the other sections, in my opinion. Therefore, I decided to keep them in my review article, even though they may reduce the overall quality of my review article. I still hope that this will not be the case. In addition, Revier 1 suggests illustrating section 2 with images, which I have now done. The sections are aimed at the cell biophysicist as a reader who is not familiar with the issues of bioprinting. Therefore, it is more or less a very basic and simple introduction to bioprinting techniques and bioink precursors. The bioprinting techniques can mechanically disturb the printed cells and the bioinks can also have an influence on the mechanical cues of the cells and organoids. I have rewritten parts of section 2 for clarification subsection 2.1 and 2.2 are combined. SLA and LIFT are now different sections.
A good example of a vast but not redundant review paper is [4*] listed above.
Answer: I have carefully read and looked at the structure of these review articles. They all seemed to be reviewers from a more bioengineering perspective and rather not from a biophysical perspective (such as my review). However, they are excellent, and I decided to make clear that I have a different approach for my review. In addition, I have eliminated redundant parts of my review.
- Regarding the illustrations, the strong points of this manuscript are Figure 1 (information-rich and well-constructed), and Table 4 (applicative, well-documented, but too verbose). The other illustrations are either too abstract or too narrow in their scope.
Answer: I see your concern and I agree. I have included 5 new figures. I eliminated table 1, table 2 and table 3. I altered table 4 (now table 1) as suggested. The figure 3 is slightly adapted, but still nice to be included there. Figure 2 provides an overview and stays in the manuscript. The new table 1 is slightly reworded.
3a. Table 1 deals with extrusion-based bioprinting (EBB). The problem is that EBB is just a small fraction of the paper's topic and is already covered in the secondary literature (see, e.g., [1*] above). Instead, the author could have included tables of recent developments in a variety of fields discussed in the main text.
Answer: I have deleted table 1. I included 5 new figures to illustrate different bioprinting techniques for the reader.
3b. Table 2 is completely useless, since the field of biomaterials developed for bioprinting is vast [3*]. In my opinion, it is hopeless to present it in tabular form.
Answer: I have deleted table 2.
3c. Table 3 is extremely narrow in scope.
Answer: I have deleted table 3. I included 5 new figures to illustrate different bioprinting techniques for the reader.
- Parts of the text are hastily written, with little or no revision. For instance, lines 195-197 are a repetition of the previous sentence.
Answer: It is now deleted. I have also checked the entire manuscript.
- Certain citations are questionable because they do not point to the most relevant original work. Regarding 4D printing (line 2117) the author discusses the third law of bioprinting, originally formulated by Farhang Momeni ( Momeni, F.; Ni, J. Laws of 4D Printing. Engineering 2020, doi:https://doi.org/10.1016/j.eng.2020.01.015.) and cites a review article, [550]. A review of reviews is of little help to the reader.
Answer: I totally agree and I have cited now Momeni and 2020 (doi:https://doi.org/10.1016/j.eng.2020.01.015) here.
Minor comments
Since many acronyms are used in this text, from a variety of research fields, a list of abbreviations would be helpful. I would use acronyms sparingly (e.g. TBARS is defined on line 1552 to be used only once, on line 1554).
Answer: I have written TBARS in full, as it is not necessary to use an acronym. A list of abbreviations is provided in the manuscript.
Line 214: the extrusions bioprinting technique => extrusion bioprinting
Answer: It is a typo. It is now corrected.
Line 386: Steriolithography => Stereolithography
Answer: It is a typo. It is now corrected.
Line 2117: The quote starts on this line but it is not closed.
Answer: It is now included.
Line 2604: Please delete the Acknowledgments paragraph or replace the explicative text with your own.
Answer: I am sorry, I have now replaced it with my acknowledgement.
English
The language of this paper is a bit verbose (e.g. lines 1195-99, 1282-84, 1364-66, 2400-02, 2459-62) but mainly clear. Still, certain sentences are confusing, especially when they are too general and cite multiple papers (e.g. lines 1233-36, 1387-88, 2389-90).
Answer: I have rephrased the “bit verbose” sentences and reworded the “confusing” sentences. I have also changed citations.

Round 2
Reviewer 2 Report
Comments and Suggestions for Authors
The author did a good job addressing the concerns expressed in my previous referee report, except for the advice to shorten the paper. I respect the option of the author and appreciate her effort to better illustrate sections 2 and 3.
The manuscript has been improved considerably. Hopefully, it will elicit interest from the cell biology and biophysics community.